# Multifunctional nanoparticle potentiates the in situ vaccination effect of radiation therapy and enhances response to immune checkpoint blockade

Ying Zhang[1,2,8], Raghava N. Sriramaneni[3,8], Paul A. Clark[3], Justin C. Jagodinsky[3], Mingzhou Ye[1,2], Wonjong Jin [3], Yuyuan Wang[1,2], Amber Bates[3], Caroline P. Kerr [3,4], Trang Le[5], Raad Allawi[3], Xiuxiu Wang[1,2], Ruosen Xie [1,2], Thomas C. Havighurst[5], Ishan Chakravarty[3], Alexander L. Rakhmilevich[3], Kathleen A. O'Leary[6], Linda A. Schuler[6], Paul M. Sondel[3,7], Kyungmann Kim[5], Shaoqin Gong [1,2] ✉ & Zachary S. Morris [3] ✉

Radiation therapy (RT) activates an in situ vaccine effect when combined with immune checkpoint blockade (ICB), yet this effect may be limited because RT does not fully optimize tumor antigen presentation or fully overcome suppressive mechanisms in the tumor-immune microenvironment. To overcome this, we develop a multifunctional nanoparticle composed of polylysine, iron oxide, and CpG (PIC) to increase tumor antigen presentation, increase the ratio of M1:M2 tumor-associated macrophages, and enhance stimulation of a type I interferon response in conjunction with RT. In syngeneic immunologically "cold" murine tumor models, the combination of RT, PIC, and ICB significantly improves tumor response and overall survival resulting in cure of many mice and consistent activation of tumor-specific immune memory. Combining RT with PIC to elicit a robust in situ vaccine effect presents a simple and readily translatable strategy to potentiate adaptive anti-tumor immunity and augment response to ICB or potentially other immunotherapies.

Cancer immunotherapy has revolutionized clinical oncology and immune checkpoint blockade (ICB) therapies have become one of the most effective cancer treatments[1,2]. However, many cancer patients do not respond to this treatment, particularly those with immunologically "cold" tumors that are commonly characterized by low neoantigen load and limited immune cell infiltration[3,4]. In situ cancer vaccination is a therapeutic strategy that aims to convert a patient's own tumor into a nidus for presentation of tumor-specific antigens in a way that will stimulate and diversify an anti-tumor T cell response[5]. Such approaches may improve the response rates for ICB therapies by promoting antigen presentation and tumor-specific T cell response in immunologically "cold" tumors[5,6].

Radiation therapy (RT) is one of the most common cancer treatment modalities and RT is delivered to ~2/3 of cancer patients at some

[1]Department of Biomedical Engineering and Wisconsin Institute for Discovery, University of Wisconsin-Madison, Madison, WI, USA. [2]Department of Ophthalmology and Visual Sciences, University of Wisconsin-Madison, Madison, WI, USA. [3]Department of Human Oncology, University of Wisconsin-Madison, Madison, WI, USA. [4]Department of Radiology, University of Wisconsin-Madison, Madison, WI, USA. [5]Department of Biostatistics and Medical Informatics, University of Wisconsin-Madison, Madison, WI, USA. [6]Department of Comparative Biosciences, University of Wisconsin-Madison, Madison, WI, USA. [7]Department of Pediatrics, University of Wisconsin-Madison, Madison, WI, USA. [8]These authors contributed equally: Ying Zhang, Raghava N. Sriramaneni. ✉e-mail: shaoqingong@wisc.edu; zmorris@humonc.wisc.edu

point in their care. RT with a dose of 8–12 Gy has been demonstrated in preclinical and clinical studies to be capable of activating an in situ vaccine response[7–11]. By stimulating immunogenic cell death, increasing tumor infiltration by immune cells, and enhancing the susceptibility of tumor cells to immune-mediated killing, RT has been shown in some preclinical models to augment response to ICBs[12–14]. Clinical studies indicate that such effects may be achieved in cancer patients but also suggest that more potent therapy will be needed if we aim to activate anti-tumor immune responses in combination with ICBs for the majority of patients with metastatic cancers[15–17]. This may reflect the fact that although radiation may elicit many effects in the tumor microenvironment that are conducive to augmenting the development and propagation of adaptive anti-tumor immunity, these effects may be weak or incomplete and radiation could also activate certain detrimental effects such as the recruitment, polarization, or activation of suppressive immune cell lineages including M2 macrophages[15,18].

Low effector T cell infiltration is one hallmark of immunologically "cold" cancers[3]. In addition to limited neoantigen recognition, immunosuppressive cells in the tumor microenvironment contribute to this "cold" phenotype and diminish the efficacy of cancer immunotherapies[19]. Tumor-associated macrophages (TAMs) and their precursors constitute the major myeloid population of the tumor microenvironment for many solid tumors[20,21]. TAMs are plastic and commonly dichotomized into two phenotypes[20]. Pro-inflammatory TAMs (M1 macrophages) can function as tumoricidal macrophages and support the efficacy of many cancer immunotherapies, whereas anti-inflammatory TAMs (M2 macrophages) predominate in many tumors and can create an immunosuppressive microenvironment that promote immune evasion[22–24].

To accentuate the capacity of RT to elicit in situ vaccination in immunologically "cold" tumors that do not respond to ICBs alone, we hypothesized that it would be beneficial to combine RT with a therapeutic agent that could not only augment the favorable effect of RT in activating T cell immunity, but also could modulate the irradiated tumor microenvironment to ameliorate some of the potentially detrimental effects of RT. For this, we developed a multifunctional nanoparticle with a scalable manufacturing protocol.

Here, we report the development and preclinical testing of this nanoparticle, which we designed specifically to enhance the in situ vaccine effect of RT and to prime a more potent systemic anticancer immune response against immunologically "cold" tumors to render these tumors responsive to ICBs. Our results suggest that this approach may offer a simple and effective strategy that allows use of off-the-shelf treatment components to achieve a personalized in situ vaccine effect that might transform a patient's tumor into a site for activation of an adaptive T cell response against that patient's own cancer cells.

## Results

### Preparation of PIC and its immune modulation in vitro

Using a rational and mechanism-based design process, we conceptualized a multifunctional nanoparticle that could be manufactured following readily scalable and reproducible methods (Fig. 1a). This multifunctional nanoparticle (PIC) was composed of poly-(L-lysine) (PLL), CpG oligodeoxynucleotide (CpG), and iron oxide nanoparticle (ION). PIC was designed to serve as a radiation sensitizer using the radiation enhancing effects of ION to improve RT efficiency, capture the tumor-associated antigens (TAAs) released from dying tumor cells via electrostatic interaction and facilitate their uptake by antigen-presenting cells (APCs) to activate CD8+ and CD4+ T cell responses, stimulate Toll-like receptor-9 (TLR-9) using CpG, enhance type-I interferon (IFN-I) production, and promote a more favorable ratio of M1: M2 polarization of TAMs following RT[9,25–30].

ION was synthesized with an Fe content of 64.1% measured by inductively coupled plasma-optical emission spectrometry (ICP-OES)

(Supplementary Fig. 1)[31]. To prepare PIC via a scalable and straightforward complexation process, the negatively charged ION was first mixed with positively charged PLL, which was subsequently complexed with negatively charged CpG. Nanoparticles with different particle sizes and zeta potentials were obtained by varying the weight ratio between ION, PLL and CpG (Supplementary Table 1). Because positively charged nanoparticles with high zeta potentials can interact strongly with negatively charged TAAs[27], we chose a PIC with a weight ratio of ION/PLL/CpG as 4.8/3.5/1 for further study, as this particle exhibited maximal zeta potentials for particle sizes deemed suitable for this application (Fig. 1b–d). Agarose gel electrophoresis assay confirmed highly efficient loading of CpG on the PIC (Fig. 1e). The PIC nanoparticle was stable at 4 °C at least for 30 days (Fig. 1f and Supplementary Fig. 2a). And the lyophilized PIC showed negligible changes in particle size and zeta potential during storage at −20 °C for at least 12 weeks (Supplementary Fig. 2b, c).

To test the protein absorption ability of PIC, cell lysates were prepared from B78 murine melanoma cells. As shown in Fig. 1g, incubation with PIC (0.14 mg/mL) caused a significant decrease of protein concentration in the tumor cell lysates, indicating the strong antigen capture ability of PIC. PIC did not show any direct cytotoxicity to either murine macrophage cell line RAW264.7 or B78 melanoma cells when the concentration was <17.5 μg/mL (Supplementary Fig. 3). Upon analysis by both flow cytometry and confocal laser scanning microscopy (CLSM), PIC showed a dose-dependent cellular uptake in RAW264.7 macrophages, and cellular uptake was also observed at higher dose in B78 melanoma cells although this was only statistically significant at 7.5 μg/mL (Supplementary Figs. 4, 5). Prior studies demonstrate that iron oxide nanoparticles can sensitize tumor cells to radiation[25,26]. Using the in vitro clonogenic assay, a standard approach to quantifying radiosensitivity, we confirmed that PIC significantly increased the sensitivity of B78 melanoma cells to RT (Fig. 1h). Immunofluorescence microscopy quantifying γH2AX foci confirmed that this effect in B78 melanoma cells correlated with a role of PIC enhancing the DNA damage resulting from RT, although the PIC alone did not directly induce DNA damage, as expected (Fig. 1i, j).

Activation of an IFN-I response via the cGAS/STING pathway in tumor cells has been shown to be critical for the effects of RT in stimulating an in situ vaccine effect and potentiating response to ICBs[32,33]. We previously observed that the activation of IFN-I via cGAS/STING pathway in B78 cells peaked at day 7 post-RT[34]. In B78 melanoma, PIC alone did not significantly influence the expression of *Ifnβ1*, a marker of cGAS/STING activation of an IFN-I response (Fig. 2a, b), and elicited negligible effects on the *Ifnβ1* expression in the radiated B78 cells when it was added after RT. However, pre-treatment with PIC before RT significantly increased the expression of *Ifnβ1* in B78 cells at day 7 post-treatment compared to RT alone (Fig. 2b and Supplementary Fig. 6a). Although RT is also noted to increase expression of *Pd-l1* in tumor cells[34,35], PIC did not influence *Pd-l1* expression in B78 cells at tested time points after RT (Fig. 2c and Supplementary Fig. 6b).

Next, we evaluated the effect of PIC on the polarization of bone marrow derived macrophages (BMDMs) (Fig. 2d). RT significantly upregulated the expression of CD206 and CD163, two markers of M2 macrophages, in CD11b+F4/80+ BMDMs at day 4 after treatment, while PIC significantly downregulated expression of these markers and increased the expression ratios of CD80: CD206 and CD80: CD163 in both non-radiated and radiated CD11b+F4/80+ BMDMs (Fig. 2e–g and Supplementary Fig. 7). Consequently, PIC treatment reduced the percentage of CD206+CD80− M2-like macrophages and enhanced the percentage of CD80+CD206− M1-like macrophages among CD11b+F4/80+ BMDMs at day 4 after RT, confirming the PIC achieved its intended effects on macrophage polarization (Fig. 2h and Supplementary Fig. 8).

We evaluated the capacity of PIC to activate dendritic cells (DCs) and enhance antigen presentation. Mouse DCs were isolated from the spleen and treated with PIC in vitro. We found PIC treatment markedly

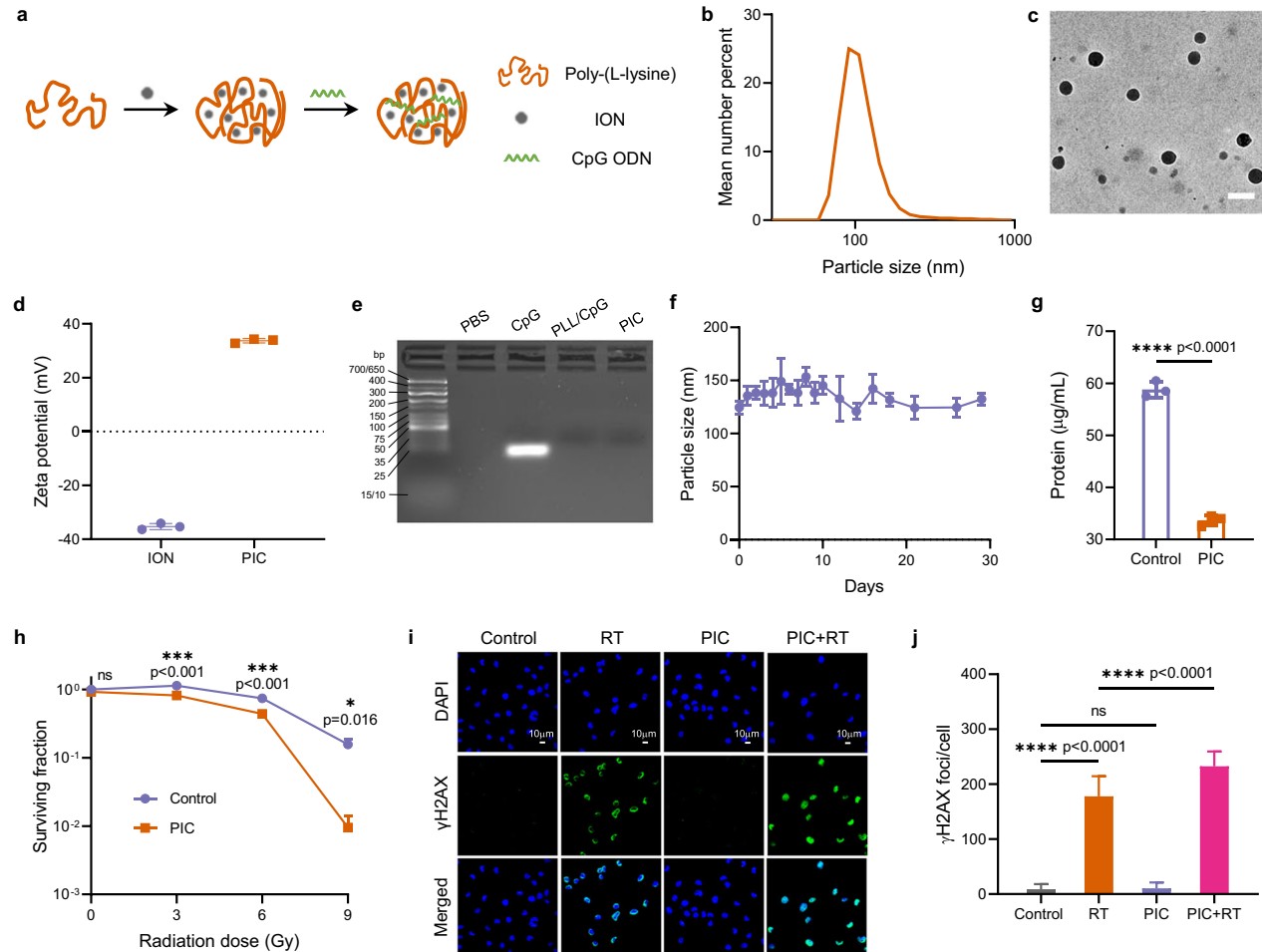

**Fig. 1 | Preparation and characterization of PIC. a** The preparation process of PIC. **b** Hydrodynamic size distribution of PIC. **c** Transmission electron microscopy image of PIC. Scale bar: 100 nm. **d** Zeta potentials of ION and PIC. ($n = 3$ independent samples). **e** CpG complexation abilities in different formulations, as measured by agarose gel electrophoresis. **f** The particle size stability of PIC during storage at 4 °C. ($n = 3$ independent samples). **g** The protein concentrations in B78 cell lysates after incubation with PIC (0.14 mg/mL) for 4 h. ($n = 3$ biologically independent samples). **h** Clonogenic assay of B78 melanoma cells after treatment with PIC (4.67 μg/mL) and indicated radiation doses. ($n = 3$ biologically independent samples). PIC was added to the cells 4 h before radiation, and fresh culture media was exchanged for this PIC treatment media 1 h after radiation. The colonies were counted at day 7. **i** The immunofluorescence images of B78 cells after indicated treatment (RT: 12 Gy; PIC: 4.67 μg/mL). **j** Quantification of foci of γH2AX as shown in **i**. 50 cells in each group were analyzed with ImageJ. PIC was added to the cells 4 h before radiation, and fresh culture media was exchanged for PIC treatment media 1 h after radiation. The immunofluorescence images were taken 1 h after radiation. Statistical significance was calculated via unpaired *t*-test in **g**, and one-way ANOVA test in **h** and **j**. Data in **d**, **f**, **g**, **h** and **j** are presented as mean ± SD. *$p < 0.05$, **$p < 0.01$, ***$p < 0.001$ and ****$p < 0.0001$. A representative image of three independent samples from each group is shown in **c**, **e** and **i**. Source data are provided in Source Data file.

increased the expression of CD80 and CD86 on the surface of CD11c[+]CD317[+] plasmacytoid dendritic cells (pDCs), which express endosomal TLR-9 that is activated by CpG (Fig. 2i)[36,37]. PIC-activated DCs exhibited production of IFN-β, suggesting the potential for PIC to directly influence the tumor immune microenvironment at an injected site (Supplementary Fig. 9a). To test the ability of PIC to improve the antigen uptake by APCs, we treated DCs with pre-complexed PIC and FITC-labeled ovalbumin (FITC-Ova) and compared the internalization of FITC-Ova in CD11c[+] cells with those treated with FITC-Ova alone. PIC significantly improved the cellular uptake of FITC-Ova by CD11c[+] DCs (Fig. 2j and Supplementary Fig. 9b). To determine whether the effects of PIC on antigen-presenting cells would augment antigen presentation and enhance T cell immunity, we injected healthy mice with pre-complexed Ova/PIC, Ova/CpG/ION, or Ova alone. We then isolated splenocytes from these mice and co-cultured these with B16-SIINFEKL cells (Supplementary Fig. 10a). Both CD4[+] and CD8[+] T cells from splenocytes of mice injected with Ova/PIC showed higher expression of CD69 (early T cell activation marker), CD44 (effector T cell marker),

and IFNγ (key mediator of T cell activation) when compared to those from Ova or Ova/CpG/ION injected mice (Supplementary Figs. 10b, c and 11). These results demonstrate that PIC can promote DC activation and antigen cross presentation enabling robust T cell immunity.

## The immune modulation by PIC in vivo and its effects on the tumor infiltrating immune cells

Before studying the ability of PIC to immunomodulate radiated tumor microenvironments, we evaluated its retention in tumor after intratumoral injection and its cellular uptake by tumor cells, macrophages and DCs in vivo. Three days after the Cy5-labeled PIC (Cy5-PIC) was intratumorally injected into B78 melanoma flank tumors, we observed strong fluorescence signal of Cy5 in the tumor sites (Supplementary Fig. 12). Using flow cytometry on tumors disaggregated 3 h after intratumoral injection of Cy5-PIC, we found Cy5-PIC was endocytosed and retained by tumor cells, DCs (CD11c[+]MHCII[+]CD45[+]), and macrophages (CD11b[+]F4/80[+]CD45[+]) (Fig. 3a and Supplementary Fig. 13). Notably, we also observed Cy5-PIC in DCs (CD11c[+]MHCII[+]CD45[+]) and

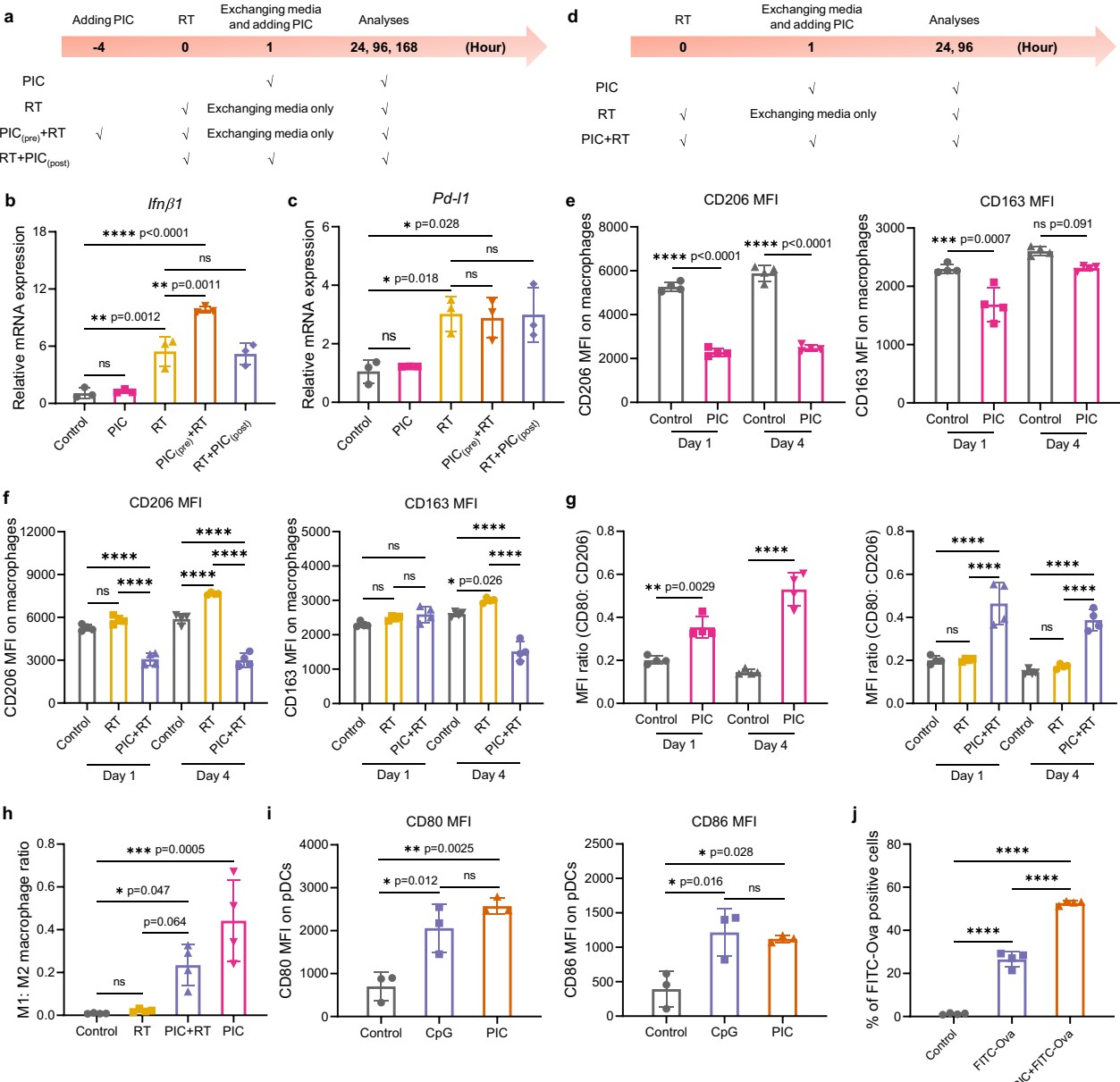

**Fig. 2 | The immunomodulatory effect of PIC in vitro. a** Scheme for the RT-qPCR studies of B78 cells. The relative mRNA expression of **b** *Ifnβ1* and **c** *Pd-l1* in B78 cells at day 7 after initiation of indicated treatments. PIC: 4.67 μg/mL; RT: 12 Gy. (*n* = 3 biologically independent samples). **d** Scheme for the study of the macrophage polarization effect of PIC on BMDMs. The MFI of CD206 and CD163 on CD11b⁺F4/80⁺ BMDMs at day 1 and day 4 after the cells were treated with **e** PIC and **f** RT + PIC. **g** The ratios of CD80 MFI to CD206 MFI on CD11b⁺F4/80⁺ BMDMs at day 1 and day 4 after indicated treatment. **h** The ratios of M1: M2 macrophage in CD11b⁺F4/80⁺ BMDMs at day 4 after indicated treatment. M1-like: CD80⁺CD206⁻; M2-like:

CD206⁺CD80⁻. PIC: 4.67 μg/mL; RT: 12 Gy. (*n* = 4 biologically independent samples). **i** The MFI of CD80 and CD86 on CD317⁺CD11c⁺ pDCs after indicated treatments. PIC: 4.67 μg/mL; CpG: 0.5 μg/mL. (*n* = 3 biologically independent samples). **j** The percentage of FITC-Ova positive cells among CD11c⁺ DCs at 24 h after treatment. FITC-Ova: 1.67 μg/mL; PIC: 4.67 μg/mL. (*n* = 4 biologically independent samples). DCs were enriched from the splenocytes collected from Flt3L treated C57BL/6 mice. Statistical significance was calculated via one-way ANOVA test in **b**, **c** and **e–j**, and data are presented as mean ± SD. *$p < 0.05$, **$p < 0.01$, ***$p < 0.001$ and ****$p < 0.0001$. Source data are provided in Source Data file.

macrophages (CD11b⁺F4/80⁺CD45⁺) upon disaggregation of tumor-draining lymph nodes (TDLNs) after intratumoral injection (Fig. 3b)[38]. 

By analyzing bulk tumor mRNA using RT-qPCR, we evaluated the immunomodulatory effect of RT and PIC + RT on B78 tumors (Fig. 3c and Supplementary Fig. 14). Consistent with our in vitro data (Fig. 2b), at day 7 after RT we observed increased expression of *Ifnβ1* and *Mx1*, markers of cGAS/STING activation, in tumors treated with PIC + RT compared to those treated with RT alone (Supplementary Fig. 15). This effect was amplified further at day 15 post-treatment (Fig. 3d). From the data shown in Supplementary Fig. 16, PIC injection alone also enhanced the expression of *Ifnβ1* and *Mx1* in B78 tumors, which may

result from the activation of TLR-9 in endocytic vesicles of antigen-presenting cells by the CpG in PIC. Nitric oxide synthase 2 (*Nos2*) and arginase 1 (*Arg1*) play important roles in regulating the functions of macrophages and serve as transcriptional markers for M1 and M2 polarization, respectively[39]. As shown in Fig. 3d, both *Arg1* and *Nos2* mRNA levels increased in the RT + PIC group compared to the RT alone treatment group at day 15, but the magnitude of *Nos2* mRNA elevation was far greater than that of *Arg1* mRNA (76-fold elevation for *Arg1* vs 490-fold elevation for *Nos2*), resulting in a higher ratio of *Nos2*: *Arg1* in the PIC + RT group compared to the RT group (Fig. 3e). IFN-γ is produced by immune cells and promotes the activity of T helper

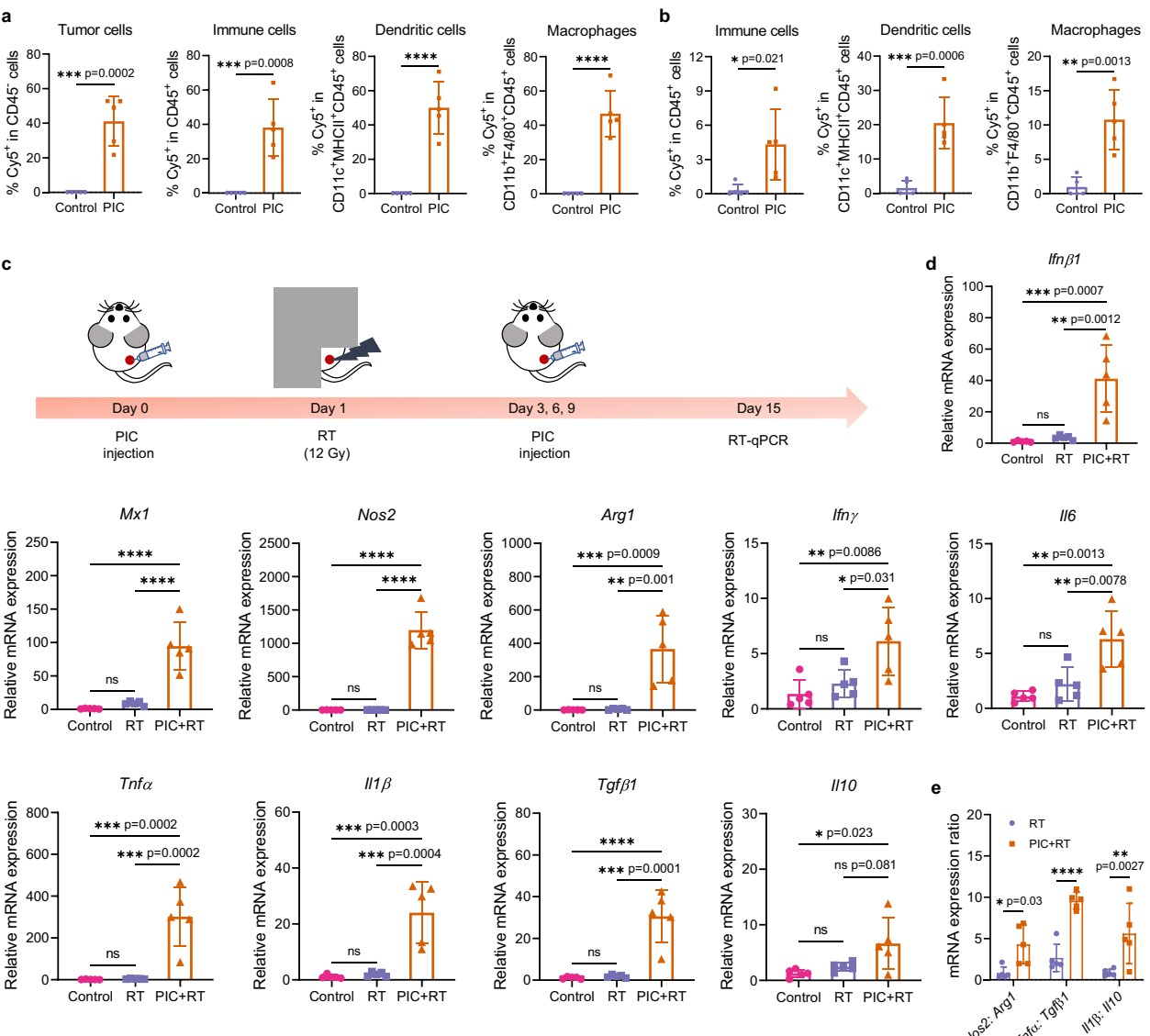

**Fig. 3 | The biodistribution and immunomodulatory effects of PIC in vivo.** The cellular uptake of Cy5-PIC in different cells within the **a** tumors and **b** TDLNs after the Cy5-PIC was intratumorally injected into B78 melanoma bearing mice. (*n* = 5 biologically independent samples). The gating strategy is provided in Supplementary Fig. 13 in Supplementary Information. **c** Scheme for the study of the immunomodulation effect of PIC + RT on the B78 tumor microenvironment. **d** The mRNA expression of select inflammatory and anti-inflammatory genes (*Ifnβ1, Mx1,*

*Nos2, Arg1, Ifnγ, Il6, Tnfα, Il1β, Il10* and *Tgfβ1*) in B78 tumors at day 15 after initiation of indicated treatments. **e** The mRNA expression ratios of *Nos2: Arg1, Tnfα: Tgfβ1* and *Il1β: Il10* in tumors at day 15 after indicated treatments. PIC: 140 μg/100 μL/ dose; RT: 12 Gy. (*n* = 5 biologically independent samples). Data in **a**, **b** and **d**, **e** are shown as mean ± SD. Statistical significance was calculated via unpaired *t*-test in **a**, **b**, and one-way ANOVA test in **d**, **e**. **p* < 0.05, ***p* < 0.01, ****p* < 0.001 and *****p* < 0.0001. Source data are provided in Source Data file.

type 1 cells and CD8+ cytotoxic T lymphocytes[40]. We found that *Ifnγ* expression increased in B78 tumors at day 15 after PIC + RT treatment compared to RT alone (Fig. 3d). PIC also increased the expression of both pro-inflammatory genes (*Tnfα* and *Il1β*) and anti-inflammatory genes (*Tgfβ1, Il10* and *Il6*) (Fig. 3d). However, the elevation was greater for pro-inflammatory genes, leading to higher ratios of pro-inflammatory genes: anti-inflammatory genes (*Tnfα*: *Tgfβ1* and *Il1β*: *Il10*) for the PIC + RT group compared to the RT alone group (Fig. 3e).

Next, we examined the effects of PIC on the populations of tumor-infiltrating immune cells in radiated B78 melanoma tumors (Fig. 4a, Supplementary Figs. 17–19 and Supplementary Tables 2, 3). We found that tumor infiltration by TAMs (CD11b+F4/80+) as a percentage of myeloid cells (CD45+CD11b+) was not significantly influenced by the RT or PIC + RT treatment (Supplementary Fig. 20a, b), and the frequency of M1 macrophages (CD80+CD206−) among TAMs in radiated tumors was similar to that in PIC + RT group, with both exhibiting higher

percentage of M1 macrophage when compared to untreated control mice (Fig. 4b and Supplementary Fig. 20c). On the other hand, we observed an elevation in the relative abundance of M2 macrophages (CD206+CD80−) in tumor following RT alone, but this effect was reduced when RT was combined with PIC injection (Fig. 4c and Supplementary Fig. 20c). As a result, PIC + RT treatment significantly increased the M1:M2 macrophage ratio when compared to either the untreated control group or the RT alone group (Fig. 4d). This macrophage polarization effect of PIC on radiated tumors was further verified by the higher expression ratios of CD80: CD206 on TAMs from PIC + RT treated tumors compared to those from tumors treated with RT alone (Supplementary Fig. 20d).

DCs are critical to enable adaptive T cell immunity against cancer cells. cDC1s promote the cross-presentation of tumor antigens to CD8+ T cells and are critical for the generation of cytotoxic effector T cell responses, while cDC2s are involved in the activation of T helper type 2

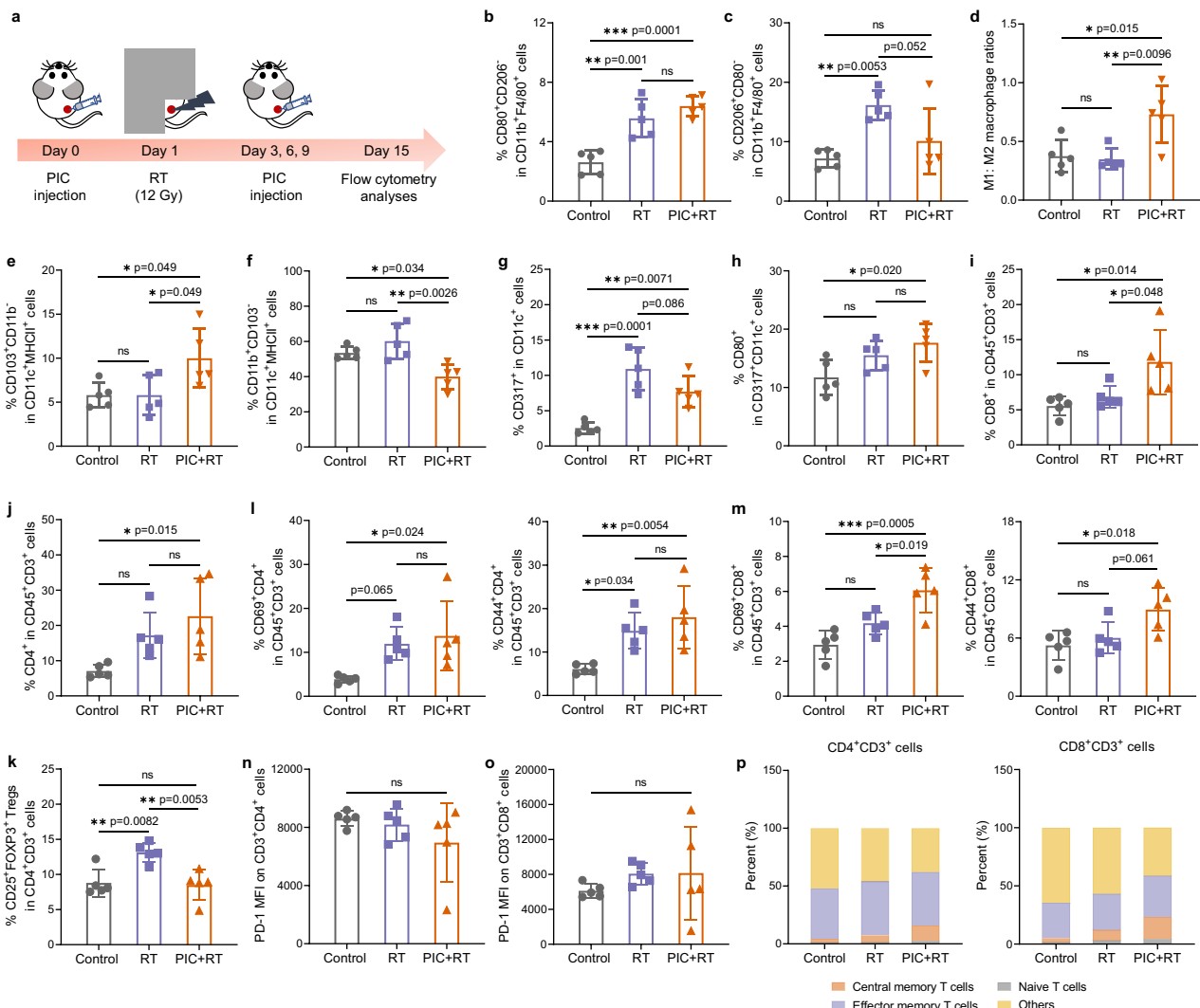

**Fig. 4 | The influence of PIC on tumor-infiltrating immune cell populations. a** Scheme for the studies. The percentage of **b** CD80⁺CD206⁻ M1-like macrophage and **c** CD206⁺CD80⁻ M2-like macrophage in CD11b⁺F4/80⁺ macrophages in B78 tumors. **d** The ratios of M1:M2 macrophages in B78 tumors after indicated treatments. The percentage of **e** CD103⁺CD11b⁻ cDC1s and **f** CD11b⁺CD103⁻ cDC2s in CD11c⁺MHCII⁺ DCs in B78 tumors. **g** The percentage of CD317⁺ pDCs in CD11c⁺ cells in B78 tumors. **h** The percentage of CD80⁺ cells in CD317⁺ CD11c⁺ pDCs in B78 tumors. The percentage of **i** CD8⁺ and **j** CD4⁺ cells in CD45⁺CD3⁺ T cells in B78 tumors. **k** The percentage of CD25⁺FOXP3⁺ Tregs in CD4⁺ T cells in B78 tumors. **l** The percentage of CD69⁺CD4⁺ and CD44⁺CD4⁺ cells in CD45⁺CD3⁺ T cells in B78

tumors. **m** The percentage of CD69⁺CD8⁺ and CD44⁺CD8⁺ cells in CD45⁺CD3⁺ T cells in B78 tumors. The MFI of PD-1 on **n** CD3⁺CD4⁺ T cells and **o** CD3⁺CD8⁺ T cells in B78 tumors. (*n* = 5 biologically independent samples for **b**–**o**). **p** The percentage of central memory T cells (CD44⁺CD62L⁺), effector memory T cells (CD44⁺CD62L⁻), naive T cells (CD44⁻CD62L⁺) and others (CD44⁻CD62L⁻) among CD3⁺CD4⁺ T cells (left) and CD3⁺CD8⁺ T cells (right) in TDLNs. PIC: 140 μg/100 μL/dose; RT: 12 Gy. The gating strategies are provided in Supplementary Figs. 18–19 in Supplementary Information. Statistical significance was calculated via one-way ANOVA test in **b**–**o**, and data are shown as mean ± SD. *\*p* < 0.05, *\*\*p* < 0.01, *\*\*\*p* < 0.001 and *\*\*\*\*p* < 0.0001. Source data are provided in Source Data file.

cells, and pDCs highly express TLR-9 and are copious producers of IFN-I when TLR-9 is stimulated[36,41,42]. By analyzing the abundance of DC subtypes in the tumor microenvironment, we found that PIC + RT treatment increased the levels of CD103⁺CD11b⁻ cDC1s and decreased the levels of CD11b⁺CD103⁻ cDC2s in B78 melanoma tumors when compared with either untreated control or RT-alone treated tumors (Fig. 4e, f). This phenomenon was also observed in the TDLNs, which exhibited a higher level of cDC1s and lower level of cDC2s in the PIC + RT group compared to RT-alone or untreated groups (Supplementary Fig. 21). RT significantly enhanced the abundance of tumor-infiltrating CD317⁺CD11c⁺ pDCs and the injection of PIC into the radiated tumors increased the percentage of CD80⁺ cells, one of the markers of TLR-9 activation, among pDCs, when compared to the untreated control group (Fig. 4g, h)[36,43].

In evaluating tumor infiltration by adaptive immune cells, we observed that RT and PIC + RT treatment did not affect the number of

CD3⁺ T cells in tumors (Supplementary Fig. 22), yet PIC + RT treated tumors showed significantly higher infiltration of CD8⁺ T cells compared to tumors treated with RT alone (Fig. 4i). PIC + RT also increased CD4⁺ T cell infiltration of tumors when compared to the control group, although the levels of CD4⁺ T cells infiltration were not significantly different between RT and PIC + RT treated tumors (Fig. 4j). Moreover, RT increased the abundance of CD25⁺FOXP3⁺ Tregs among CD4⁺ T cells, and this effect was antagonized by PIC injection (Fig. 4k and Supplementary Fig. 23). By analyzing markers of activation state (CD69), effector state (CD44) and memory state (CD62L) on tumor infiltrating T cells, we found higher abundance of activated T cells (CD69⁺CD4⁺ and CD69⁺CD8⁺ cells), effector T cells (CD44⁺CD4⁺ and CD44⁺CD8⁺ cells) and effector memory T cells (CD44⁺CD62L⁻CD4⁺ and CD44⁺CD62L⁻CD8⁺) out of CD45⁺CD3⁺ T cells among the PIC + RT treated tumors compared to other groups (Fig. 4l, m and Supplementary Fig. 24). RT decreased the ratios of infiltrating CD44⁺CD4⁺

effector T cells: Tregs and CD44+CD8+ effector T cells: Tregs (Supplementary Fig. 25). However, PIC dampened this immunosuppressive effect of RT (Supplementary Fig. 25). PD-1 is a marker of T cell exhaustion and PD-1/PD-L1 signaling plays a vital role in immune tolerance[44]. Neither RT nor PIC + RT treatment showed any significant effect on the expression of PD-1 on tumor-infiltrating CD4+ and CD8+ T cells in these B78 tumors (Fig. 4n, o). Notably, in addition to these treatment effects in the targeted tumor, we also found increased abundance of central memory T cells (CD44+CD62L+), CD44+ effector T cells and CD69+ activated T cells among CD4+ and CD8+ T cells in the TDLNs in PIC + RT group (Fig. 4p and Supplementary Figs. 26, 27)[45].

### The in situ vaccine effect of RT is potentiated by PIC

Given the favorable inflammatory and functional effects we observed from the multifunctional PIC in enhancing tumor cell sensitivity to RT, increasing the activation of an IFN-I response by RT, increasing tumor cell infiltration and antigen presentation by DCs, limiting the M2 polarization of macrophages by RT, and augmenting CD8+ T cell infiltration of tumor, we examined the potential therapeutic interaction between RT and PIC in vivo. For this, we sought to evaluate the potential cooperative effects of RT and PIC in priming a response to the ICB anti-CTLA-4 in a well-established (4 weeks post implantation), immunologically "cold" B78 melanoma model that does not respond to ICBs alone (Fig. 5a)[7]. Mice treated with PIC alone showed tumor growth that was comparable to that in control mice receiving sham treatments (Fig. 5b–d) and PIC did not improve response when added to anti-CTLA-4, as compared to anti-CTLA-4 alone. This was not surprising given the design of the PIC as a multifunctional nanoparticle to specifically enhance the in situ vaccine effect of RT. In addition, we observed that PIC did not significantly improve tumor response when combined with RT, as compared to RT alone. This was also expected as we and others have observed that in the absence of ICBs the in situ vaccine effect of RT alone is generally ineffective in activating anti-tumor immunity[7,46]. When RT and anti-CTLA-4 were combined, B78 tumor growth was reduced, but the response was still limited in this immunologically "cold" melanoma tumor model. Intriguingly, we found that PIC significantly improved the tumor response and animal survival when combined with RT + anti-CTLA-4, compared to RT + anti-CTLA-4 alone (Fig. 5b–d and Supplementary Table 6).

With the triple combination of PIC + RT + anti-CTLA-4, 69.2% (9/13) of mice were rendered tumor-free and all these mice continued to show no evidence of disease at day 90 post-treatment. At day 91 after treatment, we re-challenged a cohort of these mice with a second subcutaneous implantation of the same B78 tumor they had been cured of or an unrelated syngeneic Panc02 tumor that express different TAAs from B78 melanoma in order to assess for tumor-specific immunologic memory. Fifty days later, 80% of these mice (4/5) exhibited no B78 tumor growth and the one mouse developing a tumor showed tumor growth that was significantly slower than that observed in naïve control mice, 100% of which developed B78 tumors (Fig. 5e, f). However, the disease-free mice re-challenged with Panc02 cells all grew tumors and these exhibited similar growth to that observed Panc02 tumors in naïve control mice (Supplementary Fig. 28). From the B78 melanoma re-challenged mice, we collected splenocytes and co-cultured them with B16 melanoma cells that are parental to B78 and share common TAAs (Fig. 5g)[7,47]. Using flow cytometry on these B16-co-cultured splenocytes, we observed increased levels of early activation marker, CD69+, on both CD4+ and CD8+ T cells from mice rendered disease-free by PIC + RT + anti-CTLA-4, as compared to B16-co-cultured splenocytes from naïve control mice (Fig. 5h). Expression of granzyme B (GZMB), a cytotoxic protein that participates in T cell killing of tumor cells, was also elevated in CD8+ T cells in these splenocytes from mice rendered disease-free by PIC + RT + anti-CTLA-4 (Fig. 5i)[48]. This is consistent with the development of tumor-specific immunologic memory following PIC + RT + anti-CTLA-4 treatment and demonstrates the

achievement of an enhanced in situ vaccine effect in these mice compared to that achieved with RT + anti-CTLA-4. The PIC + RT + anti-CTLA-4 treated and re-challenged mice survived for >300 days from the first treatment without any evidence of disease recurrence, consistent with a curative treatment outcome.

To evaluate the generalizability of our observations, we tested the therapeutic efficacy of the triple combination of PIC + RT + anti-CTLA-4 in the MyC-CaP prostate tumor model in syngeneic male FVB/NTac mice and the orthotopic TC11 breast tumor model in syngeneic female FVB/NTac mice (Fig. 6a). We observed a delay in MyC-CaP tumor growth with the combination of RT + PIC (Fig. 6b–d). The combination of PIC + RT + anti-CTLA-4 in this tumor model rendered 62.5% (5/8) of mice disease-free, vs. only 37.5% (3/8) of mice treated with RT + anti-CTLA-4; this was associated with a significant improvement in tumor growth inhibition and overall mice survival (Fig. 6b–d). The estrogen-receptor-positive TC11 breast tumor is immunologically "cold" and not responsive to anti-CTLA-4 treatment alone[49]. Even when treated with combined RT and anti-CTLA-4, this TC11 tumor model was poorly responsive (Supplementary Fig. 29). However, the triple combination of PIC + RT + anti-CTLA-4 significantly suppressed tumor growth in this model and improved overall survival compared to RT + anti-CTLA-4 treatment (Fig. 6e–g).

To evaluate the necessity of forming a nanoparticle before intra-tumoral injection of the PIC components (PLL, ION and CpG) in enhancing the anti-tumor immune response to RT + anti-CTLA-4, we injected the three components (PLL, ION and CpG) separately using a technique in which needle entry was made into three different sides of a single B78 melanoma tumor to minimize the possibility of spontaneously forming PIC nanoparticle in vivo after injection. We evaluated the anti-tumor response when combined with RT and anti-CTLA-4, as compared to the combination of RT and anti-CTLA-4 with fully formed PIC (Supplementary Fig. 30a). We found the combination of the components with RT and anti-CTLA-4 conferred inferior anti-tumor efficacy, with no mice rendered disease-free at day 60 post-treatment (Supplementary Fig. 30b, c). However, 5/8 mice were rendered disease-free in the PIC + RT + anti-CTLA-4 group (Supplementary Fig. 30b,c). These results indicated that the complexation of the components (PLL, CpG and ION) to form PIC nanoparticle before injection was critical for its multifunctional design.

### PIC + RT + anti-CTLA-4 activates an effective systemic anti-tumor immune response

In settings of metastatic disease or circulating tumor cells, it is essential that any in situ vaccine strategy not only activate an effective anti-tumor immune response at a targeted tumor site but also at distant tumor sites elsewhere in the body that are not directly treated with the in situ vaccine regimen[50,51]. To assess whether local injection of PIC into an RT-treated tumor could improve the systemic anti-tumor immune response when given in conjunction with anti-CTLA-4, we generated mice bearing two B78 melanoma tumors, one on the right flank and the other one implanted 7 days later on the left flank (to simulate a smaller distant site of metastasis). RT and intratumoral injection of PIC were delivered to the larger tumor on the right flank only and the growth of both tumors was monitored (Fig. 7a). For the directly treated right flank tumors, tumor growth was again significantly decreased in the RT + anti-CTLA-4 and the PIC + RT + anti-CTLA-4 groups with the latter exhibiting significantly reduced tumor growth compared to RT + anti-CTLA-4 (Fig. 7b, d). Importantly, these same treatment effects were observed at the left flank tumors, which were not directly treated with RT or injected with PIC. We observed that PIC + RT + anti-CTLA-4 treatment significantly reduced the progression of these left flank tumors compared to all other treatment groups and the triple treatment combination (PIC + RT + anti-CTLA-4) rendered 4/9 mice completely disease-free; no mice in any other group became tumor-free (Fig. 7b–d). Moreover, the PIC + RT + anti-CTLA-4

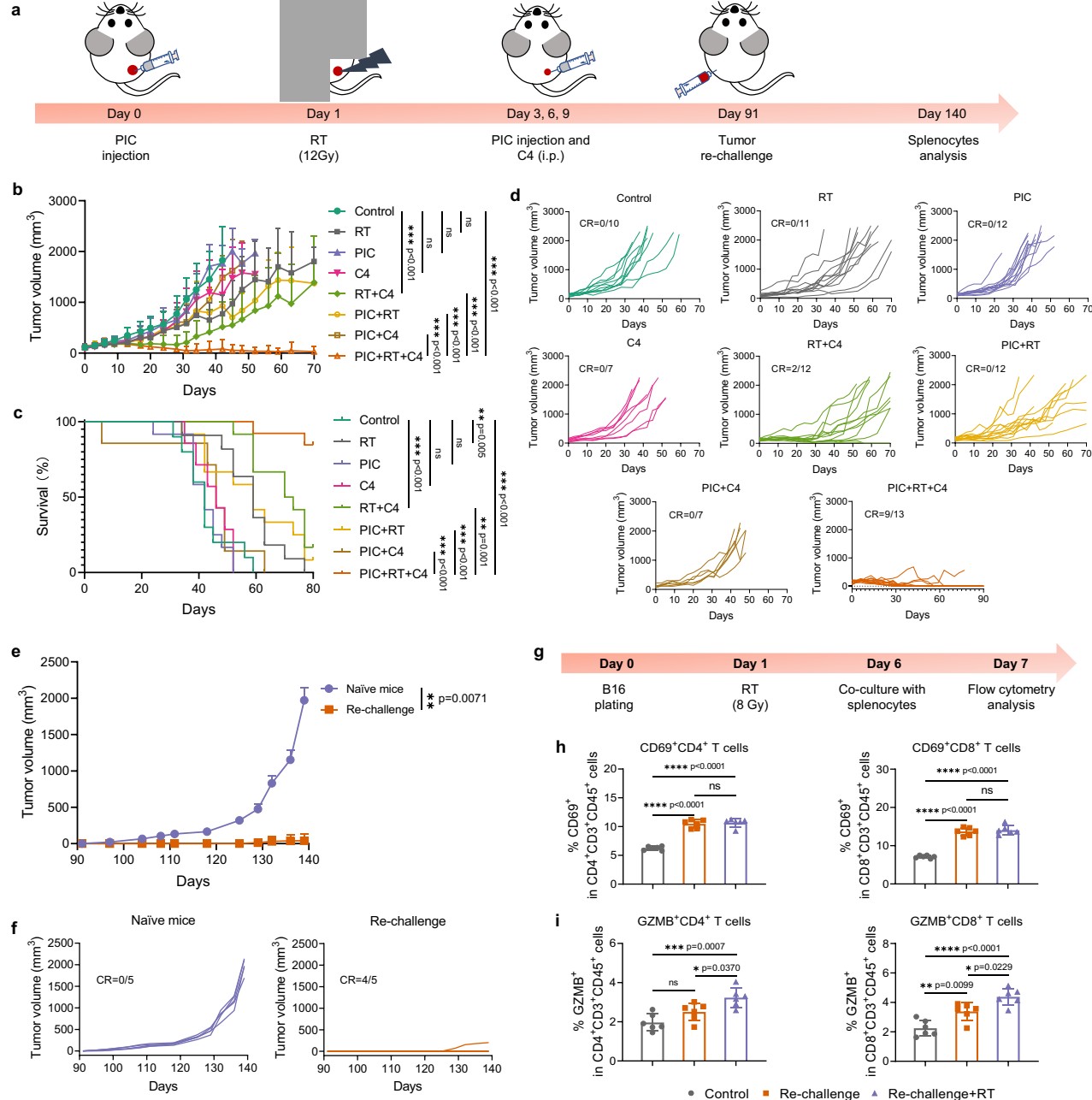

**Fig. 5 | The triple combination of PIC + RT + anti-CTLA-4 (C4) improves tumor response and mouse survival leading to anti-tumor memory in mice bearing a B78 melanoma. a** Scheme for the studies. **b** Average tumor growth curves and **c** survival rate of mice after the treatments indicated. (Control: *n* = 10; RT: *n* = 11; C4 and PIC + C4: *n* = 7; PIC + RT + C4: *n* = 13; others: *n* = 12 biologically independent animals). **d** Individual tumor growth curves in **b**. **e** Average tumor growth curves after the naïve and tumor-free mice were re-challenged with B78 melanoma cells. (*n* = 5 biologically independent animals). **f** Individual tumor growth curves in **e**. **g** Scheme for the co-culture of splenocytes extracted from tumor re-challenged mice with B16 melanoma cells. Quantification of **h** CD69+ and **i** GZMB+ in CD4+ and

CD8+ T cells in splenocytes by flow cytometry. (*n* = 6 biologically independent samples). Control: non-radiated B16 cells co-cultured with splenocytes from naïve mice; Re-challenge: non-radiated B16 cells co-cultured with splenocytes from re-challenged mice; Re-challenge+RT: radiated B16 cells co-cultured with splenocytes from re-challenged mice. CR: complete response. C4: anti-CTLA-4. i.p.: intraperitoneal injection. PIC: 140 µg/100 µL/dose. C4: 100 µg/100 µL/dose. Data in **b**, **e**, **h** and **i** are shown as mean ± SD. Statistical significance was calculated via linear mixed effects modeling in **b**, log-rank test in **c**, time-weighted average in **e**, and one-way ANOVA test in **h**, **i**. *$p < 0.05$, **$p < 0.01$, ***$p < 0.001$ and ****$p < 0.0001$. Source data are provided in Source Data file.

treatment group showed longer mouse survival compared to the RT + anti-CTLA-4 group (Fig. 7c).

Finally, we evaluated the toxicity of PIC + RT and PIC + RT + anti-CTLA-4 in vivo (Supplementary Fig. 31a). We did not observe changes in the body weight of mice during treatment (Supplementary Fig. 31b). By analyzing complete blood counts, we found the percentage of lymphocytes was slightly reduced in the blood of PIC + RT and PIC + RT + anti-CTLA-4 treated mice at 1 week and 2 weeks after RT. At

3 weeks after radiation, the percentage of lymphocytes returned to normal levels (Supplementary Fig. 32). This transient lymphopenia may have resulted from the radiosensitivity of lymphocytes or from extravasation of these cells in the setting of an activated immune response. Reductions of lymphocytes counts are commonly observed after large field radiation therapy in preclinical and clinical studies[52,53]. The levels of monocytes, red blood cells and platelets were not significantly influenced during or after these treatments (Supplementary

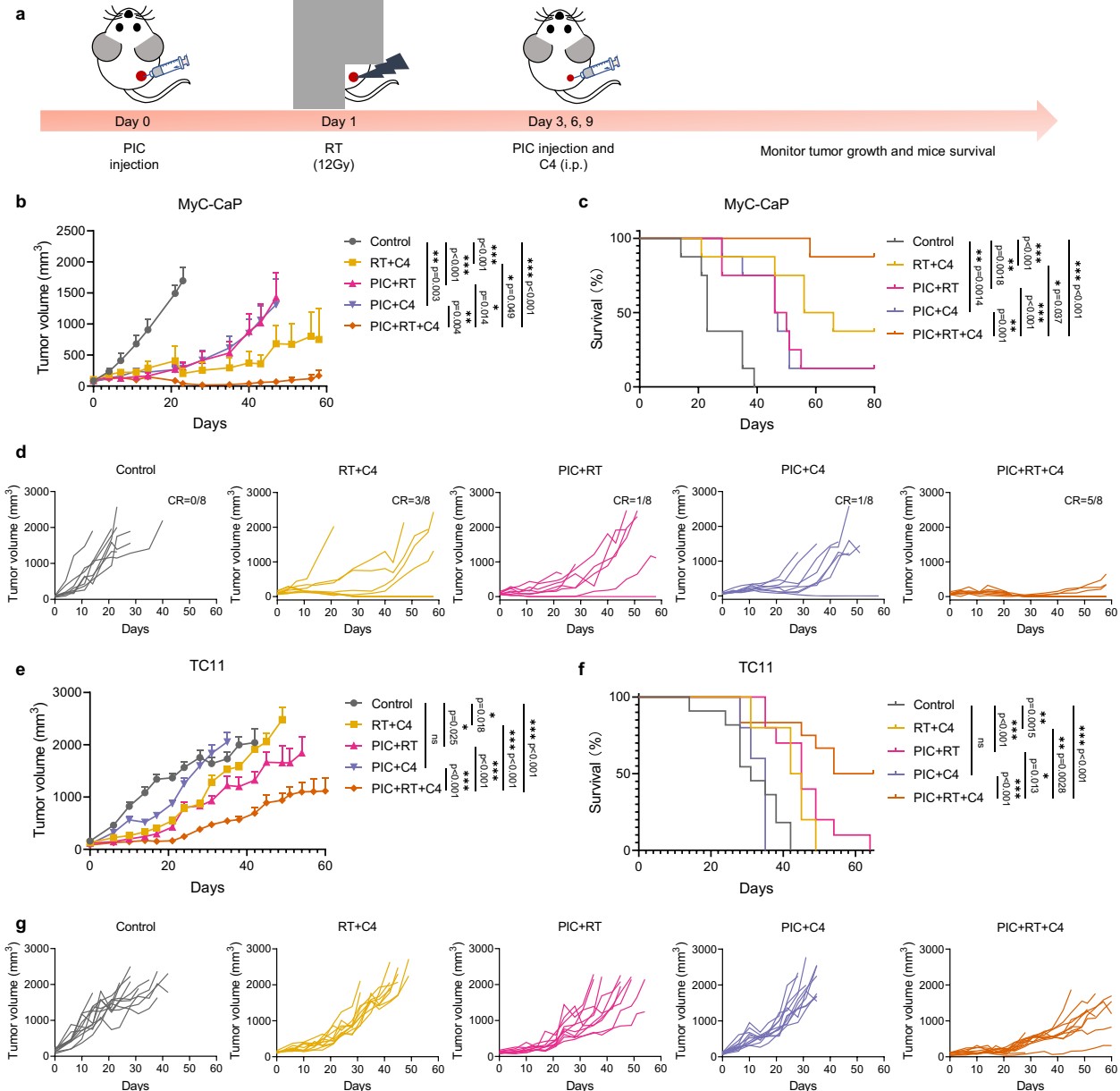

**Fig. 6 | The combination of PIC + RT + anti-CTLA-4 (C4) improves tumor response and survival in both the MyC-CaP prostate and TC11 breast tumor models. a** Scheme for the studies. **b** Average tumor growth curves of MyC-CaP prostate tumors and **c** survival rate of mice after the indicated treatments. (*n* = 8 biologically independent animals). **d** Individual tumor growth curves in **b**. **e** Average tumor growth curves of TC11 tumor and **f** survival rate of mice after the indicated treatments (control: *n* = 11; PIC + RT + C4: *n* = 12; others: *n* = 10 biologically independent animals). **g** Individual tumor growth curves in **e**. CR: complete response. C4: anti-CTLA-4. i.p.: intraperitoneal injection. For MyC-CaP prostate cancer model, tumor was implanted on the right flank of mice. For TC11 breast cancer model, tumor was implanted on the right mammary fat pad of mice. PIC: 140 μg/100μL/dose. C4: 100 μg/100 μL/dose. Data in **b** and **e** are shown as mean ± SD. Statistical significance was calculated via linear mixed effects modeling in **b** and **e**, and log-rank test in **c** and **f**. \**p* < 0.05, \*\**p* < 0.01, \*\*\**p* < 0.001 and \*\*\*\**p* < 0.0001. Source data are provided in Source Data file.

Fig. 32). In addition, no significant variations were found in basic metabolic panels during PIC + RT or PIC + RT + anti-CTLA-4 treatment (Supplementary Fig. 33). Analysis of normal tissue histology did not reveal any apparent effect of these treatments on liver, kidney, spleen, intestine or bone (Supplementary Fig. 34).

## Discussion

Response to cancer immunotherapies, such as ICB, is limited in immunologically "cold" tumors, which are characterized by few tumor-infiltrating effector T cells, low tumor neoantigen load, and activation of immune suppressive mechanisms in the tumor microenvironment[3]. RT offers a readily translatable approach to overcome many of these

characteristics, which otherwise antagonize anti-tumor immunity. Multiple clinical studies have demonstrated safety for combining RT with ICB and in many cases these combinations have resulted in improved survival and increased rates of response to ICBs, particularly among those patients with immunologically "cold" tumors[17,54,55]. By converting a patient's own tumor into a nidus for presentation of tumor-specific antigens in a way that will stimulate and diversify an anti-tumor T cell response, RT can activate an in situ anti-tumor vaccine effect, thereby priming an adaptive immune response in "cold" tumors that have not otherwise been effectively recognized by the patient's immune system[7,13]. However, RT also elicits potentially detrimental local effects on the radiated tumor-immune microenvironment

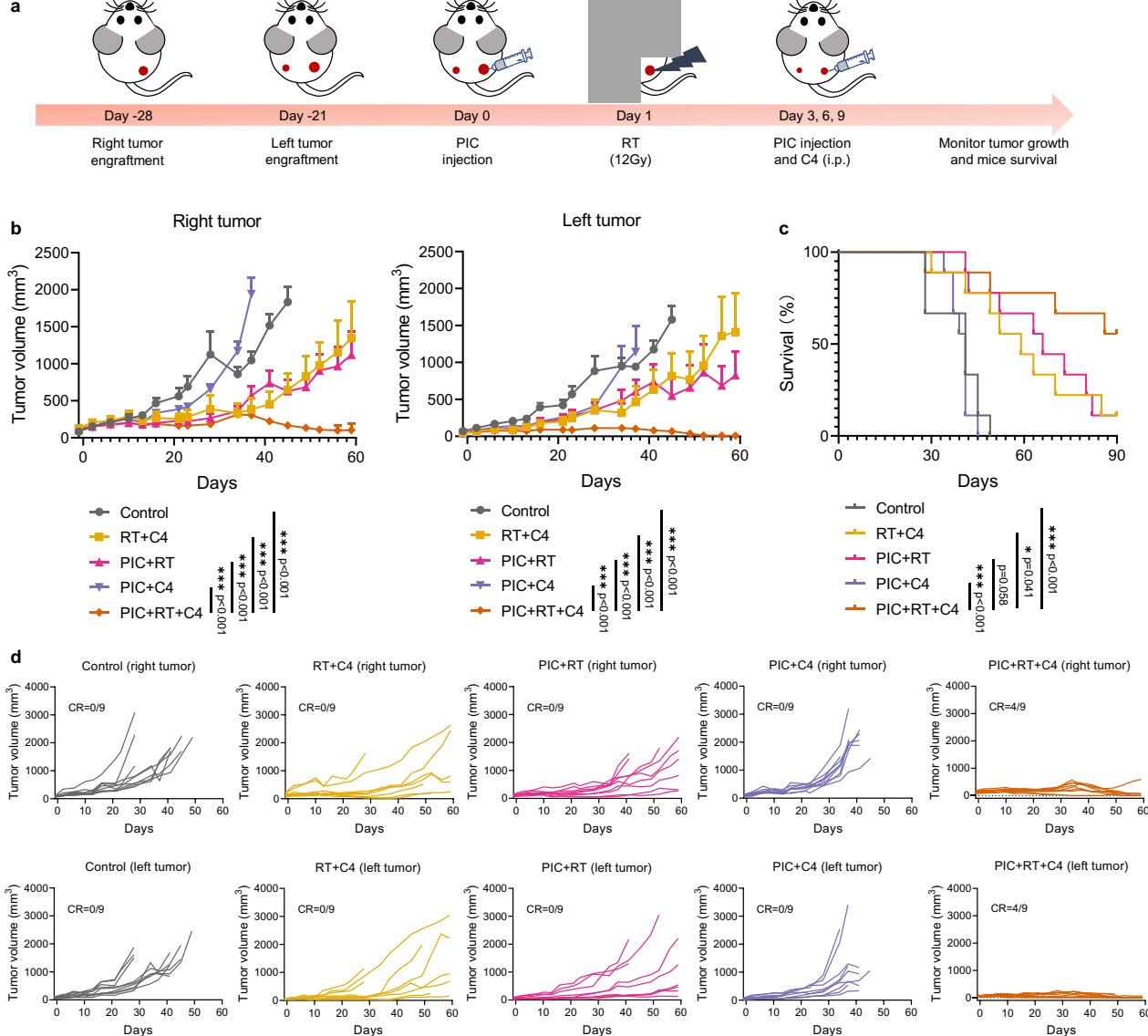

**Fig. 7 | The systemic anti-tumor immune response induced by PIC + RT + anti-CTLA-4 (C4). a** Scheme for the study of the abscopal effect of PIC + RT + anti-CTLA-4. **b** Average tumor growth curves of both right and left tumor, and **c** survival rate of mice after the indicated treatment regimen. ($n = 9$ biologically independent animals). **d** Individual tumor growth curves in **b**. CR: complete response. C4: anti-CTLA-4. i.p.: intraperitoneal injection. PIC: 140 µg/100µL/dose. C4: 100 µg/100 µL/ dose. Data in **b** are shown as mean ± SD. The statistical significance was calculated via linear mixed effects modeling in **b**, and log-rank test in **c**. *$p < 0.05$, **$p < 0.01$, ***$p < 0.001$ and ****$p < 0.0001$. We compared only the PIC + RT + C4 group with others to specifically test the hypothesis that this combination treatment would achieve greater tumor response and survival compared to doublet combination or control groups. Source data are provided in Source Data file.

including the recruitment, polarization, and activation of suppressive M2 macrophages that may critically limit the priming and propagation of an adaptive anti-tumor T cell response[15,18]. To potentiate the in situ vaccination effect of RT, we developed a multifunctional nanoparticle (PIC) by a facile complexation method for the specific purpose of immuno-modulating the radiated tumor microenvironment—preventing some of the detrimental immunologic effects of RT and augmenting some of its favorable effects (Fig. 8).

The positively charged PIC with a hydrodynamic diameter about 110 nm showed internalization in B78 murine melanoma cells. Clonogenic assays and immunofluorescence analysis demonstrated that PIC potentiated the sensitivity of B78 cells to RT (Fig. 1). Prior studies suggest that the radiosensitizing effect of IONs can be attributed to the catalytic effects of the released iron ions and the active surfaces of IONs under RT, leading to the generation of ROS in cancer cells[26,56]. While PIC will also likely enhance the radiosensitivity of tumor

infiltrating immune cells, any transient loss of immune cells will be repleted by circulating immune cells drawn into the tumor by the enhanced inflammatory effects of RT + PIC. Notably, PIC accentuated the capacity of RT to activate an IFN-I response in B78 melanoma cells (Fig. 2). In addition, PIC directly activated an IFN-I response in pDCs, likely via the stimulation of TLR-9 (Figs. 2, 3). PIC may also contribute to additional mechanisms that lead to the activation of IFN-I response, such as capturing the cell-free DNA released from RT-treated cells and facilitating its internalization in immune cells to stimulate an IFN-I response[57]. This will be an important topic for further investigation on the interactions between PIC with RT in future studies. Activation of an IFN-I response in tumor and antigen-presenting cells plays an important role in the recruitment and activation of immune cells and has been demonstrated to play a critical role in the capacity of RT to augment response to ICIs[34,58–60]. In addition to activating DCs, PIC captured tumor antigens, improved internalization of these in APCs,

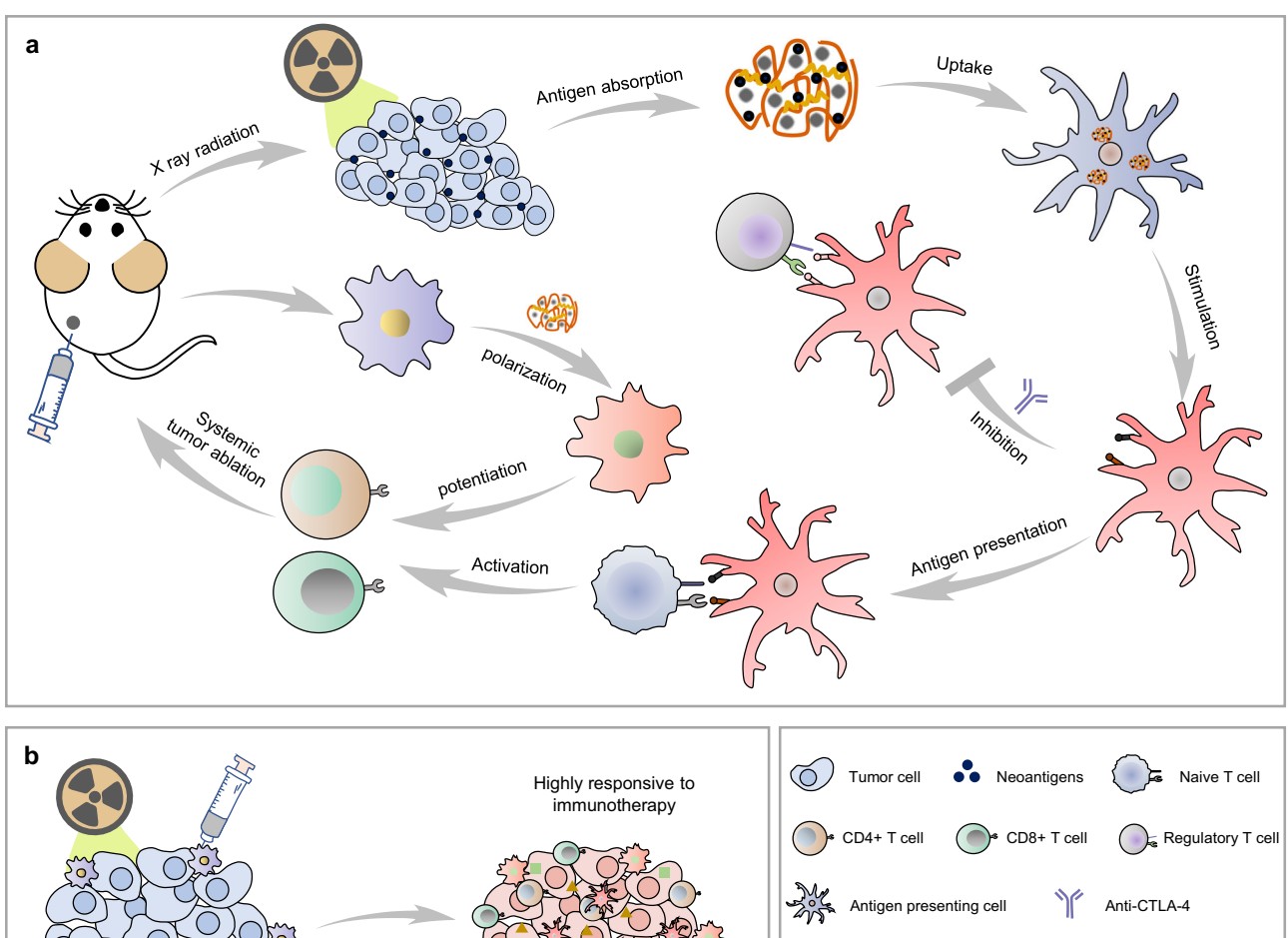

**Fig. 8 | Schematic illustration of the mechanisms whereby the multifunctional PIC may potentiate the in situ vaccination effect of RT. a** The proposed mechanisms of action for PIC in potentiating the in situ vaccination effect of RT. PIC can sensitize tumor cells to RT, capture TAAs released from dying tumor cells, promote the uptake of these captured TAAs in APCs, activate and mature APCs, promote a favorable ratio of M1:M2 TAMs, and increase tumor infiltration by APCs and effector T cells in part by more effectively activating an IFN-I response among radiated cells. **b** As a result of these mechanisms, PIC combined with RT can help convert a radiated immunologically "cold" tumor to an immunologically "hot" microenvironment.

and enhanced the capacity for generating antigen-specific adaptive T cell immunity (Figs. 1 and 2).

Increasing the ratio of M1: M2 polarized macrophages in a tumor may enable more effective development of anti-tumor immune responses[23,61]. While RT is associated with a potentially detrimental increase in proportion of TAMs that are M2-polarized, in radiated BMDMs we observed that PIC antagonized this effect of RT and thereby increased the ratio of M1: M2 macrophages (Fig. 2)[18,62]. Consistent with this, combination of RT and PIC in vivo resulted in elevated bulk tumor mRNA levels of M1-associated *Nos2* expression relative to M2-associated *Arg1* expression at 15 days after radiation and flow cytometry confirmed increased ratio of TAMs expressing M1 markers versus M2 markers with PIC + RT compared to RT alone (Figs. 3, 4). The combination of RT and PIC also increased the level of effector CD8⁺ T cell infiltration and decreased the level of immunosuppressive regulatory T cells in tumors relative to RT alone (Fig. 4). These results demonstrate the capacity of the PIC to favorably immunomodulate the radiated tumor microenvironment.

When combined with anti-CTLA-4, the PIC + RT in situ vaccination enabled greater tumor response and improved survival as well as tumor-specific immune memory and robust systemic anti-tumor immunity at tumors not directly treated by RT or PIC injection (Figs. 5 and 7). The effects of RT + PIC were broadly recapitulated in two additional difficult to treat, immunologically "cold" tumor models of prostate and breast cancer using a distinct mouse strain (Fig. 6). These results demonstrate that the potentiation effect of PIC on the in situ vaccination of RT can be applied in diverse tumor settings and can prime a potent systemic anti-tumor immune response when combined with anti-CTLA-4 for the treatment of metastatic disease. Intratumoral injection of PIC minimizes its potential for triggering systemic toxicity. Consistent with this, in mice treated with PIC + RT or PIC + RT + anti-CTLA-4 we observed no evidence of hepatic, renal, gastrointestinal, or autoimmune toxicities and modest hematological effects that did not appear to result in symptoms or adverse effects (Supplementary Figs. 32, 33).

This study has several limitations. Among these is the exclusive use of syngeneic murine tumor models for in vivo studies. This reflects a necessity to evaluate and test the effects of PIC on the immunologic response to radiotherapy in the setting of a tumor and an intact native-endogenous immune system. The results obtained now support

initiation of clinical testing to validate these observations in the setting of human tumors and immune cells. Here, we have specifically focused on the effect of PIC in modulating the local immune response to a limited range of RT dose (12 Gy) in a single fraction. This dose has been observed to be optimal with respect to activation of the IFN-I response[10] and can be employed in clinical studies testing the potential interaction of RT and ICB[14,16,17]. In future studies it will be valuable to further evaluate the effects of PIC across a broader range of radiotherapy doses and in the setting of fractionated regimens[11,63]. Notably, although PIC requires intratumoral injection and this could potentially limit clinical application in some patients, such a delivery route is increasingly employed in clinical trials and is standardly used clinically for delivery of oncolytic viral therapies[64]. In this context, it may be valuable to minimize the number of required intratumoral injections. In follow-up studies, the PIC + RT therapeutic regimen will be further optimized with the goal of further improving the anti-tumor immunity and clinical translatability. Specifically, we will be investigating the importance of each injection time point, the possible need for additional injections, and possible approaches to limit the number of injections while maximizing efficacy including the use of hydrogels and other novel approaches for formulation in conjunction with immune adjuvants.

Despite these limitations, this study provides compelling evidence to demonstrate that a multifunctional, purposefully-designed nanoparticle with readily reproducible and scalable production methods can favorably modulate the tumor-immune microenvironment following RT in a manner that favors activation of an in situ vaccine effect. Because of the growing importance of immunotherapies and ICBs in particular, the durability of tumor response to these agents among responding patients, and the limited rates of response to these therapies for most cancer patients, the potential clinical impact of a treatment approach that augments response rates to ICBs is substantial. In this case, that magnitude of possible benefit is further enhanced by the fact that RT and PIC are not limited in application to any particular tumor type but instead have potential to benefit patients with any type of solid tumor. The promising results reported here justify further preclinical and early phase clinical testing of PIC in combination with RT and ICB for treatment of metastatic cancers.

A multifunctional nanoparticle (PIC) with readily scalable and reproducible production methods was engineered to potentiate the in situ vaccination effect of RT. This study illuminates an immediately translatable approach to augment response to ICB in settings of immunologically "cold" metastatic disease.

## Methods

All animal studies in this research were approved by the Institutional Animal Care and Use Committee at the University of Wisconsin – Madison (protocol: M005670).

### Materials

Poly-L-lysine hydrobromide (PLL) with a molecular weight of 30 to 70 kDa, ammonium hydroxide ($NH_4OH$) solution and sucrose were purchased from Sigma-Aldrich. Ferric chloride hexahydrate ($FeCl_3 \cdot 6H_2O$) and ferrous sulfate heptahydrate ($FeSO_4 \cdot 7H_2O$) were purchased from Fisher Scientific. Citric acid was obtained from Acros Organics. CpG oligodeoxynucleotides 1826 (CpG 1826) was purchased from Integrated DNA Technologies. Fluorescein isothiocyanate (FITC) was purchased from Chemodex Ltd. 3-(4,5-dimethylthiazol-2-yl)-2,5-diphenyltetrazolium bromide (MTT) was purchased from VWR International, LLC. Cyanine5-NHS easter (Cy5-NHS) was purchased from Lumiprobe. Ovalbumin (Ova) and fluorescein isothiocyanate labeled ovalbumin (FITC-Ova) were purchased from ThermoFisher. Macrophage colony-stimulating factor (M-CSF), granulocyte-macrophage colony-stimulating factor (GM-CSF) and Interleukin 4 (IL-4) were

purchased from Biolegend. The information of Taqman probes used for RT-qPCR and the information of antibodies used for flow cytometry and immunofluorescence staining are listed in Supplementary Table 4 and Supplementary Table 5 in the Supplementary Information, respectively. α-CTLA-4 (IgG2c, clone 9D9) was produced and purified by Neoclone.

### Synthesis of iron oxide nanoparticle (ION)

ION was synthesized according to a previously reported method[31]. Briefly, $FeCl_3 \cdot 6H_2O$ (3.30 g, 12.2 mmol) and $FeSO_4 \cdot 7H_2O$ (2.37 g, 8.5 mmol) were dissolved in 20 mL deionized water. The solution was stirred under nitrogen atmosphere for 30 min. After the solution was heated to 80 °C, $NH_4OH$ solution (11.25 mL, 28–30%) was added dropwise. The solution was then heated to 90 °C. A citric acid solution (4.00 mL, 475 mg/mL) was subsequently added to this solution and the mixture was stirred at 90 °C for 60 min. Finally, ION was obtained after dialysis against deionized water using a dialysis tubing (MWCO: 8000 Da).

### Preparation of PLL/CpG/iron oxide (PIC)

ION was mixed with PLL in deionized water at different weight ratios through vortexing (for 20 s). The mixture was incubated at room temperature for 20 min. Then, CpG was added and the resulting solution was vortexed for 20 s and incubated at room temperature for another 20 min to yield PIC. PIC was stored at 4 °C in deionized water or in 1 mM PBS, and the particle size was monitored for 30 days. In addition, PIC was lyophilized in the presence of 1% sucrose (wt/wt), and the dry powder was stored at −20 °C for 12 weeks. The particle size and zeta potential were monitored during the storage. Gel electrophoresis was used to confirm the complexation of CpG in the PIC nanoparticle. The weight ratios of PLL/CpG and ION/PLL/CpG were 3.5/1 and 4.8/3.5/1, respectively. The amount of CpG for each sample was 0.75 μg. Electrophoresis was performed using 4% agarose gel and TBE (Tris-Borate-EDTA) buffer with a voltage of 100 V for 15 min.

### FITC-labeled and Cy5-labeled PIC

FITC-labeled PIC was prepared through the complexation of FITC-conjugated PLL with ION and CpG, while Cy5-labeled PIC was prepared through the complexation of Cy5-conjugated PLL with ION and CpG. To synthesize FITC-labeled PLL, a FITC solution (3 mL, 0.1 mg/mL in DMSO) was added in PLL aqueous solution (20 mL, with 10 mg PLL), and the mixture was stirred at room temperature in the dark for 24 h. To synthesize Cy5-labeled PLL, a Cy5-NHS solution (20 μL, 5 mg/mL in DMSO) was added into a PLL aqueous solution (10 mL, with 10 mg PLL and 60 mM $NaHCO_3$), and the mixture was stirred at room temperature in the dark for 24 h. The FITC-conjugated PLL and Cy5-conjugated PLL were obtained by dialysis against deionized water using a dialysis tubing (MWCO: 3500) followed by lyophilization. FITC-labeled PIC or Cy5-labeled PIC were prepared using the dye-labeled PLL through the same method mentioned above.

### Characterization

The hydrodynamic diameter and zeta potential of ION and PIC were characterized by a dynamic light scattering (DLS) spectrometer (Malvern Zetasizer Nano ZS) with sample concentration at 0.1 mg/mL. The morphologies of ION and PIC were characterized by transmission electron microscopy (TEM, Philips CM200 Ultra Twin). The Fe content in ION was measured by inductively coupled plasma-optical emission spectrometry (ICP-OES, Agilent 5110).

### Cell culture

B78 (B78-D14, GD2+) melanoma originated from B16 melanoma and was obtained from Ralph Reisfeld (Scripps Research Institute) in 2002[65]. B16 melanoma cells were obtained from Memorial Sloan Kettering Cancer Center. MyC-CaP and RAW264.7 cells were purchased

from ATCC (MyC-CaP: CRL-3255; RAW264.7: TIB-71). Panc02 pancreatic cancer cells were obtained from the National Cancer Institute. B16 cells were transduced to express SIINFEKL via lentiviral transduction pLV[Exp]-Hygro-CBh>SIINFEKL (VectorBuilder; VB210327-1014dyd), which is a lentiviral plasmid that we designed using VectorBuilder's platform. Positively transduced cells were referred to as B16-SIINFEKL (a kind gift from Dr. Amy Erbe), and were selected for using hygromycin (50 ug/ml). Stably transduced cells were single-cell cloned. Clones were selected for downstream use following IFNγ (100 U/mL; cat #505702, Biolegend) stimulation, and screened for MHC-I presentation of SIINFEKL via flow cytometry on an Attune NxT Flow Cytometer (Thermofisher) using anti-mouse H-2Kb bound SIINFEKL-APC (clone 25-D1.16, cat # 141605, Biolegend). TC11 cells were generated from an ER + mammary tumor that developed in an NRL-PRL female[49]. B78, B16, B16-SIINFEKL, MyC-CaP, Panc02 and RAW264.7 cells were cultured in Roswell Park Memorial Institute (RPMI)−1640 medium. TC11 cells were cultured in Dulbecco's Modified Eagle Medium (DMEM). Primary macrophages were derived from bone marrow (BMDM) and cultured in Modified Eagle Medium (MEM) supplemented with 12 ng/mL M-CSF. Dendritic cells (DCs) were enriched from the splenocytes that were collected from Flt3L injected C57BL/6 mice using an EasySep™ Mouse Pan-DC Enrichment Kit (STEMCELL), and cultured in RPMI-1640 medium supplemented with 25 ng/mL GM-CSF and 20 ng/mL IL-4. RPMI-1640 medium, MEM and DMEM were supplemented with 10% (vol/vol) fetal bovine serum (FBS), 100 U/mL penicillin and 100 μg/mL streptomycin. Cell authentication was performed per ATCC guidelines using morphology, growth curves and Mycoplasma testing within 6 months of use and routinely thereafter. All the cells were cultured at 37 °C in a humidified atmosphere containing 5% $CO_2$.

### Cell lysate absorption of PIC in vitro

B78 melanoma cell lysates were prepared by ultrasonication. The concentration of proteins in the cell lysate was determined by BCA assay (ThermoFisher). To determine cell lysate absorption, PIC (0.1 mL, 0.7 mg/mL) was added to the B78 cell lysates (0.4 mL). B78 cell lysates (0.4 mL) dispersed in deionized water (0.1 mL) was used as a control. The mixtures were incubated at 37 °C for 4 h followed by centrifugation at 12000 × g for 20 min. The concentrations of proteins in the supernatants were determined by BCA assay.

### Cytotoxicity assay

B78 cells and RAW264.7 cells were seeded and cultured in 96-well plates with 8000 cells per well 24 h before treatments. Cells were treated with PIC at different concentrations (i.e., 0, 1.1, 2.2, 4.4, 8.8, 17.5 and 35 μg/mL). After incubation for 48 h, cells were washed with PBS three times and subsequently treated with fresh medium containing 0.5 mg/mL MTT. After incubation for another 4 h, the medium was aspirated and 150 μL DMSO was added to each well to dissolve the formazan product. The absorbance at 560 nm was then measured using a GloMax-Multi Microplate Multimode Reader (Promega). Cell viability was calculated as a percentage of the absorbance relative to that of untreated cells.

### In vitro cellular uptake

For flow cytometry studies, B78 cells and RAW264.7 cells were seeded and cultured in 96-well plates with 10,000 cells per well 24 h before treatments. Cells were treated with FITC-labeled PIC at different concentrations (i.e., 1.9, 3.8 and 7.5 μg/mL). Two hours later, the cells were collected by trypsinization and centrifugation. The cellular uptake of FITC-labeled PIC was measured by flow cytometry (Attune NxT flow cytometer system, ThermoFisher) quantifying FITC-positive cells.

For confocal laser scanning microscopy, B78 cells and RAW264.7 cells were seeded on coverslips in 6-well plates and incubated for 24 h. Cells were treated with FITC-labeled PIC at 3.8 μg/mL. After 2 h of

incubation, cells were washed with PBS and fixed with 4% paraformaldehyde. Thereafter, cells were stained by 4'−6-diamidino-2-phenylindole (DAPI, 1 mg/mL, 1 μL/well) and washed with PBS. The coverslips were carefully taken out from the wells, placed on slides and enclosed with anti-fade mounting medium. The samples were imaged with a Nikon A1R-Si high speed spectral laser scanning confocal inverted microscope (Nikon, Melville).

### In vitro immunofluorescence of damaged DNA

B78 cells were seeded on coverslips in 12-well plates with $1 \times 10^5$ cells per well and incubated at 37 °C for 24 h. Cells were treated with PIC at 4.67 μg/mL. Four hours later, radiation (12 Gy, RS225 Cell Irradiator (Xstrahl)) was delivered to the cells. After another 1 h of incubation, the cells were washed with PBS and fixed with 4% paraformaldehyde solution. After the cells were permeabilized with 0.1% Triton X-100, non-specific binding was blocked with SuperBlock (TBS T20, ThermoFisher). Then primary antibody (Phospho-Histone H2A.X (Ser139) (20E3) Rabbit mAb, #9718, Cell Signaling Technology, 1:400) was applied and incubated overnight at 4 °C. After washing, a fluorescently labeled secondary antibody (Catalog #A-11008, ThermoFisher) was applied and incubated for 1 h at room temperature, the cell nuclei were stained with DAPI. Then the coverslips were washed with PBS and DI water, and then carefully taken out from the wells, placed on slides and enclosed with anti-fade mounting medium. The samples were imaged with a Nikon A1R-Si high speed spectral laser scanning confocal inverted microscope (Nikon, Melville).

### cGAS-STING activation

B78 cells were seeded in 6-cm cell culture plates with $1 \times 10^6$ (for the collection on day 1), $0.5 \times 10^6$ (for the collection on day 4) or $0.25 \times 10^6$ (for the collection on day 7) cells per well. After incubation overnight, the following treatments were performed on the cells: (1) un-treated; (2) radiating the cells at a dose of 12 Gy and changing to fresh media 1 h after the radiation; (3) adding PIC at a concentration of 4.67 μg/mL; (4) adding PIC at a concentration of 4.67 μg/mL, radiating the cells at a dose of 12 Gy 4 h later, and changing to fresh media after another 1 h; and (5) radiating the cells at a dose of 12 Gy and changing to fresh media containing 4.67 μg/mL PIC at 1 h later. At day 1, 4 and 7, the cells were washed with cold PBS three times followed by direct addition of Trizol (1 mL). RNA was extracted using RNeasy Mini Kit (QIAGEN, Cat # 74106) according to the manufacturer's instructions. cDNA was synthesized using QuantiTect Reverse Transcription Kit (QIAGEN, Cat # 205314) according to the manufacturer's standard protocol. Quantitative polymerase chain reaction (RT-qPCR) was performed using Taqman Fast Advanced Master Mix and predesigned Taqman gene expression assays for *Ifnβ1* and *Pd-l1* (the IDs of the Taqman assays have been provided in Supplementary Table 4). Thermal cycling conditions (QuantStudio 6, Applied Biosystems) included the UNG incubation stage at 50 °C for 2 min, followed by AmpliTaq™ Fast DNA polymerase activation stage at 95 °C for 2 min followed by 40 cycles of each PCR step (denaturation) 95 °C for 1 s and (annealing/extension) 60 °C for 20 s. For data analyses, Ct values were transferred to an Excel file and fold change was determined using the ΔΔCt method. HPRT was used as the endogenous control.

### Polarization of macrophages

BMDMs were seeded in 6-cm cell culture plates with $0.5 \times 10^6$ cells per well. After incubation overnight, fresh media containing 6 ng/mL M-CSF was changed, and the following treatments were performed on these cells: (1) un-treated; (2) radiating the cells at a dose of 12 Gy and changing fresh media 1 h after the radiation; (3) adding PIC at a concentration of 4.67 μg/mL; (4) radiating the cells at a dose of 12 Gy and changing fresh media containing 4.67 μg/mL PIC at 1 h later. When the culture media was changed, 6 ng/mL M-CSF was supplemented. At day 1 and 4, the cells were collected and stained with antibodies (or dyes):

anti-F4/80 PE-Dazzle 594, anti-CD206 BV421, anti-CD11b BV711, anti-CD80 APC, Live/Dead Ghost Red 780. After the cells were fixed and permeabilized, anti-CD163 PE-Cy7 was added to the cells for intracellular staining. The UltraComp Beads eBeads (Invitrogen) were used for compensation. All samples were incubated with CD16/CD32 (Fc block) for 5 min at room temperature before staining. Flow cytometry was performed on an Attune Cytometer (ThermoFisher).

### TLR-9 activation and antigen uptake by DCs

DCs were seeded in 12-well plates with $2 \times 10^5$ cells/well and incubated for 24 h. The following treatments were performed on these cells: (1) un-treated; (2) adding CpG at a concentration of 0.5 µg/mL; (3) adding PIC at a concentration of 4.67 µg/mL. Twenty four hours later, the supernatants were collected for the quantitative analysis of IFN-β using a mouse IFN-β Elisa kit (Biolegend, Cat # 439407), and the cells were collected and stained with antibodies (or dyes): anti-CD11c PerCP-Cy5.5, anti-CD80 PE, anti-CD86 BV605, anti-CD317 Alexa 700, Live/Dead Ghost Red 780. To study the cellular uptake of FITC-Ova, the following treatments were performed on the DCs: (1) un-treated; (2) adding FITC-Ova at a concentration of 1.67 µg/mL. (3) adding the mixture of FITC-Ova and PIC (1.67 µg/mL FITC-Ova and 4.67 µg/mL PIC). The FITC-Ova and PIC were mixed 20 min before adding to the cells. Twenty four hours later, the cells were collected and stained with anti-CD11c PerCP-Cy5.5 and Live/Dead Ghost Red 780. The UltraComp Beads eBeads (Invitrogen) were used for compensation for flow cytometry. All samples were incubated with CD16/CD32 (Fc block) for 5 min at room temperature before staining. Flow cytometry was performed on an Attune Cytometer (ThermoFisher).

### Analysis of T cells

C57BL/6 female mice (7–8 weeks) were randomized into three groups and subcutaneously injected with 100 µL of (1) the mixture of Ova and PIC (1.4 mg/mL of PIC and 0.5 mg/mL of Ova. PIC and Ova were mixed 20 min before injection); (2) Ova solution (0.5 mg/mL of Ova); (3) the mixture of Ova with CpG and ION (0.5 mg/mL of Ova, 0.15 mg/mL of CpG and 0.72 mg/mL of ION. Ova, CpG and ION were mixed 20 min before injection). At day 13 after the injection, B16-SIINFEKL cells were seeded in 6-well plates with $0.5 \times 10^6$ cells per well. At day 14, the mice were euthanized and the spleens were collected aseptically, dissociated into a single-cell suspension, incubated in RBC lysis buffer for 10 mins and then PBS was added to neutralize the lysis buffer. $0.5 \times 10^6$ splenocytes were added to the B16-SIINFEKL culture and these cells were incubated overnight. The next day, 1 µL/sample of BD Cytofix/Cytoperm Plus kit was added to the cells for 4–6 h. The cells were collected and stained with antibodies (or dyes): anti-CD4 FITC, anti-CD69 PE-Cy5, anti-CD45 PE-Cy7, anti-CD3 BV605, anti-CD44 BV711, anti-CD8a Alexa 700 and Ghost Red 780. After the cells were fixed and permeabilized, anti-IFN-γ PE-Dazzle 594 was added to the cells for intracellular staining. UltraComp Beads eBeads (Invitrogen) were used for compensation. All samples were incubated with CD16/CD32 (Fc block) for 5 min at room temperature before staining. Flow cytometry was performed on an Attune Cytometer (ThermoFisher).

### Clonogenic assay in vitro

The clonogenic assay was performed as previously described[7]. 1000 B78 cells were plated into the 6-cm cell culture plates. One day later, PIC or vehicle control solution was added with a final concentration of 4.67 µg/mL. 4 h later, radiation treatment was delivered at doses of 0 Gy, 3 Gy, 6 Gy or 9 Gy to the cells. Fresh culture media was changed 1 h after radiation. 7 days later, when the control plates had sufficient colonies formed, the cell medium was discarded, plates were rinsed with PBS, and the colonies were fixed using 6% glutaraldehyde and 0.5% crystal violet for 30 min. Then, the cells were rinsed carefully with tap water and dried at room temperature[66]. The colonies were counted using stereomicroscope and colony counter pen, the log

surviving fraction of control and RT treated colonies were calculated and plotted.

### Tumor models

All mice (C57BL/6 and FVB/NTac, 7–8 weeks) were purchased from Taconic. All mice were maintained under a tightly controlled temperature (22 °C), humidity (40–50%), light/dark (12/12 h) cycle conditions, with water and food ad libitum. To establish tumor-bearing mice (C57BL/6 male and female mice for B78, and FVB/NTac male mice for MyC-CaP), mice were intradermally engrafted with tumor cells (B78 melanoma model: $2 \times 10^6$ cells on right flank; MyC-CaP prostate tumor model: $1 \times 10^6$ cells on right flank; B78 melanoma two-tumor model: $2 \times 10^6$ cells on right flank, and 1 week later $2 \times 10^6$ cells engrafted on left flank). For TC11 breast tumor model, $5 \times 10^4$ TC11 cells were injected on the mammary fat pad of female FVB/NTac mice. Once tumor volumes reached ~100 mm³, mice were randomized and then treatment was begun. Tumors were measured twice weekly for at least 60 days after starting treatment unless mice died or were euthanized because of large tumor size (according to the animal study protocol, mice were euthanized when the diameter of tumors was ~20 mm), tumor necrosis, or evidence of pain or distress. Tumor diameters were measured with a Vernier caliper, and tumor volume was calculated through the equation: tumor volume = longer diameter × shorter diameter² × 0.5.

### Biodistribution

Cy5-labeled PIC (100 µL, 1.4 mg/mL) was intratumorally injected into the B78 melanoma bearing mice. The whole body of the mice was scanned with an in vivo imaging system (IVIS) at 3, 8, 24, 48, and 72 h after injection. For each scan, mice were anesthetized with isoflurane (4% induction and 2% maintenance) and placed on the scanner bed in a prone position. The mice were shaved in the tumors and tumor-draining lymph nodes sites before the scan.

For flow cytometry, mice were euthanized and the tumors and tumor-draining lymph nodes were collected at 3 h after injection. Tumors and tumor-draining lymph nodes were enzymatically dissociated with DNase and collagenase on a Gentle MACS Octodissociator (Miltenyi Biotec) and then filtered through a 70 µm cell strainer and red blood cells were lysed using RBC lysis buffer. Single-cell suspensions were stained with antibodies (or dyes): anti-CD11c FITC, anti-F4/80 PE-Dazzle 594, anti-CD45 PE-Cy7, anti-MHCII BV510, anti-CD11b BV711 and Live/Dead Ghost Red 780. The UltraComp Beads eBeads (Invitrogen) were used for compensation. All samples were incubated with CD16/CD32 (Fc block) for 5 min at room temperature before staining. Flow cytometry was performed on an Attune Cytometer (ThermoFisher).

### In vivo treatments

PIC (100 µL, 1.4 mg/mL) was intratumorally injected on days 0, 3, 6, and 9. The injection time points were selected to fulfill the various intended functions of PIC, which may occur at different time points relative to RT. External beam radiation therapy (EBRT) was delivered to the targeted tumors with a dose of 12 Gy on treatment day 1 using an XRad 320 cabinet irradiator (Precision X-Ray, Inc) with custom lead shielding of tissues outside of the targeted tumor site. At day 3, 6, 9, anti-CTLA-4 (IgG2c, clone 9D9, 100 µL, 1 mg/mL) was intraperitoneally injected into the mice. Tumors were measured as described above.

### RT-qPCR gene expression study in vivo

Tumor samples were collected on treatment day 15 from B78 melanoma bearing mice and homogenized using a Bead Mill Homogenizer (Bead Ruptor Elite, Omni International). Total RNA was extracted after sample homogenization using RNeasy Mini Kit (QIAGEN) according to the manufacturer's instructions. cDNA was synthesized using QuantiTect Reverse Transcription Kit (QIAGEN) according to the

manufacturer's instructions. Quantitative polymerase chain reaction (RT-qPCR) was performed using Taqman Fast Advanced Master Mix and predesigned Taqman gene expression assays for *Ifnb1*, *Mx1*, *Arg1*, *Nos2*, *Ifny*, *Il6*, *Tnfα*, *Il1β*, *Pd-l1*, *Il10* and *Tgfβ1* (the IDs of Taqman assays have been provided in Supplementary Table 4). Thermal cycling conditions and data analysis were as indicated above and HPRT was used as the endogenous control.

### Analysis of immune cells in tumors and tumor-draining lymph nodes

Tumor and tumor-draining lymph node samples were collected on day 15 post-initiation of treatment from B78 melanoma bearing mice for flow cytometry analysis (all the lymph nodes from each group were combined for analysis due to small size). Briefly, samples were enzymatically dissociated with DNase and collagenase on a Gentle MACS Octodissociator (Miltenyi Biotec) and then filtered through a 70 μm cell strainer and red blood cells were lysed using RBC lysis buffer. The single-cell suspensions were divided for the innate immune cells staining and adaptive immune cells staining separately. For innate immune cells staining, the single-cell suspensions were stained with antibodies (or dyes): anti-CD11c FITC, anti-MHCII PerCP-Cy5.5, anti-F4/80 PE-Dazzle 594, anti-CD45 PE-Cy7, anti-CD103 BV421, anti-CD206 BV605, anti-CD11b BV711, anti-CD80 APC, anti-CD317 Alexa 700, Live/Dead Ghost Red 780. For adaptive immune cells staining, the single-cell suspensions were stained with antibodies (or dyes): anti-CD4 FITC, anti-CD69 PE-Cy5, anti-CD45 PE-Cy7, anti-PD-1 BV421, anti-CD62L BV510, anti-CD3 BV605, anti-CD44 BV711, anti-CD25 APC, anti-CD8a Alexa 700, Live/Dead Ghost Red 780. After the cells were fixed and permeabilized, anti-FOXP3 PE was added to the cells for intracellular staining. The UltraComp Beads eBeads (Invitrogen) were used for compensation. All samples were incubated with CD16/CD32 (Fc block) for 5 min at room temperature before staining. Flow cytometry was performed on an Attune Cytometer (ThermoFisher) and compensation matrix and data were analyzed using FlowJo software following published flow cytometry guidelines[67].

### Evaluation of immune memory

At day 91 after the initiation of treatment, tumor-free mice in the PIC + RT + anti-CTLA-4 group in B78 melanoma model were randomized into two groups and re-challenged by engraftment of $2 \times 10^6$ B78 melanoma cells or $2 \times 10^6$ Panc02 cells on the left flank. A group of age-matched naïve mice were also engrafted with $2 \times 10^6$ B78 melanoma cells or $2 \times 10^6$ Panc02 cells for tumor growth as control. Tumor growth at these sites was monitored for another 30–50 days. After 50 days, the mice re-challenged with B78 melanoma cells were euthanized and their spleens were isolated for further analysis.

For in vitro assays, B16 melanoma cells were plated in 12-well plates (50,000 cells per well) and irradiated with sham or 8 Gy radiation in a single fraction. Five days following irradiation, spleens from naïve and re-challenged disease-free mice were collected aseptically, dissociated into a single-cell suspension, incubated in RBC lysis buffer for 10 mins and then an equal amount of PBS was added to neutralize the lysis buffer. Splenocytes were then washed with PBS. $1 \times 10^6$ splenocytes were added to each B16 culture and these cells were incubated overnight. The next day, 1 μL/sample of BD Cytofix/Cytoperm Plus kit was added to the cells for 4–6 h. The cells were then labeled with antibodies or dyes (anti-CD45 PE-Cy7, anti-CD3 FITC, anti-CD4 BV510, anti-CD8 PerCp-Cy5.5, anti-CD69 BV421, anti-GZMB PE, and Ghost Red 780) and flow cytometry was performed using an Attune cytometer (ThermoFisher) and UltraComp Beads (Invitrogen) were used for compensation[67].

### Toxicity assays

At day 7, 14, and 21 after the first injection, mice were euthanized and the blood and major organs (liver, kidney, spleen, intestine and femur)

were collected from B78 melanoma bearing mice. Blood metabolic profile analysis was performed using the VetScan Preventive Care Profile Plus rotors (Abaxis) in a VetScan VS2 blood chemistry analyzer (Abaxis). Complete blood count was analyzed by a VetScan HM5 hematology analyzer (Abaxis). Moreover, to evaluate the systemic or local toxicity, the major organs (liver, spleen, kidney, intestine and femur) were sectioned and stained with hematoxylin and eosin (H&E) and observed under an optical microscope.

### Components study

On day 0, 3, 6 and 9, 50 μL of each component (PLL: 1.05 mg/mL; ION: 1.42 mg/mL; CpG: 0.3 mg/mL) was injected into different sites of a single B78 tumor separately. EBRT was delivered to the tumor with a dose of 12 Gy on treatment day 1 using an XRad 320 cabinet irradiator with custom lead shielding of tissues outside of the targeted tumor site. At day 3, 6, 9, anti-CTLA-4 (IgG2c, clone 9D9, 100 μL, 1 mg/mL) was intraperitoneally injected into the mice. Tumors were measured as described above.

### Statistics

Prism 8 (GraphPad Software) and R (v 4.0.5) were used for all statistical analyses. Unpaired *t*-test was used for two-group comparisons. One-way ANOVA was used for the analysis of stability of PIC, cytotoxicity, clonogenic assay, cellular uptake, gene expression (in vitro and in vivo), flow cytometry studies and toxicity studies. For B78 tumor re-challenge, time-weighted average of tumor volume was calculated for each mouse, then compared by using Kruskal–Wallis test and following multiple comparison Wilcoxon rank sum test with Benjamini–Hochberg adjustment for *p*-values. For B78 melanoma tumor growth, MyC-CaP prostate tumor growth, TC11 breast tumor growth, components study and B78 melanoma two-tumor growth, a linear mixed effects model after log transformation of tumor volume data was fitted in which treatment group, day, and two-way interactions were considered as fixed effects. A complete case analysis was used, which discards only the missing measurements of deceased mice, to handle the missing data. A log-rank test was conducted to compare the survival curves, followed by multiple comparison with Benjamini–Hochberg adjustment for *p*-value. All data presented are reported as mean ± SD unless otherwise noted. For all graphs, *$P < 0.05$; **$P < 0.01$; ***$P < 0.001$; and ****$P < 0.0001$.

### Reporting summary

Further information on research design is available in the Nature Research Reporting Summary linked to this article.

## Data availability

All data reported in this work are available within the Article, Supplementary Information, or Source Data file. Source data are provided with this paper.

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

## Acknowledgements

This work was supported by the University of Wisconsin/WARF UW2020 award and in part by public health service grants P30CA014520, DP5OD024576, U01CA233102, and P01CA250972 from the National Cancer Institute. J.C.J. was supported by the NIH-NCI award F30CA250263, NIH award TL1TR002375, and NIH award T32GM008692. C.P.K. was supported by the UW-Madison Radiology MD-PhD Graduate Student Fellowship and NIH award T32GM008692.

## Author contributions

Y.Z. and R.N.S. contributed equally to this work. Y.Z., R.N.S., S.G., and Z.S.M. designed this in situ vaccination potentiation strategy. Y.Z. and S.G. designed the NP. Y.Z. fabricated, characterized, and optimized the NP. R.N.S. designed the immune and animal studies. Y.Z., R.N.S. and X.W. conducted cell experiments. R.N.S., J.C.J., Y.Z., R.A. and I.C. performed qPCR studies. R.N.S., Y.Z., P.A.C., W.J., C.P.K., and A.B. performed flow cytometry studies. R.N.S., P.A.C., Y.Z., K.A.O., C.P.K., and L.A.S. did animal experiments, including injection, measurement, and survival observation. M.Y., Y.W. and R.X. participated in nanoparticle development and characterization. T.L., T.C.H. and K.K. performed the statistical analysis. Y.Z. and R.N.S. drafted the manuscript. P.M.S. and A.L.R. contributed to the initial concept of the study and edited the manuscript. S.G. and Z.S.M. supervised the project and edited the manuscript.

## Competing interests

The authors declare the following competing interests: S.G., Z.S.M., Y.Z. and R.N.S. are inventors on a filed patent (title: nanoparticles for potentiating effects of radiation therapy on anticancer immunotherapy) managed by the Wisconsin Alumni Research Foundation relating to this work. Z.S.M. is a member of the scientific advisory board for Archeus Technologies and Seneca Therapeutics and received equity options for these companies. Z.S.M. is an inventor on patents or filed patents managed by the Wisconsin Alumni Research Foundation relating to the interaction of targeted radionuclide therapies and immunotherapies, nanoparticles designed to augment the anti-tumor immune response following radiation therapy, and the development of a brachytherapy catheter capable of delivering intra-tumor injectables. The remaining authors declare no competing interests.
