## [Peer Review File · Nature Communications]

Multifunctional nanoparticle potentiates the in situ vaccination effect of radiation therapy and enhances response to immune checkpoint blockadeReviewers' comments:

Reviewer #1 (Remarks to the Author): with expertise in iron-based nanoparticles, radiotherapy

The manuscript describes results obtained from a series of experiments attempting to demonstrate that a specific (see below for details) nanoparticle construct is responsible for observed potentiation of in situ vaccination effects of radiation therapy with immune checkpoint inhibitor (anti-CTLA4) combination therapies.

The argument(s) and data presented in the manuscript assert that the nanoparticle construction (PCFe), which comprises poly-(L-lysine) (PLL), CpG oligodeoxynucleotide (CpG), and iron oxide nanoparticle (ION) form a ~100 nm "multifunctional" entity that a) uniquely sensitizes cells to the effects of radiation; b) via CpG acting as Toll-like receptor-9 (TLR9) agonist will stimulate antigen presentation by dendritic cells to T cells (paraphrasing); and, c) the positively charged surface (zeta potential ~30-40 mV) would increase adsorption of local cancer-proteins or neo-antigens to the surface of PCFe released by RT, thus enhancing/facilitating phagocytosis of PCFe by innate cells to begin the process of antigen presentation, leading to the adaptive immune cascade which is further enhanced by treatment with CTLA-4 blockade, one of the known immune checkpoints upon which cancers may depend to maintain an immunosuppressive microenvironment ensuring continuing immune escape.

The results presented are intriguing, and certainly there is merit to them - but the presentation is lacking in critical data to support the claims made by the authors. Data presented are descriptive and provide some association to demonstrate there is an effect that correlates with some of the variables presented, but there is insufficient data to support the claims as neither mechanistic data nor data refuting alternative/competing hypotheses are presented. Specific examples follow:

It is claimed that "...both ION and PCFe significantly enhanced the radiosensitivity of B78 melanoma to radiation, demonstrating the RT sensitizing function of the ION component and PCFe." This is not supported by data in Fig 2 as there are no data to demonstrate unambiguously a "sensitizing" effect. A slightly increased toxicity is shown in Fig 2I in clonogenic survival assays with B78 melanoma cells when they were treated with a combination of one RT dose (9Gy) and a single PCFe (dose and time unclear). Other combined RT+PCFe doses produced no measurable difference against controls. Other doses of PCFe were tested with viability assays, not clonogenic survival making comparisons impossible. Whether the decrease in reproductive cell capacity shown in 2I, at only one dose combination constitutes "sensitization" or simply compounded toxicities is never demonstrated - e.g. Fe catalyzes production of free radicals via Fenton reaction which increases DNA damage, simply adding to the same biological insults of the RT; but, it has not been shown that the PCFe act to increase biological sensitivity of cells to RT-induced DNA damage. What the authors mean by "sensitization" thus is unclear, and results presented are misleading while also not supporting claims.

Data provided in Fig. 2 j-n are qualitatively similar to the work of Zanganeh et al. as cited by the authors. This is a noteworthy confirmation of mechanistic action of Fe generating oxidative stress, which does support the authors claim that Fe is a key ingredient in their PCFe to induce pro-inflammatory stress phenotypes in innate immune cells. But this has already been demonstrated, and is known, and was presumably part of the design. What is not demonstrated in the manuscript is how this leads (by what mechanism) to the tumor suppression (relatively minor concern).

What is also not demonstrated (major concern) in the manuscript is that the CpG on the nanoparticles is acting to stimulate the immune cascade via TLR-9 pathway activation. This was a design criterion for CpG, and its verification and validation should be demonstrated with data. CpG, acting as a free agent, or bound to another construct, has been demonstrated by others but the burden of providing evidence to validate the claim made here must be provided by the authors of this manuscript. This was not done. Similarly, the PLL coating - demonstrated to have some effect on protein adsorption is also not verified and validated to lead to the intended biological effect(s).

It is also not demonstrated that the PLL, CpG, and Fe must be in the form of a nanoparticle as constructed, i.e. "intact". It was a design parameter to construct the nanoparticles in a specific manner (Fig 1), with intent to induce specific biological responses as part of the choice of materials and assembly. Data are not provided to validate the design inputs with data outputs as claimed from Results in Fig 6. Each component, was tested against "...PCFe+RT+C4 group with others to specifically test the hypothesis that the intact PCFe would achieve greater tumor response and survival compared to a single component when combined with RT and anti-CTLA-4." The experimental design a) should not thus include a no-treat control; and b) to rigorously test this hypothesis, specifically the "intact" claim, should include a mixture of the components injected into tumors. From the presented data, all individual components produced tumor growth inhibition, leading to the likely alternate hypothesis that a combination of these agents in any combination treatment could induce similar (statistically indistinguishable) response, thus the claim that "intact" PCFe is necessary is not supported by the data presented.

It is thus throughout the manuscript - some, and arguably intriguing data are presented - but data critical and necessary to support claims are lacking. In its current form it is unfit for publication, as more data are needed to enable readers an independent assessment.

Additional questions/issues raised:

1) Figure 3g, after treatment with the combination RT+PCFe, the population of M2 macrophages is >2:1 over M1. Perhaps this is an error, but the significance of this is not discussed. As much of the claim rests on the PCFe acting to effect a change in TAMs to the M1 phenotype, this result is odd.

2) Rechallenge experiments (Fig 4) are very interesting, and provide support that the combination induces anti-cancer immunity, i.e. in situ vaccination. Nevertheless, (and notwithstanding comments on Fig 6 regarding "intact" PCFe) data provided in Fig 5 are unconvincing for "survival" or even disease progression as the study was terminated at Day 60. The data show that tumor growth was delayed following PCFe combination treatment, when compared among groups, but they were growing. As these data were collected from syngeneic (implanted) tumors, and not spontaneous, with the arbitrary (and perhaps premature?) study cutoff - claims of survival advantage(s) are unsupported.

3) Similar to above, experiments in bilateral implanted models (note that secondary tumor was implanted later than the first), while providing insights are not considered by many to be reliable models for making claims of "cure" and "clinical translation". Claims should be more realistic and modest. Tumor immunology input could be helpful to interpret results from these models and experiments.

Reviewer #2 (Remarks to the Author): with expertise in nanoparticles, cancer immunotherapy

Though radiation therapy has been demonstrated to activate an in situ vaccination effect in preclinical and clinical settings, this effect may be limited because it may also stimulate suppressive immune mechanisms in the tumor-immune microenvironment. In this paper, Ying Zhang and co-authors developed a PCFe nanoparticle to augment the in situ vaccination effect of RT. This nanoparticle sensitized tumors to RT, stimulated antigen presenting cells, favored M1 polarization of macrophages, and elicited a type I interferon response. These thereby induced improved tumor responses in immunologically "cold" murine models of melanoma, prostate cancer, and breast cancer. Overall this is a novel and interesting study for cancer immunotherapy and an impressive amount of work was conducted. However, there are a few points that should be clarified/further addressed:

1. Concerning the in situ vaccination strategy, the antigens released by RT to activate DCs and the antigen presentation is not clear. The authors should consider doing a proteomics experiment to further confirm the antigens released by the tumor cells and nanoparticle capture of the antigen. Similar to this paper: (Wang Y, et al. An amphiphilic dendrimer as a light-activable immunological adjuvant for in situ cancer vaccination, Nature communications, 2021, 12(1): 1-16.).

2. How the PCFe nanoparticles absorb the tumor-associated antigens and whether they can be delivered to antigen presenting cells in the lymph nodes is unknown. The biodistribution of the PCFe nanoparticles after intratumoral injection should be investigated using IVIS imaging and flow.
3. The focus on the number of CD8+ T cells is a little confusing. While CD8+ responses are required, other markers are of equal importance. PD-1, for instance, can give insights into the exhaustion state of T cells in the tumor environment. This is to say that it is marginally interesting to count the cells, but much more interesting to look at their effector state (CD44), their memory state (CD62L), and their exhaustion state. In addition, CD4+ responses are also important in anti-tumoral vaccination, so CD4+ cell effector state and exhaustion state should also be analyzed. Overall, characterization of the tumor immune microenvironment is missing in almost all of the in vivo antitumor experiments.
4. It is very interesting that the combination of PCFe with RT activates the cGAS/STING pathway. However, the mechanism behind this phenomenon should be further investigated. One possible mechanism could be that free DNA released from dying tumor cells was internalized by immune cells, which with the help of PLL may enter the cytoplasm and activate the cGAS/STING pathway. Investigating the potential mechanisms behind this phenomena would help the readers to better understand cGAS/STING pathway activation by RT + PCFe treatment.
5. Whether or not the PCFe + RT + CTLA-4 treatment induced toxicity to the mice is unknown. The authors should report mice body weight change during RT, RT + PCFe or RT + PCFe + CTLA-4 treatments. Moreover, histology analysis and blood biochemistry assays are needed to investigate the biocompatibility and toxicity of the RT + PCFe + CTLA-4 therapy.
6. P values in Figures S2, S4, S5, S6, S7, S8, S10, and S13 should be reported.
7. In the stability experiment (Figure 2e), the solvent used to disperse the PCFe nanoparticles is unclear. Usually the stability of nanoparticles in both PBS and serum should be reported.
8. Since an intratumoral injection was performed in this study in different tumor models, the authors should include some discussion on how to translate these PCFe nanoparticles into the clinic.
9. PCFe nanoparticle and CTLA-4 dose information should be added in the figure legends in Figures 3-6
10. In Figure S3 the scale bar is missing.

Reviewer #3 (Remarks to the Author): with expertise in radio-immunotherapy

This well-written manuscript addresses the important problem of overcoming the resistance of many poorly immunogenic “cold” tumors to immunotherapy. The authors have a strong track record of work in this area and specifically in the use of radiation therapy to generate an in situ vaccine leading to T cell responses that are therapeutically effective in combination with immune checkpoint blockade (ICB). Here they leverage preclinical tumor models of melanoma, prostate cancer, and breast cancer that they have previously characterized for the response to radiation + ICB to test the ability of a multifunctional nanoparticle composed of polylysine, CpG, and iron oxide (PCFe) to augment the in situ vaccination effect of RT. Each component of the nanoparticles is rationally chosen to increase radiation sensitivity (iron oxide nanoparticles), stimulate the production of type I interferon via TLR9 (CpG), and capture antigens released by the cancer cells (PLL).

Although the results are encouraging, the mechanistic studies are superficial and the novelty is somewhat limited. The ability of CpG to increase responses to radiation has been previously demonstrated in several preclinical studies (e.g., PMID 15289307, 33340886) and some clinical trials (e.g., PMID 20697067). No experiments are performed to test whether PLL improves tumor antigen delivery to the relevant antigen-presenting cells, dendritic cells (DC), and whether it enables priming of a broader T cell response. Although the novelty comes from the design of the nanoparticles, in the absence of novel mechanisms the results obtained are incremental, and somewhat preliminary.

Specific comments:

- 1) The main hypothesis formulated by the authors is that immunosuppressive effects of radiation hamper its immunogenicity. However, the nanoparticles do not directly address immunosuppressive effects of radiation (e.g., TGFbeta, etc.), and there is only a superficial attempt at characterizing

them, focused on M2 macrophages. Results shown in Figure 3d-g do not support the premise of the study that RT increases M2 macrophages.

2) TLR9 is chiefly expressed by plasmacytoid DC, which are copious producers of IFN-I. However, no attempt is made to test the role of pDC and/or the expression of TLR9 by different cells.

3) Figure 2: wouldn't the particles also sensitize DC and TAM to RT? Why a clonogenic assay is not shown for RAW cells?

4) Figure 2i-j: it is unclear how important is the induction of IFN-I in B78 melanoma cells if it takes 7 days to detect it, while in RAW cells it occurs after 24 hours. Also, how many cells survive at 7 days? Are the cells senescent and do they produce other inflammatory factors? Why ION seems to radiosensitize B78 more than the PCFe at doses > 6 Gy?

5) Experiments in vitro with macrophages are performed using a cell line. Are RAW cells representative of the behavior of primary macrophages? Do they express TLR9?

6) Figure 3: the radiation dose used in vivo is 12 Gy while in vitro is 8 Gy. It is unclear what is the radio-sensitizing factor of the particles at this dose. Panel 3c: it is unclear how CD8 T cells are quantified: percentage of CD45+ cells? Percentage of CD3+ cells? Total number is needed here, and the changes in tumor volume need to be shown for the different treatment groups. Also, why CD4 T cells are ignored? What is the ratio of CD8/CD4 effector versus Treg?

Panel 3h: given the known importance of cDC1, and of their activation state in the tumor for effective anti-tumor immune rejection (PMID 30352680), a more informative analysis of DC phenotype is required. In addition, their frequency is expressed as percentage w/o specifying the denominator.

Panels 3i-n: the data are difficult to interpret w/o knowing what is the tumor volume at day 7 and 15. Also, the increase in Mhc-1 expression could be reflecting an increased expression on cancer cells and/or the increased infiltration of the tumor by immune cells that express more Mhc-1 than the cancer cells. This bulk analysis is not informative.

7) Figure 4: why PCFe injections are repeated after RT if the main effect is to increase the radiation sensitivity to RT? Is this necessary?

8) Figure 6 f and g: why survival is not different between RT+C4, RT+PCF3 and RT+C4+PcFe given that only the latter leads to a much better local control of the irradiated and the abscopal tumor? Is this control short lived? The expected radio-sensitization by the nanoparticles in vivo is not obvious.

9) Figure S11: the sharp demarcation of vitiligo, which involved the lower but not upper body is intriguing, and begs the question whether there is a relationship with exposure to radiation. Although mice were shielded, was this area exposed to some scattered radiation?

Reviewer #4 (Remarks to the Author): with expertise in radio-immunotherapy

In this manuscript, the authors developed a nanoparticle that are capable uptake antigens and increase antigen presentation to activate T cells. Below are my comments and suggestions.

- "Moderate" word used in Abstract – line 24 –and in Introduction – line 55- is vague. Please define.
- Figure 1 in Results is not a result and it should be added as last Figure depicting the work model for this study.
- Is there a risk of radiation to increase the release of self antigens and promote autoimmunity using PCFe approach?
- "Preparation, characterization of PCFe" – Results – line 129 – 140 should be in Methods not in

Results.

- Figure 2K – How RT alone did not increase IFN β 1 compared to control? Based on previous studies, we should see increase after radiation.
- Ifn β 1 should be also measured using Elisa.
- CD206, CD204 and CD163 is more reliable marker for M2 than Arg1 and it would be interesting to analyze this marker after PCFe+RT. In addition, RAW cell line is very different from primary macrophages and the results maybe be very different compared to RAW.
- Since RT+PCFe increased Arg1 and the method measuring both Arg1 and Nos2 is relative quantification, the data do not support the conclusion related to M2>M1 polarization effect in this study.
- Figure S3. – Merged Images have different adjust compared to DAPI and FITC. Images should have exact the same adjust for all groups.
- “PCFe immunomodulates radiated tumor cells and the radiated tumor microenvironment in vivo” line 211- it should be irradiated and not ‘radiated”
- In vivo studies- – how the authors picked radiation dose of 12 Gy?

Point-by-point response to Reviewer comments

Note: We changed the abbreviation of the nanoparticle from PCFe to PIC in this revised manuscript.

Reviewer #1 (Remarks to the Author): with expertise in iron-based nanoparticles, radiotherapy

The manuscript describes results obtained from a series of experiments attempting to demonstrate that a specific (see below for details) nanoparticle construct is responsible for observed potentiation of in situ vaccination effects of radiation therapy with immune checkpoint inhibitor (anti-CTLA4) combination therapies.

The argument(s) and data presented in the manuscript assert that the nanoparticle construction (PCFe), which comprises poly-(L-lysine) (PLL), CpG oligodeoxynucleotide (CpG), and iron oxide nanoparticle (ION) form a ~100 nm "multifunctional" entity that a) uniquely sensitizes cells to the effects of radiation; b) via CpG acting as Toll-like receptor-9 (TLR9) agonist will stimulate antigen presentation by dendritic cells to T cells (paraphrasing); and, c) the positively charged surface (zeta potential ~30-40 mV) would increase adsorption of local cancer-proteins or neo-antigens to the surface of PCFe released by RT, thus enhancing/facilitating phagocytosis of PCFe by innate cells to begin the process of antigen presentation, leading to the adaptive immune cascade which is further enhanced by treatment with CTLA-4 blockade, one of the known immune checkpoints upon which cancers may depend to maintain an immunosuppressive microenvironment ensuring continuing immune escape.

The results presented are intriguing, and certainly there is merit to them - but the presentation is lacking in critical data to support the claims made by the authors. Data presented are descriptive and provide some association to demonstrate there is an effect that correlates with some of the variables presented, but there is insufficient data to support the claims as neither mechanistic data nor data refuting alternative/competing hypotheses are presented. Specific examples follow:

1. It is claimed that "...both ION and PCFe significantly enhanced the radiosensitivity of B78 melanoma to radiation, demonstrating the RT sensitizing function of the ION component and PCFe." This is not supported by data in Fig 2 as there are no data to demonstrate unambiguously a "sensitizing" effect. A slightly increased toxicity is shown in Fig 2I in clonogenic survival assays with B78 melanoma cells when they were treated with a combination of one RT dose (9Gy) and a single PCFe (dose and time unclear). Other combined RT+PCFe doses produced no measurable difference against controls. Other doses of PCFe were tested with viability assays, not clonogenic survival making comparisons impossible. Whether the decrease in reproductive cell capacity shown in 2I, at only one dose combination constitutes "sensitization" or simply compounded toxicities is never demonstrated - e.g. Fe catalyzes production of free radicals via Fenton reaction which increases DNA damage, simply adding to the same biological insults of the RT; but, it has not been shown that the PCFe act to increase biological sensitivity of cells to RT-induced DNA damage. What the authors mean by "sensitization" thus is unclear, and results presented are misleading while also not supporting claims.

Response: We appreciate the reviewer's generally supportive assessment of the manuscript. The clonogenic assay is generally considered a gold-standard for evaluating radio-sensitivity in vitro and for

quantifying sensitization to radiation in vitro. Prior studies suggest that the radiosensitizing effect of IONs can be attributed to the catalytic effects of the released iron ions and the active surfaces of IONs under RT, leading to the generation of ROS in cancer cells. We have added references to support this (PMID: 28643452, 22842461, 29556359). We have clarified the dose and timing of PIC administration in our Clonogenic assays. Specifically, 4.67 $\mu\text{g}/\text{mL}$ PIC was added to media 4 hours prior to delivery of RT at doses of 0, 3, 6 or 9 Gy, and media was changed 1 hour after radiation. The colonies were quantified 7 days later. This dose of PIC was used because according to the cytotoxicity assay (**Figure S3a**), it had no direct impact on cell viability, thus allowing us to discern the effects of radiosensitization in vitro. The prior clonogenic studies, while showing trends towards radiosensitizing effects at all doses and significant difference at 9 Gy, were underpowered to detect effects at lower doses. We have replicated these studies to increase experimental power at the lower doses and our updated analysis confirms that appropriately powered studies do demonstrate radiosensitization at all doses in this assay (updated clonogenic assay in **Figure 1h**). To further evaluate the radiosensitizing effect of PIC, we quantified γH2AX foci as a measure of the DNA damage in B78 cells after treatment with RT, PIC or PIC+RT. We have added this as new data in **Figure 1i** and **1j**. We found that PIC alone did not cause DNA damage but significantly augmented the DNA damage elicited by RT.

Figure 1h-1i. (h) Clonogenic assay of B78 melanoma cells after treatment with PIC (4.67 $\mu\text{g}/\text{mL}$) and indicated radiation doses. (n=3) PIC was added to the cells 4 hours before radiation, and fresh culture media was exchanged for this PIC treatment media 1 hour after radiation. The colonies were counted at day 7. (i) The immunofluorescence images of B78 cells after indicated treatment (RT: 12 Gy; PIC: 4.67 $\mu\text{g}/\text{mL}$). (j) Quantification of foci of γH2AX as shown in (i). 50 cells in each group were analyzed with ImageJ. PIC was added to the cells 4 hours before radiation, and fresh culture media was exchanged for PIC treatment media 1 hour after radiation. The immunofluorescence images were taken 24 hours after radiation.

The qPCR analysis of *Ifn β 1* also supports a radiosensitizing effect of PIC. As shown in **Figure 2b** and **S6a**, PIC alone (4.67 $\mu\text{g}/\text{mL}$) did not increase the *Ifn β 1* expression in B78 cells at day 1, 4 and 7 post-treatment. However, at day 7, the pre-treatment of PIC significantly increased the *Ifn β 1* expression in radiated B78 cells. *Ifn β 1* is a marker of cGAS-STING activation, which is triggered by the damaged DNA caused by RT. Our results indicated that the pre-treatment with PIC can amplify this effect.

Figure S6a and 2b. The relative mRNA expression of *Ifnβ1* in B78 cells at day 1, 4 and 7 after initiation of indicated treatments. (n=3) PIC: 4.67 μg/mL; RT: 12 Gy.

The Fenton reaction is an important feature of iron-based nanoparticles. During this reaction, hydroxyl radicals can be generated to cause the damage of DNA and the death of cancer cells. In order to simulate the high concentration of H₂O₂ in tumor microenvironment, the concentration of H₂O₂ used to test the Fenton reaction of iron-based nanoparticle in vitro is typically about 100 μM (PMID: 29665677, 31423690). For the clonogenic assay and other in vitro experiments, we did not add H₂O₂ into the cell culture media, so the cytotoxicity induced by the Fenton reaction of PIC is expected to be negligible in these studies.

2. Data provided in Fig. 2 j-n are qualitatively similar to the work of Zanganeh et al. as cited by the authors. This is a noteworthy confirmation of mechanistic action of Fe generating oxidative stress, which does support the authors claim that Fe is a key ingredient in their PCFe to induce pro-inflammatory stress phenotypes in innate immune cells. But this has already been demonstrated, and is known, and was presumably part of the design. What is not demonstrated in the manuscript is how this leads (by what mechanism) to the tumor suppression (relatively minor concern).

Response: Zanganeh et al. (PMID: 27668795) demonstrated that ferumoxytol showed therapeutic effects on early mammary cancers, and lung cancer metastases in liver and lungs because of its effects in macrophage polarization. We utilized this known function of iron oxide-based nanoparticle in our design to augment the anti-tumor immunity activated by RT – yet prior studies have not conclusively demonstrated these effects in the setting of RT or their influence on the in situ vaccine effect of RT.

3. What is also not demonstrated (major concern) in the manuscript is that the CpG on the nanoparticles is acting to stimulate the immune cascade via TLR-9 pathway activation. This was a design criterion for CpG, and its verification and validation should be demonstrated with data. CpG, acting as a free agent, or bound to another construct, has been demonstrated by others but the burden of providing evidence to validate the claim made here must be provided by the authors of this manuscript. This was not done. Similarly, the PLL coating - demonstrated to have some effect on protein adsorption is also not verified and validated to lead to the intended biological effect(s).

Response: We tested the TLR-9 stimulation effect of PIC in plasmacytoid dendritic cells (pDCs, CD11c⁺CD317⁺) in vitro, and added this data to the revised manuscript. As shown in new **Figure 2i**, PIC significantly increased the expression of CD80 and CD86 on pDCs, which chiefly express TLR-9. Also, the

production of IFN- β was detected after PIC was added to the dendritic cells (**Figure S9a**). These effects (elevating CD80 and CD86 expressions and stimulating the IFN-I production) were similar for PIC (4.67 $\mu\text{g}/\text{mL}$, containing 0.5 $\mu\text{g}/\text{mL}$ CpG) and CpG (0.5 $\mu\text{g}/\text{mL}$), indicating that the function of TLR-9 stimulation for CpG was not impaired during its complexation with PLL and ION. Moreover, from the in vivo data shown in **Figure 4h**, PIC significantly increased the proportion of pDCs that are CD80 $^+$ in the PIC+RT treated tumors, consistent with a TLR-9 activation effect of PIC.

Figure 2i. The MFI of CD80 and CD86 on CD317 $^+$ CD11c $^+$ pDCs after indicated treatments. PIC: 4.67 $\mu\text{g}/\text{mL}$; CpG: 0.5 $\mu\text{g}/\text{mL}$. (n=3)

Figure S9a. The concentration of IFN- β secreted from CD11c $^+$ dendritic cells after indicated treatment for 24 hours. CpG: 0.5 $\mu\text{g}/\text{mL}$; PIC: 4.67 $\mu\text{g}/\text{mL}$. (n=3)

Figure 4h. The percentage of CD80 $^+$ cells in CD317 $^+$ CD11c $^+$ pDCs from B78 tumors. (n=5)

As to the function of PLL, we tested the antigen absorption ability of PIC by using the cell lysates generated from B78 cells. We found PIC significantly decreased the protein concentrations in the cell lysates (**Figure 1g**), indicating the antigen absorption ability of PIC. In additional new studies using a fluorophore-labeled ovalbumin antigen (FITC-Ova), we found that pre-complexation of FITC-Ova with PIC enhanced the cellular uptake of FITC-Ova in dendritic cells (**Figure 2j** and **Figure S9b**). To evaluate the capacity of PIC and specifically PLL in PIC to enhance antigen cross presentation in DCs and activation of antigen-specific T cell immunity, we injected Ova, Ova/PIC, and Ova/CpG/ION to healthy mice, and 14 days later, we collected the splenocytes from these mice and co-cultured these with B16-Ova cells in vitro (**Figure 2k**). In new **Figure 2l-2m**, we observed that after co-cultured with B16-Ova, the T cells in the splenocytes that were harvested from the Ova/PIC injected mice showed higher CD44 $^+$, CD69 $^+$ and IFN γ $^+$ percentage among both CD4 $^+$ T cells and CD8 $^+$ T cells compared to those from Ova or Ova/CpG/ION injected mice. These results demonstrated that PLL played an important role in enabling PIC-dependent priming of antigen-specific T cell immunity.

Figure 1g. The protein concentrations in B78 cell lysates after treatment with PIC (0.14 µg/mL) for 4 hours. (n=3)

Figure 2j. The percentage of FITC-Ova positive cells among CD11c⁺ DCs at 24 h after treatment. FITC-Ova: 1.67 µg/mL; PIC: 4.67 µg/mL. (n=4)

Figure S9b. The mean fluorescence intensity (MFI) of FITC-Ova in CD11c⁺ DCs at 24 h after treatment. FITC-Ova: 1.67 µg/mL; PIC: 4.67 µg/mL. (n=4)

Figure 2k-2m. (k) Scheme for the co-culture of B16-Ova cells with splenocytes extracted from Ova, Ova/PIC, or Ova/CpG/ION injected mice. Quantification of CD44⁺, CD69⁺ and IFN γ ⁺ cells out of (l) CD4⁺CD3⁺CD45⁺ and (m) CD8⁺CD3⁺CD45⁺ cells in splenocytes by flow cytometry. (n=8)

4. It is also not demonstrated that the PLL, CpG, and Fe must be in the form of a nanoparticle as constructed, i.e. "intact". It was a design parameter to construct the nanoparticles in a specific manner (Fig 1), with intent to induce specific biological responses as part of the choice of materials and assembly. Data are not provided to validate the design inputs with data outputs as claimed from Results in Fig 6. Each component, was tested against "...PCFe+RT+C4 group with others to specifically test the hypothesis that the intact PCFe would achieve greater tumor response and survival compared to a single component when combined with RT and anti-CTLA-4." The experimental design a) should not thus include a no-treat control; and b) to rigorously test this hypothesis, specifically the "intact" claim, should include a mixture of the components injected into tumors. From the presented data, all individual components produced tumor growth inhibition, leading to the likely alternate hypothesis that a combination of these agents in any combination treatment could induce similar (statistically indistinguishable) response, thus the claim that "intact" PCFe is necessary is not supported by the data presented.

Response: Thank you for this suggestion. We have performed this experiment and added this data in **Figure S28**. Due to the strong electrostatic interaction between these components (PLL, CpG and ION) and the rapid formation of nanoparticle after these components are mixed, we injected the three components in

small injection volumes (50 μL) to different sites within a B78 tumor to minimize the formation of nanoparticle via the electrostatic interaction between these components in vivo. We found the PIC+RT+C4 treatment exhibited improved tumor suppression compared to components+RT+C4 treatment. And the PIC+RT+C4 treated group showed 5/8 mice tumor-free at day 60 post-treatment, while 0/8 mice tumor-free in the components+RT+C4 treatment group.

Figure S28. (a) Scheme for the treatment of mice bearing a B78 melanoma flank tumor. (b) Average tumor growth curves of mice are displayed following the indicated treatment regimen. (n=8) (c) Individual tumor growth curves for mice in (b). RT: 12 Gy. PIC: 140 $\mu\text{g}/100\mu\text{L}/\text{dose}$. C4 (anti-CTLA-4): 100 $\mu\text{g}/100\mu\text{L}/\text{dose}$.

Additional questions/issues raised:

1) Figure 3g, after treatment with the combination RT+PCFe, the population of M2 macrophages is >2:1 over M1. Perhaps this is an error, but the significance of this is not discussed. As much of the claim rests on the PCFe acting to effect a change in TAMs to the M1 phenotype, this result is odd.

Response: We repeated the flow cytometry studies following in vivo treatments and analyzed more tumor infiltrating immune cells to increase our ability to discern significant effects in the revised manuscript (Figure 4). From Figure 4b-4d, PIC+RT treated tumors showed similar percentage of TAMs that were $\text{CD80}^+\text{CD206}^-$ M1 macrophages compare to tumors treated with RT alone, both of which were significantly higher than in tumors from untreated control mice. However, RT alone increased the percentage of TAMs that were $\text{CD206}^+\text{CD80}^-$ M2 macrophages, whereas PIC+RT treated tumors exhibited levels of M2 polarized TAMs that were comparable with that observed in untreated control tumors and reduced relative to RT alone treated tumors. Consequently, PIC favorably increased the ratio of M1: M2 macrophages among TAMs following RT. PIC+RT treated tumors showed the highest M1:M2 macrophages ratios (about 0.75 in average) among the three groups, while the ratios for untreated control and RT treated tumors are about

0.3-0.4. We modified the manuscript to better describe this effect of PIC as we agree with the reviewer that this was poorly worded in the previous version. From the data shown in **Figure 4b-4d**, PIC did not alter the favorable effect of RT in elevating the levels of M1 macrophages and eliminated a detrimental effect of RT by reducing the levels of M2 TAMs.

Figure 4. The percentage of (b) CD80⁺CD206⁻ M1-like macrophage and (c) CD206⁺CD80⁻ M2-like macrophage in CD11b⁺F4/80⁺ macrophages in B78 tumors. (d) The ratios of M1:M2 macrophages in B78 tumors after indicated treatments. PIC: 140µg/100µL/dose; RT: 12 Gy. (n=5)

2) Rechallenge experiments (Fig 4) are very interesting, and provide support that the combination induces anti-cancer immunity, i.e in situ vaccination. Nevertheless, (and notwithstanding comments on Fig 6 regarding "intact" PCFe) data provided in Fig 5 are unconvincing for "survival" or even disease progression as the study was terminated at Day 60. The data show that tumor growth was delayed following PCFe combination treatment, when compared among groups, but they were growing. As these data were collected from syngeneic (implanted) tumors, and not spontaneous, with the arbitrary (and perhaps premature?) study cutoff - claims of survival advantage(s) are unsupported.

Response: By convention we have commonly reported survival at day 60 after RT as a primary endpoint in our prior studies. However, to address the reviewer's concern, we have extended the survival data to day 80 in **Figure 6c** in the revised manuscript. For all studies in the manuscript, the mice were euthanized if the tumor size was approximately 20 mm in diameter, for occurrence of tumor ulceration, or for evidence of pain or distress. The PIC+RT+C4 triple treatment group showed improved survival compared to other groups even when this endpoint was extended to day 80.

Figure 6b-6c. (b) Average tumor growth curves of MyC-CaP prostate tumors and (c) survival rate of mice after the treatments indicated. (n=8)

As for the tumor growth shown in **Figure 6b** and **6d**, at day 60, there were 5/8 mice tumor-free in PIC+RT+C4 treatment group, and these tumor-free mice did not show any evidence of disease during the following observation for more than one month. For the other 3/8 mice, the tumor size became smaller until around one month from treatment, but these tumors began to grow beyond one month. This tumor relapse may

suggest the anti-tumor immunity induced by the PIC+RT+C4 treatment was still not potent enough to eradicate the tumors in these mice (3/8). Despite the tumor relapse, the tumor growth in these PIC+RT+C4 treated mice was slower than that observed in all other groups, demonstrating the effects of PIC in augmenting the in situ vaccination effect of RT and improving the therapeutic efficacy of RT+C4.

3) Similar to above, experiments in bilateral implanted models (note that secondary tumor was implanted later than the first), while providing insights are not considered by many to be reliable models for making claims of "cure" and "clinical translation". Claims should be more realistic and modest. Tumor immunology input could be helpful to interpret results from these models and experiments.

Response: We agree with the reviewer that these models, while useful, also have considerable limitations. We have modified the wording used in our description and discussion of these results and have further highlighted some of the limitations of our tumor models.

Reviewer #2 (Remarks to the Author): with expertise in nanoparticles, cancer immunotherapy

Though radiation therapy has been demonstrated to activate an in situ vaccination effect in preclinical and clinical settings, this effect may be limited because it may also stimulate suppressive immune mechanisms in the tumor-immune microenvironment. In this paper, Ying Zhang and co-authors developed a PCFe nanoparticle to augment the in situ vaccination effect of RT. This nanoparticle sensitized tumors to RT, stimulated antigen presenting cells, favored M1 polarization of macrophages, and elicited a type I interferon response. These thereby induced improved tumor responses in immunologically “cold” murine models of melanoma, prostate cancer, and breast cancer. Overall this is a novel and interesting study for cancer immunotherapy and an impressive amount of work was conducted. However, there are a few points that should be clarified/further addressed:

1. Concerning the in situ vaccination strategy, the antigens released by RT to activate DCs and the antigen presentation is not clear. The authors should consider doing a proteomics experiment to further confirm the antigens released by the tumor cells and nanoparticle capture of the antigen. Similar to this paper: (Wang Y, et al. An amphiphilic dendrimer as a light-activable immunological adjuvant for in situ cancer vaccination, *Nature communications*, 2021, 12(1): 1-16.).

Response: We have not determined the antigen(s) identified in B78, MyC-CaP, or TC11 cells following this in situ vaccine as this is not necessary to demonstrate the in situ vaccine effect and is beyond the scope of this manuscript. In separate studies with other treatments, we are evaluating this and observing that the antigens released following an in situ vaccination are not always identical, despite treating the same tumor model in syngeneic mice. However, a major translational appeal of the in situ vaccine approach is that identification of tumor neo-antigens is not required in order to achieve therapeutic efficacy and this is a major of appeal of the RT+PIC treatment combination in that it allows tumor antigen-specific vaccination without the need to first identify tumor associated antigens. In order to further address the reviewer’s point, however, we have performed an additional study using a known antigen (Ovalbumin) to demonstrate the capacity of PIC to facilitate antigen cross-presentation and antigen-specific T cell immunity. Now, in this revised manuscript, we demonstrated the antigen absorption ability of PIC (**Figure 1g**), PIC-facilitated antigen uptake by dendritic cells (**Figure 2j** and **S9b**), and the importance of the positive charges introduced by PLL in PIC-mediated priming of antigen-specific T cell immunity (**Figure 2k-2m**). In the context of our in vivo treatment regimen, we confirmed the tumor antigen-specificity of the immune response induced by PIC+RT+C4 by re-challenging the mice that were rendered disease-free from B78 melanoma with an additional B78 engraftment or with an engraftment of unrelated syngeneic Panc02 cells that do not share tumor antigens with B78 cells. As the data shown in **Figure S26**, the Panc02 tumor growth in tumor-free mice was similar with that in naive control mice, while B78 tumors grew only in naïve control mice and not those which had been treated with in situ vaccination. This is consistent with the development of tumor antigen-specific immune response.

Figure 1g. The protein concentrations in B78 cell lysates after treatment with PIC (0.14 µg/mL) for 4 hours. (n=3)

Figure 2j. The percentage of FITC-Ova positive cells among CD11c⁺ DCs at 24 h after treatment. FITC-Ova: 1.67 µg/mL; PIC: 4.67 µg/mL. (n=4)

Figure S9b. The mean fluorescence intensity (MFI) of FITC-Ova in CD11c⁺ DCs at 24 h after treatment. FITC-Ova: 1.67 µg/mL; PIC: 4.67 µg/mL. (n=4)

Figure 2k-2m. (k) Scheme for the co-culture of B16-Ova cells with splenocytes extracted from Ova, Ova/PIC, or Ova/CpG/ION injected mice. Quantification of CD44⁺, CD69⁺ and IFN γ ⁺ cells out of (l) CD4⁺CD3⁺CD45⁺ and (m) CD8⁺CD3⁺CD45⁺ cells in splenocytes by flow cytometry. (n=8)

Figure S26. (a) Scheme for the treatment of mice bearing a B78 melanoma and subsequent implantation with an unrelated Panc02 tumor. (b) Average Panc02 tumor growth curves are shown after these tumors were engrafted in naïve control mice or in mice rendered disease-free from a B78 melanoma by PIC+RT+anti-CTLA-4. (Control: n=5; Tumor-free mice: n=4) (c) Individual mouse tumor growth curves from (b).

2. How the PCFe nanoparticles absorb the tumor-associated antigens and whether they can be delivered to antigen presenting cells in the lymph nodes is unknown. The biodistribution of the PCFe nanoparticles after intratumoral injection should be investigated using IVIS imaging and flow.

Response: The negatively charged TAAs can be absorbed by PIC through electrostatic interaction. We tested this using the B78 cell lysates. From **Figure 1g**, the protein concentration in the B78 cell lysates significantly decreased after incubating with PIC (0.14 $\mu\text{g}/\text{mL}$) for 4 h, indicating the antigens absorption ability of PIC.

Figure 1g. The protein concentrations in B78 cell lysates after treatment with PIC (0.14 mg/mL) for 4 hours. (n=3)

We appreciate the value of the reviewer's suggestion to evaluate the biodistribution of PIC after intratumoral injection. To achieve this, we labeled PIC with Cy5 (the synthesis details are added in the methods section). After intratumoral injection, the Cy5-PIC signal can be detected in the tumor site for at least 3 days. Furthermore, the Cy5 signal was quickly detected in the tumor draining lymph nodes (TDLNs) after intratumoral injection (**Figure S10**). A different group of animals were euthanized 3h after the intratumoral injection, and the tumors and TDLNs were collected. By analyzing the single cell suspensions with flow cytometry, we found the Cy5-PIC was taken up by tumor cells, dendritic cells ($\text{CD}45^+\text{CD}11\text{c}^+\text{MHCII}^+$) and macrophages ($\text{CD}45^+\text{CD}11\text{b}^+\text{F4}/80^+$) in tumors (**Figure 3a**). TDLNs play important roles in priming anti-tumor immunity, and we found significant internalization of Cy5-PIC in dendritic cells ($\text{CD}45^+\text{CD}11\text{c}^+\text{MHCII}^+$) and macrophages ($\text{CD}45^+\text{CD}11\text{b}^+\text{F4}/80^+$) at TDLNs (**Figure 3b**).

Figure S10. IVIS images of B78 melanoma-bearing mice at indicated timepoints after Cy5-labeled PIC (Cy5-PIC) was intratumorally injected.

Figure 3a-3b. The cellular uptake of Cy5-PIC in different cells within the (a) tumors and (b) TDLNs after the Cy5-PIC was intratumorally injected into B78 melanoma bearing mice. (n=5)

3. The focus on the number of CD8⁺ T cells is a little confusing. While CD8⁺ responses are required, other markers are of equal importance. PD-1, for instance, can give insights into the exhaustion state of T cells in the tumor environment. This is to say that it is marginally interesting to count the cells, but much more interesting to look at their effector state (CD44), their memory state (CD62L), and their exhaustion state. In addition, CD4⁺ responses are also important in anti-tumoral vaccination, so CD4⁺ cell effector state and exhaustion state should also be analyzed. Overall, characterization of the tumor immune microenvironment is missing in almost all of the in vivo antitumor experiments.

Response: Thank you for the suggestions. In the revised manuscript, we repeated the flow cytometry studies on disaggregated B78 melanoma tumors following in vivo treatments and analyzed how PIC+RT influenced the tumor immune microenvironment (**Figure 4**). At day 15 after the first injection (**Figure 4a**), the CD8⁺ T cells infiltration (CD8⁺ out of CD3⁺CD45⁺ cells) in B78 tumors was higher in PIC+RT treated tumors than those treated with RT alone (p=0.048) (**Figure 4i**). We did not observe any significant changes in the PD-1 expression on CD8⁺ T cells after treatment (**Figure 4o**), but found the percentage of activated CD8⁺ T cells (CD69⁺CD8⁺) and effector CD8⁺ T cells (CD44⁺CD8⁺) out of CD3⁺CD45⁺ cells increased in PIC+RT treated tumors compared to RT alone treated tumors (p=0.019 for CD69⁺CD8⁺ cells; p=0.061 for CD44⁺CD8⁺ cells) (**Figure 4m**). By analyzing the CD4⁺ T cells, we found PIC+RT treated tumors showed higher percentage of CD4⁺ T cells out of CD3⁺CD45⁺ cells compared to untreated control tumors, although there was no significant difference between RT vs PIC+RT and control vs RT (**Figure 4j**). Interestingly, we found RT alone increased the percentage of CD25⁺FOXP3⁺ Tregs out of CD4⁺ T cells, but PIC+RT eliminated this effect and the percentage of Tregs (CD25⁺FOXP3⁺) in PIC+RT group was similar to untreated control tumors (**Figure 4k**). Similar to CD8⁺ T cells, we did not observe obvious changes of PD-1 expression on CD4⁺ T cells after treatment (**Figure 4n**) and although the percentage of activated CD4⁺ T cells (CD69⁺CD4⁺) and effector CD4⁺ T cells (CD44⁺CD4⁺) did not significantly increased in PIC+RT treated tumors compared to those treated with RT, we observed a more significant increase when comparing PIC+RT group with untreated control relative to the comparison of the RT alone and untreated control groups (**Figure 4l**). Due to the small size of TDLNs, we combined the TDLNs samples from each treatment group to obtain a higher quality flow cytometry data for the study of activated, effector and memory T cells in the TDLNs. In **Figure S24**, we observed an elevated CD69⁺ percentage and CD44⁺ percentage among both CD4⁺CD3⁺CD45⁺ and CD8⁺CD3⁺CD45⁺ T cells in the TDLNs in PIC+RT group than that in RT group. And from **Figure 4p**, the percentage of central memory T cells (CD44⁺CD62L⁺) increased after PIC+RT treatment. We have updated the manuscript with this additional data.

Figure 4. The percentage of (i) CD8⁺ and (j) CD4⁺ cells in CD45⁺CD3⁺ T cells in B78 tumors. (k) The percentage of CD25⁺FOXP3⁺ Tregs in CD4⁺ T cells in B78 tumors. (l) The percentage of CD69⁺CD4⁺ and CD44⁺CD4⁺ cells in CD45⁺CD3⁺ T cells in B78 tumors. (m) The percentage of CD69⁺CD8⁺ and CD44⁺CD8⁺ cells among CD45⁺CD3⁺ T cells in B78 tumors. The MFI of PD-1 on (n) CD3⁺CD4⁺ T cells and (o) CD3⁺CD8⁺ T cells in B78 tumors. (n=5) (p) The percentage of central memory T cells (CD44⁺CD62L⁺), effector memory T cells (CD44⁺CD62L⁻), naive T cells (CD44⁻CD62L⁺) and others (CD44⁻CD62L⁻) among CD3⁺CD4⁺ T cells (left) and CD3⁺CD8⁺ T cells (right) in TDLNs. PIC: 140µg/100µL/dose; RT: 12 Gy.

Figure S24. The percentage of (a) CD44⁺ and (b) CD69⁺ cells out of CD3⁺CD4⁺ T cells (left) and CD3⁺CD8⁺ T cells (right) in TDLNs after indicated treatments. (n=5) PIC: 140µg/100µL/dose; RT: 12 Gy.

4. It is very interesting that the combination of PCFe with RT activates the cGAS/STING pathway. However, the mechanism behind this phenomenon should be further investigated. One possible mechanism could be that free DNA released from dying tumor cells was internalized by immune cells, which with the help of PLL may enter the cytoplasm and activate the cGAS/STING pathway.

Response: This is a good point. Yes, there is a possibility that PIC can potentiate the internalization of released cell-free DNA from the dying tumor cells by immune cells and activate the cGAS/STING pathway. Because PIC can be taken up by both tumor cells and immune cells, and we have demonstrated that PIC can directly activate a type I IFN response in some immune cells (e.g. pDCs, in contrast to our observation in tumor cells), it is challenging to test and identify or exclude various routes whereby PIC may increase cGAS/STING activation. We do show that pre-treatment of tumor cells with PIC increased the induction of *Irf1* following radiation, and in tumor cells this could not be replicated by treating with PIC after RT (Figure 2b and S6a). In contrast, we show that in at least some immune cells (e.g. pDCs) PIC can directly stimulate

TLR-9 (**Figure S14**) leading to increased expression of *Ifnβ1*. We agree with the reviewer that PIC may also contribute to additional mechanisms that lead to further activation of a type I interferon response in various immune cell lineages and we have added discussion of this in the manuscript now, noting that this is an important topic for further investigation.

Figure S6a and 2b. The relative mRNA expression of *Ifnβ1* in B78 cells at day 1, 4 and 7 after initiation of indicated treatments. (n=3) PIC: 4.67 μg/mL; RT: 12 Gy.

Figure S14. The mRNA expression of *Ifnβ1* and *Mx1* in the B78 tumors at day 15 after PIC or control treatment. (n=5) PIC: 140μg/100μL/dose. PIC was intratumorally injected at day 0, 3, 6 and 9.

5. Whether or not the PCFe + RT + CTLA-4 treatment induced toxicity to the mice is unknown. The authors should report mice body weight change during RT, RT + PCFe or RT + PCFe + CTLA-4 treatments. Moreover, histology analysis and blood biochemistry assays are needed to investigate the biocompatibility and toxicity of the RT + PCFe + CTLA-4 therapy.

Response: Thank you for the suggestions. We supplemented the toxicity studies of both PIC+RT and PIC+RT+C4 in B78 melanoma bearing mice in the revised manuscript (**Figure S29a**). At days 0 (control), 7, 14, and 21 after the first PIC injection, we tested the body weight of mice, and then the mice were euthanized, and the blood was collected to test complete blood counts, basic metabolic panels, and to evaluate major organ histology (liver, kidney, spleen, intestine, and femur). We did not observe any obvious body weight changes during the PIC+RT and PIC+RT+C4 treatments (**Figure S29b**).

Figure S29. (a) Scheme for delivery of treatment for in vivo toxicity studies. (b) No significant change was observed in the body weight of mice bearing a B78 melanoma after treatment with PIC + RT or PIC+RT+C4. (n=3) RT: 12 Gy. PIC: 140 $\mu\text{g}/100\mu\text{L}/\text{dose}$. C4 (anti-CTLA-4): 100 $\mu\text{g}/100\mu\text{L}/\text{dose}$.

About 100 μL freshly collected blood was used to analyze complete blood counts. From the data shown in **Figure S30**, we found LY% (percentage of lymphocytes) slightly decreased one week after the first PIC injection, and this effect was resolved after 3 weeks. This may be attributed to the sensitivity of lymphocytes to radiation, which is consistent with previous reports from pre-clinical and clinical studies in that RT may induce a transient reduction of LY% (PMID: 29490633, 32656085). PIC+RT and PIC+RT+C4 did not show significant influence on the MON% (percentage of monocytes), and the counts of RBC (red blood cells) and PLT (platelets).

Figure S30. The complete blood counts of B78 melanoma bearing mice after indicated treatments. (n=3) RT: 12 Gy. PIC: 140 $\mu\text{g}/100\mu\text{L}/\text{dose}$. C4 (anti-CTLA-4): 100 $\mu\text{g}/100\mu\text{L}/\text{dose}$.

We conducted metabolic evaluations using the serum from mouse blood. As shown in new **Figure S31**, the kidney function (BUN, CRE) and liver function (ALT, AST, ALP and TBIL) were not impaired by the PIC+RT and PIC+RT+C4 treatments. We also found that these treatments did not influence the levels of glucose, total proteins, albumin, electrolytes (CA, NA^+ , K^+ and Cl^-), and total carbon dioxide in the blood.

Figure S31. The blood metabolic profiles of mice bearing a B78 melanoma after indicated treatments. (n=3) RT: 12 Gy. PIC: 140 $\mu\text{g}/100\mu\text{L}/\text{dose}$. C4 (anti-CTLA-4): 100 $\mu\text{g}/100\mu\text{L}/\text{dose}$.

We sectioned the major organs (liver, kidney, spleen, intestine, and femur) and studied the effects of the treatments on these organs by histology analysis. From the H&E staining images shown in **Figure S32**, we did not observe any obvious damage in these organs 1, 2, and 3 weeks after PIC+RT and PIC+RT+C4 treatment.

Figure S32. Images of H&E stained sections of major organs (liver, spleen, kidney, intestine and femur) from mice bearing a B78 melanoma and treated with PIC+RT or PIC+RT+C4. RT: 12 Gy. PIC: 140 $\mu\text{g}/100\mu\text{L}/\text{dose}$. C4 (anti-CTLA-4): 100 $\mu\text{g}/100\mu\text{L}/\text{dose}$.

6. P values in Figures S2, S4, S5, S6, S7, S8, S10, and S13 should be reported.

Response: We regret this omission and have added the p value for all figures in the revised manuscript.

7. In the stability experiment (Figure 2e), the solvent used to disperse the PCFe nanoparticles is unclear. Usually the stability of nanoparticles in both PBS and serum should be reported.

Response: The solvent we used to test the stability of PIC in **Figure 1f** in the revised manuscript is DI water. We dispersed PIC in DI water, stored it at 4°C, and monitored the changes of particle size. The stability of the PIC has been updated to 1 month in **Figure 1f**. Moreover, we tested the stability of PIC in 1mM PBS. In **Figure S2a**, during the storage at 4°C in 1mM PBS, we did not observe any obvious change in the particle size of PIC. We did not test the stability of PIC in the presence of serum, because the positively charged PIC was designed to absorb negatively charged tumor-associated antigens after intratumor injection, thus there would be a high likelihood that PIC would absorb the negatively charged proteins in serum. However, since PIC was intratumorally injected into tumor bearing mice to improve the in situ vaccination effect of RT, its stability in serum would not directly affect its intended functions.

Figure 1f. The particle size stability of PIC during storage at 4 °C. (n=3)

To further explore the storability of PIC, we lyophilized PIC (**Figure S2b-S2c**). We found the particle size and zeta potential did not change during the lyophilization in the presence of 1% sucrose. Then we stored the lyophilized PIC at -20 °C, and the changes of particle size and zeta potential were negligible during the 3 months storage.

Figure S2. (a) The particle size of PIC in the presence of 1mM PBS during storage at 4°C. The (b) particle size and (c) zeta potential of lyophilized PIC in the presence of 1% sucrose during storage for 12 weeks at -20°C. (n=3) L: PIC after lyophilization.

8. Since an intratumoral injection was performed in this study in different tumor models, the authors should include some discussion on how to translate these PCFe nanoparticles into the clinic.

Response: We expect this particle would be delivered by intratumoral injection in patients and we have added discussion of this in the manuscript, noting there is precedence FDA approval of intratumoral injected agents in patients with metastatic cancers.

9. PCFe nanoparticle and CTLA-4 dose information should be added in the figure legends in Figures 3-6

Response: This information has been added in the figure legends.

10. In Figure S3 the scale bar is missing.

Response: The scale bar has been added in **Figure S5** in the revised manuscript.

Figure S5. Confocal laser scanning microscopy (CLSM) images of B78 cells and RAW264.7 cells after treatment with FITC-labeled PIC for 2h.

Reviewer #3 (Remarks to the Author): with expertise in radio-immunotherapy

This well-written manuscript addresses the important problem of overcoming the resistance of many poorly immunogenic “cold” tumors to immunotherapy. The authors have a strong track record of work in this area and specifically in the use of radiation therapy to generate an in situ vaccine leading to T cell responses that are therapeutically effective in combination with immune checkpoint blockade (ICB). Here they leverage preclinical tumor models of melanoma, prostate cancer, and breast cancer that they have previously characterized for the response to radiation + ICB to test the ability of a multifunctional nanoparticle composed of polylysine, CpG, and iron oxide (PCFe) to augment the in situ vaccination effect of RT. Each component of the nanoparticles is rationally chosen to increase radiation sensitivity (iron oxide nanoparticles), stimulate the production of type I interferon via TLR9 (CpG), and capture antigens released by the cancer cells (PLL).

Although the results are encouraging, the mechanistic studies are superficial and the novelty is somewhat limited. The ability of CpG to increase responses to radiation has been previously demonstrated in several preclinical studies (e.g., PMID 15289307, 33340886) and some clinical trials (e.g., PMID 20697067). No experiments are performed to test whether PLL improves tumor antigen delivery to the relevant antigen-presenting cells, dendritic cells (DC), and whether it enables priming of a broader T cell response. Although the novelty comes from the design of the nanoparticles, in the absence of novel mechanisms the results obtained are incremental, and somewhat preliminary.

Response: We have added data to the manuscript in which we evaluate and demonstrate the functional importance of PLL for the efficacy of PIC in improving the in situ vaccine response to RT. The designed functions of PLL in PIC include capturing tumor antigens and promoting their uptake in antigen presenting cells, thereby facilitating the priming of a potent T cell response. We tested the ability of PIC to capture the tumor antigens by using the cell lysates prepared from B78 cells. As shown in **Figure 1g**, PIC significantly decreased the protein concentrations in the cell lysates, suggesting its antigen absorption ability. Then, using a fluorophore-labeled ovalbumin antigen, FITC-Ova, we found that pre-complexation of FITC-Ova with PIC improved the cellular uptake of FITC-Ova in dendritic cells compared to FITC-Ova alone (**Figure 2j** and **Figure S9b**). To study the importance of the positive charges introduced by PLL in PIC on the activation of antigen-specific T cell immunity, we injected healthy mice with Ova, Ova/PIC, and Ova/ION/CpG. 14 days later, we collected the splenocytes from these mice and co-cultured them with B16-Ova cells in vitro (**Figure 2k**). As presented in new **Figure 2l-2m**, after co-culture with B16-Ova, a greater percentage of CD4⁺ T cells and CD8⁺ T cells in splenocytes from Ova/PIC injected mice exhibited expression of CD44⁺, CD69⁺ and IFN γ ⁺ compared to those T cells from the splenocytes of Ova or Ova/CpG/ION injected mice. These results demonstrate that PLL plays an important role in facilitating priming of a potent antigen-specific T cell immunity by PIC.

Figure 1g. The protein concentrations in B78 cell lysates after treatment with PIC (0.14 mg/mL) for 4 hours.

(n=3)

Figure 2j. The percentage of FITC-Ova positive cells among CD11c⁺ DCs at 24 h after treatment. FITC-Ova: 1.67 µg/mL; PIC: 4.67 µg/mL. (n=4)

Figure S9b. The mean fluorescence intensity (MFI) of FITC-Ova in CD11c⁺ DCs at 24 h after treatment. FITC-Ova: 1.67 µg/mL; PIC: 4.67 µg/mL. (n=4)

Figure 2k-2m. (k) Scheme for the co-culture of B16-Ova cells with splenocytes extracted from Ova, Ova/PIC, or Ova/CpG/ION injected mice. Quantification of CD44⁺, CD69⁺ and IFN γ ⁺ cells out of (l) CD4⁺CD3⁺CD45⁺ and (m) CD8⁺CD3⁺CD45⁺ cells in splenocytes by flow cytometry. (n=8)

Specific comments:

1) The main hypothesis formulated by the authors is that immunosuppressive effects of radiation hamper its immunogenicity. However, the nanoparticles do not directly address immunosuppressive effects of radiation (e.g., TGF β , etc.), and there is only a superficial attempt at characterizing them, focused on M2 macrophages. Results shown in Figure 3d-g do not support the premise of the study that RT increases M2 macrophages.

Response: In the revised manuscript, we tested the expression of more inflammatory genes (*Ifn β 1*, *Mx1*, *Nos2*, *Arg1*, *Ifn γ* , *Il6*, *Tnf α* , *Il1 β* , *Il10*, *Tgf β 1*) in the tumors to study the effects of PIC on the immune microenvironment of radiated tumors (Figure 3c). As shown in Figure 3d-3e, PIC+RT treated tumors exhibited higher expression of IFN-I genes (*Ifn β 1*, *Mx1*) compared to those treated with RT, which may be associated with the radio-sensitization effect and TLR-9 stimulation effect of PIC. The expression levels of *Nos2* (M1 macrophage marker) and *Arg1* (M2 macrophage marker) in tumors also increased after PIC was injected into the radiated tumors, with greater magnitude of *Nos2* mRNA elevation compared to that of *Arg1* mRNA. We found that the expression levels of *Ifn γ* , which plays critical roles in the anti-tumor immunity by promoting the activity of T helper type 1 cells and CD8⁺ cytotoxic T lymphocytes, also significantly increased in the PIC+RT group compared to the RT alone group. Furthermore, we found PIC increased the expression of both pro-inflammatory genes (*Tnf α* , *Il1 β*) and anti-inflammatory genes (*Tgf β 1*, *Il10* and *Il6*). But the elevation was greater for the expression of pro-inflammatory genes, leading to higher expression ratios of pro-inflammatory genes: anti-inflammatory genes (*Tnf α* : *Tgf β 1* and *Il1 β* : *Il10*) for PIC+RT group than RT treated tumors.

In addition, we repeated the flow cytometry study of disaggregated tumors following in vivo treatments and evaluated the effects of RT and PIC+RT on the populations of additional immune cell lineages. As shown

in new **Figure 4b-4d**, RT increased the percentage of both tumor-infiltrating M1 (CD80⁺CD206⁻) and M2 (CD206⁺CD80⁻) macrophages out of CD11b⁺F4/80⁺ macrophages; PIC maintained the increased levels of M1 macrophages induced by RT but reduced the levels of M2 macrophages when it was injected into the irradiated tumors, resulting in higher ratios of M1:M2 macrophages in PIC+RT treated tumors compared to those treated with RT. We have modified the wording used to describe this effect in the manuscript and agree this was poorly phrased in the prior draft. At day 15 post-treatment, we also observed higher ratios of CD25⁺FOXP3⁺ Tregs out of CD4⁺ T cells in RT treated tumors compared to control tumors, while PIC eliminated this detrimental effect of RT and reduced the percentage of Tregs to levels similar to control tumors (**Figure 4k**).

Figure 3c-3e. (c) Scheme for the study of the immunomodulation effect of PIC+RT on the B78 tumor microenvironment. (d) The mRNA expression of select inflammatory and anti-inflammatory genes (*Ifnβ1*, *Mx1*, *Nos2*, *Arg1*, *Ifnγ*, *Il6*, *Tnfa*, *Il1β*, *Il10* and *Tgfβ1*) in B78 tumors at day 15 after initiation of indicated treatments. (e) The mRNA expression ratios of *Nos2: Arg1*, *Tnfa: Tgfβ1* and *Il1β: Il10* in tumors at day 15 after indicated treatments. (n=5) PIC: 140μg/100μL/dose; RT: 12 Gy.

Figure 4. The percentage of (b) CD80⁺CD206⁻ M1-like macrophage and (c) CD206⁺CD80⁻ M2-like

macrophage in CD11b⁺F4/80⁺ macrophages in B78 tumors. **(d)** The ratios of M1:M2 macrophages in tumors after indicated treatments. **(k)** The percentage of CD25⁺FOXP3⁺ Tregs in CD4⁺ T cells in B78 tumors. (n=5)

2) TLR9 is chiefly expressed by plasmacytoid DC, which are copious producers of IFN-I. However, no attempt is made to test the role of pDC and/or the expression of TLR9 by different cells.

Response: In the revised manuscript, we tested the stimulation of TLR-9 in pDCs by PIC both in vitro and in vivo. For in vitro studies, we treated dendritic cells with CpG (0.5 µg/mL) or PIC (4.67 µg/mL, containing 0.5 µg/mL CpG). As the data shown in **Figure 2i**, both CpG and PIC increased the expression of CD80 and CD86, two markers of TLR-9 stimulation, on CD11c⁺CD317⁺ pDCs. The effects on CD80 and CD86 were similar for CpG treated pDCs with those treated with PIC, indicating that the TLR-9 stimulation function of CpG was not impaired after it was complexed with PLL and ION to form a nanoparticle. IFN-I secretion is another marker of TLR-9 activation in pDCs. In new **Figure S9a**, we found that both CpG and PIC stimulated pDCs to secrete IFN-β. We further tested these effects of PIC on pDCs in vivo. We found PIC+RT treated tumors exhibited a higher percentage of CD80⁺ cells among CD11c⁺CD317⁺ pDCs than RT-alone treated tumors (**Figure 4h**), and PIC treatment increased the mRNA expression of *Ifnβ1* (type I interferon) and *Mx1* (induced by type I interferon) in both control tumors and RT treated tumors (**Figure 3d** and **S14**).

Figure 2i. The MFI of CD80 and CD86 on CD317⁺CD11c⁺ pDCs after indicated treatments. PIC: 4.67 µg/mL; CpG: 0.5 µg/mL. (n=3)

Figure S9a. The concentration of IFN-β secreted from CD11c⁺ dendritic cells after indicated treatment for 24 hours. CpG: 0.5 µg/mL; PIC: 4.67 µg/mL. (n=3)

Figure 4h. The percentage of CD80⁺ cells in CD317⁺ CD11c⁺ pDCs from B78 tumors. (n=5)

Figure 3d. The mRNA expression of *Ifnβ1* and *Mx1* in B78 tumors at day 15 after initiation of indicated treatments. (n=5)

Figure S14. The mRNA expression of *Ifnβ1* and *Mx1* in the B78 tumors at day 15 after PIC or control treatment. (n=5)

3) Figure 2: wouldn't the particles also sensitize DC and TAM to RT? Why a clonogenic assay is not shown for RAW cells?

Response: Although the myeloid cells (such as dendritic cells and macrophages) are relatively radio-resistant compared to tumor cells, there is a possibility that PIC sensitizes dendritic cells and macrophages to RT, and some of the dendritic cells and macrophages are killed by radiation. However, dendritic cells and macrophages can be recruited to the tumor sites by the pro-inflammatory cytokines generated after radiation. Our flow cytometry data shown in **Figure 4** and **Figure S18a** demonstrate that tumor infiltration by dendritic cells and macrophages was not weakened in the RT and PIC+RT treated tumor at day 15 after RT. Given this, we did not study the radio-sensitization of PIC on dendritic cells and macrophages and instead focused on evaluating modulation of the tumor immune microenvironment by PIC.

Figure 4e-4g. The percentage of (e) CD103⁺CD11b⁺ cDC1s and (f) CD11b⁺CD103⁻ cDC2s in CD11c⁺MHCII⁺ DCs in B78 tumors. (g) The percentage of CD317⁺ pDCs in CD11c⁺ cells in B78 tumors. (n=5)

Figure S18a. Quantification of F4/80⁺ cells in CD11b⁺CD45⁺ myeloid cells in B78 tumors after indicated treatments. (n=5)

4) Figure 2i-j: it is unclear how important is the induction of IFN-I in B78 melanoma cells if it takes 7 days to detect it, while in RAW cells it occurs after 24 hours. Also, how many cells survive at 7 days? Are the cells senescent and do they produce other inflammatory factors?

Why ION seems to radiosensitize B78 more than the PCFe at doses > 6 Gy?

Response: In a previous study (PMID: 33995649), we reported kinetic differences in the time to peak activation of a type I IFN response downstream of the STING pathway in different cell lines including the observation of a day 7 peak in B78 melanoma. The molecular basis for such variation remains unclear. In a separate manuscript combining a form of RT with ICBs, we showed that despite this delayed time to maximal IFN-I activation, this pathway was critical to the cooperative interaction between RT and ICBs (PMID: 34261797). For the in vitro qPCR experiment in this manuscript (**Figure 2b**), we plated 2.5x10⁵ cells in each plate for the collection at day 7. And at day 7, we counted the cells and found there were about 1.8x10⁵ live cells in each plate before further analysis. As shown in **Figure 2b** and **Figure S6a**, we found the *Ifnβ1* expression in RT-treated B78 cells peaked at day 7 post-RT, which was consistent with our previous findings (PMID: 33995649). PIC augmented the *Ifnβ1* expression in radiated B78 cells, which we and others have shown to be critical to the in situ vaccination effect of RT.

Figure S6a and 2b. The relative mRNA expression of *Ifnβ1* in B78 cells at day 1, 4 and 7 after initiation of indicated treatments. (n=3) PIC: 4.67 μg/mL; RT: 12 Gy.

When the radiation dose was higher than 6 Gy, there was no significant difference in the radio-sensitization effects of PIC and ION (at 6 Gy, PIC vs ION: p=0.82; at 9 Gy, PIC vs ION: p=0.235). Since we focused on the effect of PIC in this study, we removed ION from the clonogenic assay in the revised manuscript.

5) Experiments in vitro with macrophages are performed using a cell line. Are RAW cells representative of the behavior of primary macrophages? Do they express TLR9?

Response: We agree with the reviewer that the use of this cell line was a weakness. We therefore have removed the in vitro qPCR data with RAW264.7 cells, which demonstrated the macrophage polarization effect of PIC. In place of this data in the revised manuscript, we have evaluated this effect in vitro using primary macrophages that are derived from bone marrow (BMDMs). The data was included in new **Figure 2e-2h** and **Figure S7-S8**. We found that RT increased the expression of CD206 and CD163, two M2 macrophages markers, in BMDMs at day 4 post-treatment; PIC treatment significantly decreased the expression of these two markers (CD206 and CD163) in BMDMs or radiated BMDMs. After calculating the expression ratios of M1:M2 macrophages markers, we found the ratios of CD80: CD206 and CD80: CD163 expression were much higher in PIC+RT treated BMDMs, when compared to RT treated cells or untreated control cells at day 4 after treatment. Moreover, PIC enhanced the ratios of M1:M2 macrophages (M1: CD80⁺CD206⁻; M2: CD206⁺CD80⁻) in RT-treated BMDMs, further supporting the role of PIC in favorably modulating the effects of RT on macrophage polarization.

Figure 2e-2g. The MFI of CD206 and CD163 on CD11b⁺F4/80⁺ BMDMs at day 1 and day 4 after the cells were treated with (e) PIC and (f) RT+PIC. (g) The ratios of CD80 MFI to CD206 MFI on CD11b⁺F4/80⁺ BMDMs at day 1 and day 4 after indicated treatment. (n=4)

Figure S7b. The ratios of CD80 MFI to CD163 MFI on CD11b⁺F4/80⁺ BMDMs at day 1 and day 4 after indicated treatments. RT: 12 Gy. PIC: 4.67 µg/mL. (n=4)

Figure S8b and S8d. The quantification of M1-like macrophages (CD80⁺CD206⁻) and M2-like macrophages (CD206⁺CD80⁻) among CD11b⁺F4/80⁺ BMDMs and their ratios at (b) day 1 and (d) day 4 after indicated treatments. (n=4)

Figure 2h. The ratios of M1: M2 macrophage in CD11b⁺F4/80⁺ BMDMs at day 4 after indicated treatment. M1-like: CD80⁺CD206⁻; M2-like: CD206⁺CD80⁻. (n=4) PIC: 4.67 µg/mL; RT: 12 Gy.

6) Figure 3: the radiation dose used in vivo is 12 Gy while in vitro is 8 Gy. It is unclear what is the radiosensitizing factor of the particles at this dose. Panel 3c: it is unclear how CD8 T cells are quantified: percentage of CD45⁺ cells? Percentage of CD3⁺ cells? Total number is needed here, and the changes in tumor volume need to be shown for the different treatment groups. Also, why CD4 T cells are ignored? What is the ration of CD8/CD4 effector versus Treg?

Response: Thank you for these questions. We repeated and supplemented some in vitro experiments (qPCR using B78 cells, flow cytometry of BMDMs and immunofluorescence of B78 cells) in the revised manuscript and used 12 Gy in these experiments. Both the immunofluorescence and qPCR studies using B78 cells demonstrated the radiation-enhancing effects of PIC at 12 Gy. From the immunofluorescence images shown in **Figure 1i-1j**, PIC alone showed negligible effects in inducing DNA damage, but significantly augmented DNA damage when it was added to the B78 cells before radiation. From the qPCR data shown in **Figure S6a** and **Figure 2b**, PIC alone did not increase the *Ifnβ1* expression in B78 cells, but the pre-treatment with PIC significantly increased the *Ifnβ1* expression in radiated B78 cells. These results suggested the radiation-

enhancing effects of PIC at 12 Gy. We used 12 Gy in vivo as we have considerable experience with this dose for in situ vaccination in this tumor model (PMID: 27197149; 33858849; 32849544). We have previously demonstrated that alternative dosing regimens including 8 Gy fractions can also be used to elicit in situ vaccination in this model. We elected to use 12 Gy here as we have previously optimized the timing of RT + anti-CTLA-4 combination in this model at this dose (PMID: 27197149).

Figure 1i-1j. (i) The immunofluorescence images of B78 cells after indicated treatment (RT: 12 Gy; PIC: 4.67 μg/mL). (j) Quantification of foci of γH2AX as shown in (i). 50 cells in each group were analyzed with ImageJ. PIC was added to the cells 4 hours before radiation, and fresh culture media was exchanged for PIC treatment media 1 hour after radiation. The immunofluorescence images were taken 24 hours after radiation.

Figure S6a and 2b. The relative mRNA expression of *Ifnβ1* in B78 cells at day 1, 4 and 7 after initiation of indicated treatments. (n=3) PIC: 4.67 μg/mL; RT: 12 Gy.

We repeated the flow cytometry studies on disaggregated tumor specimens following in vivo treatments and analyzed more types of tumor-infiltrating immune cells in the revised manuscript. The changes of tumor volume during treatment for the mice used in the flow cytometry study have been included in **Figure S15**. The denominators to calculate the percentage of immune cells have been indicated in the y-axis in each figure. As requested by the reviewer, we provided the numbers of various immune cell lineages in tumor samples analyzed by flow cytometry in **Table S2** and **Table S3** of the supplemental document. We did not compare the absolute numbers of various immune cells between different treatment groups, because we did not use the whole tumor samples for this flow analysis. We divided the tumor samples for staining innate immune cells and adaptive immune cells, respectively, and took out some cells for full-staining and FMOs (full-staining minus one) staining. Therefore, we presented the data by comparing the relative percentage of certain types of immune cells between different treatment groups and we re-worded the corresponding description to indicate the denominators when analyzing immune cells in the revised manuscript. Although the relative percentage is employed by many researchers to analyze the flow

cytometry data, we agree with the reviewer that it is important that data be displayed and interpreted in accordance to whether it was analyzed as absolute/total cell numbers or as relative/percentage changes. Either or both approaches can be valuable, but we agree that interpretation needs to reflect to approach that was used. Because of variations in total cells analyzed and because we do not process the whole tumor for flow cytometry, we feel most comfortable displaying the relative changes in immune cell populations and per the reviewers request we have also added total cell count data in **Tables S2** and **S3**. In the new flow cytometry data, the percentage of CD8⁺ T cells and CD4⁺ T cells were calculated out of CD3⁺CD45⁺ cells. The percentage of Tregs (CD25⁺FOXP3⁺) were calculated out of CD4⁺ T cells. We found PIC+RT treatment enhanced the percentage of both CD8⁺ and CD4⁺ T cells and decreased the percentage of Tregs when compared to RT treated group or untreated control group (**Figure 4i-4k**).

By calculating the cell number ratios of effector CD4⁺ or CD8⁺ T cells (CD44⁺CD4⁺CD3⁺CD45⁺ and CD44⁺CD8⁺CD3⁺CD45⁺) versus Tregs (CD25⁺FOXP3⁺CD4⁺CD3⁺CD45⁺), we found RT significantly decreased these ratios. But when PIC was injected in radiated tumors, these ratios increased to levels that were similar to control mice (**Figure S23**).

Figure S15. The growth of B78 melanoma tumors in syngeneic mice (left) and average tumor volumes at day 14 (right) after indicated treatments. The mice were euthanized at day 15 for flow cytometry analyses of tumors and TDLNs. (n=5) PIC: 140µg/100µL/dose; RT: 12 Gy.

Figure 4i-4k. The percentage of (i) CD8⁺ and (j) CD4⁺ cells in CD45⁺CD3⁺ T cells in B78 tumors. (k) The percentage of CD25⁺FOXP3⁺ Tregs in CD4⁺ T cells in B78 tumors. (n=5)

Figure S23. The cell number ratios of (a) CD4⁺ effector T cells:Tregs and (b) CD8⁺ effector T cells:Tregs in B78 tumors after indicated treatments. CD4⁺ effector T cells: CD44⁺CD4⁺CD3⁺CD45⁺; CD8⁺ effector T cells: CD44⁺CD8⁺CD3⁺CD45⁺; Tregs: CD25⁺FOXP3⁺CD4⁺CD3⁺CD45⁺. (n=5) PIC: 140µg/100µL/dose; RT: 12 Gy.

Table S2. The cell number data for flow cytometry analyses of innate immune cells from B78 tumors. (The figures are shown in **Figure 4b-4h** and **S18**)

Sample labels	Live cells	Myeloid cells	Macrophages	M1 macrophages	M2 macrophages	pDCs	CD80 ⁺ pDCs	cDC1	cDC2
Control-1	711242	92795	63274	1895	5332	1265	178	529	3650
Control-2	654187	82500	56437	989	3295	745	79	509	4903
Control-3	630078	68331	47612	1598	4105	1350	211	327	3735
Control-4	665761	77567	51967	1652	2775	960	97	447	4948
Control-5	771599	68742	46944	845	3609	509	42	279	2427
RT-1	594992	137753	106056	4765	16117	6716	901	1222	7189
RT-2	359319	61320	49818	2734	8908	5242	869	541	8413
RT-3	532121	86181	75594	5522	10854	5084	637	726	4725
RT-4	494551	64257	45407	2878	8900	7826	1458	412	4941
RT-5	450643	87702	65644	2786	8995	5043	833	516	11326
PIC+RT-1	427793	78873	43160	2608	3172	2254	413	328	2003
PIC+RT-2	608191	129232	105789	7566	20887	14215	2769	1991	4825
PIC+RT-3	435215	87134	56004	3558	3456	2545	438	413	2448
PIC+RT-4	728356	153189	78271	5455	7856	4303	535	1980	5098
PIC+RT-5	705359	140193	76108	4166	5605	4045	845	555	2925

Myeloid cells: CD45⁺CD11b⁺; Macrophages: CD45⁺CD11b⁺F4/80⁺;

M1 macrophages: CD45⁺CD11b⁺F4/80⁺CD80⁺CD206⁻; M2 macrophages: CD45⁺CD11b⁺F4/80⁺CD206⁺CD80⁻;

pDCs: CD45⁺CD11c⁺CD317⁺; cDC1: CD45⁺CD11c⁺MHCII⁺CD103⁺CD11b⁻;

cDC2: CD45⁺CD11c⁺MHCII⁺CD11b⁻CD103⁻

Table S3. The cell number data for flow cytometry analyses of adaptive immune cells from B78 tumors. (The figures are shown in **Figure 4i-4o** and **S20-S23**)

Sample labels	Live cells	T cells	CD4 ⁺ T cells	CD8 ⁺ T cells	Tregs	Effector CD4 ⁺ T cells	Effector CD8 ⁺ T cells	Activated CD4 ⁺ T cells	Activated CD8 ⁺ T cells
Control-1	575638	64391	6167	2136	520	4750	1778	3142	1111
Control-2	496732	84314	7074	5441	864	6026	5131	3582	3228
Control-3	490462	113835	6700	7709	504	6221	7495	3924	3825
Control-4	525719	130090	6935	7651	560	6335	7409	3576	3461
Control-5	575461	87590	5411	4662	418	4885	4461	3564	2751
RT-1	455784	70180	19881	3905	2753	14873	3171	12686	2360
RT-2	334984	93720	12715	5745	1640	11561	5265	9145	3661
RT-3	460347	128827	14716	8037	1632	13287	6686	10795	5256
RT-4	439941	128199	21171	8738	3143	19276	7658	16424	6361
RT-5	380795	138363	22326	13160	2901	21477	12128	14899	6328
PIC+RT-1	354867	75672	14231	8798	1270	13065	7159	9175	5583
PIC+RT-2	523847	82994	28682	6409	3109	24211	5037	22558	4889
PIC+RT-3	387528	140353	15591	11301	1374	14613	10366	10058	5763
PIC+RT-4	570609	183122	60942	35054	2962	36019	18655	23758	11084
PIC+RT-5	597722	109240	16718	13769	1521	14783	12715	10094	7539

T cells: CD45⁺CD3⁺; CD4⁺ T cells: CD45⁺CD3⁺CD4⁺; CD8⁺ T cells: CD45⁺CD3⁺CD8⁺; Tregs: CD45⁺CD3⁺CD4⁺CD25⁺FOXP3⁺;

Effector CD4⁺ T cells: CD45⁺CD3⁺CD4⁺CD44⁺; Effector CD8⁺ T cells: CD45⁺CD3⁺CD8⁺CD44⁺;

Activated CD4⁺ T cells: CD45⁺CD3⁺CD4⁺CD69⁺; Activated CD8⁺ T cells: CD45⁺CD3⁺CD8⁺CD69⁺

Panel 3h: given the know importance of cDC1, and of their activation state in the tumor for effective anti-tumor immune rejection (PMID 30352680), a more informative analysis of DC phenotype is required. In addition, their frequency is expressed as percentage w/o specifying the denominator.

Response: We repeated the flow cytometry studies as noted and analyzed the phenotypes of dendritic cells in tumors. As shown in new **Figure 4e** and **4f**, RT treatment did not significantly influence the percentages of cDC1s (CD103⁺CD11b⁻) and cDC2s (CD11b⁺CD103⁻) out of CD11c⁺MHCII⁺ cells in tumors, but PIC+RT treatment significantly increased the levels of cDC1s and lowered the levels of cDC2s when compared to both RT treated tumor and untreated control groups. Since cDC1s are important antigen presenting cells in

activating cytotoxic CD8⁺ T cells, and cDC2s were reported to be involved in the activation of Th2 immune response, this effect of PIC+RT in elevating the cDC1s levels and lowering the cDC2 levels may be favorable for priming a potent TH1 cytotoxic T cell response against tumor cells.

In addition to cDC1 and cDC2, we found RT and PIC+RT significantly increased the percentage of CD317⁺ pDCs out of CD11c⁺ cells (**Figure 4g**). TLR-9 is chiefly expressed by pDCs and can be activated by CpG in PIC to facilitate the secretion of IFN-I. So, the increased pDCs may contribute to the high IFN-I mRNA expression in the tumors after PIC+RT treatment.

The denominators to calculate the percentage of immune cells have been indicated in the y-axis in each figure.

Figure 4e-4g. The percentage of (e) CD103⁺CD11b⁻ cDC1s and (f) CD11b⁺CD103⁻ cDC2s in CD11c⁺MHCII⁺ DCs in B78 tumors. (g) The percentage of CD317⁺ pDCs in CD11c⁺ cells in B78 tumors. (n=5)

Panels 3i-n: the data are difficult to interpret w/o knowing what is the tumor volume at day 7 and 15. Also, the increase in Mhc-1 expression could be reflecting an increased expression on cancer cells and/or the increased infiltration of the tumor by immune cells that express more Mhc-1 than the cancer cells. This bulk analysis is not informative.

Response: The changes of tumor volume during treatment for the mice used in the qPCR study and the flow cytometry studies have been added in **Figure S12** and **Figure S15**. Because we were not able to determine from bulk tumor mRNA whether the increased Mhc-1 expression in the PIC+RT group was from upregulation in tumor cells or increased tumor-infiltration of immune cells, we removed that qPCR data and the corresponding discussion to avoid confusion.

Figure S12. (a) Scheme for the treatment. The tumor growth curves and average tumor volumes of B78 melanoma bearing mice at (b) day 6 and (c) day 14 after indicated treatments. The mice were euthanized at (b) day 7 and (c) day 15 for qPCR analysis of bulk tumor samples. (n=5) PIC: 140µg/100µL/dose. RT: 12 Gy.

Figure S15. The growth of B78 melanoma tumors in syngeneic mice (left) and average tumor volumes at day 14 (right) after indicated treatments. The mice were euthanized at day 15 for flow cytometry analyses of tumors and TDLNs. (n=5) PIC: 140µg/100µL/dose; RT: 12 Gy.

7) Figure 4: why PCFe injections are repeated after RT if the main effect is to increase the radiation sensitivity to RT? Is this necessary?

Response: PIC was designed to achieve multiple different functions, including radio-sensitizing tumor cells, absorbing tumor-associated antigens and promoting their uptake by antigen presenting cells, stimulating TLR-9, and re-polarizing tumor-associated macrophages. According to the Refs (PMID: 28643452, 22842461, 29556359), the radiosensitizing effect of IONs can be attributed to the catalytic effects of the released iron ions and the active surfaces of IONs under RT, leading to the generation of ROS in cancer cells. So, PIC needs to be injected before radiation to perform its radio-sensitizing effect and generate ROS. Therefore, we

administered the first injection of PIC one day before radiation. After radiation, the released inflammatory cytokines and chemokines can recruit more immune cells to the tumor sites. PIC will capture the released tumor antigens and improve their uptake in antigen presenting cells, thus promoting the antigen presentation and the development of anti-tumor T cell immunity. Another intended function of PIC is modulating the effects of RT on TAMs. These effects of PIC will create a more immune-supportive tumor microenvironment but are expected to occur at some delay following RT. In our previous study, we found the infiltration of cytotoxic T cells in B78 melanoma increased between days 4-12 after RT (PMID: 29748391). So, in addition to the first injection at day 0, we performed injections at day 3, 6 and 9 to enable presence of PIC during these critical periods when its intended functions are needed. We have added discussion of this on page 19 and page 25 (in vivo treatments) in the revised manuscript and in future studies we will be investigating the importance of each injection time point, the possible need for additional injections, and possible approaches to limit the number of injections including the use of hydrogels for formulation and sustained release of PIC.

8) Figure 6 f and g: why survival is not different between RT+C4, RT+PCF3 and RT+C4+PcFe given that only the latter leads to a much better local control of the irradiated and the abscopal tumor? Is this control short lived? The expected radio-sensitization by the nanoparticles in vivo is not obvious.

Response: We updated the survival data of the B78 two-tumor model to extend to day 90 in **Figure 7c** in the revised manuscript. The PIC+RT+anti-CTLA-4 treatment group showed significantly improved mice survival at day 90 when compared to control group, RT+anti-CTLA-4 group and PIC+anti-CTLA-4 group. Although there is no statistical significance ($p=0.058$), the PIC+RT+anti-CTLA-4 treatment group showed a trend toward improved survival compared to PIC+RT group. The PIC+RT+anti-CTLA-4 triple treatment significantly suppressed the growth of both primary and distant tumors in all mice and at least in some mice this effect was not short lived as the 4/9 mice rendered tumor-free by this combinational treatment did not show any evidence of tumor recurrence. For the other 5/9 mice, one mouse was euthanized due to the appearance of a wound from ulceration and/or self-grooming at a tumor site; one mouse was found dead at day 28 after treatment without any visible tumor or tumor metastases and without any obvious sign of toxicity in organs; three mice showed tumor regression and later were euthanized for tumor recurrence/progression. Hence, although the mice in PIC+RT+anti-CTLA-4 treatment group showed slower tumor growth than those in other groups, indicating the potentiation effects of PIC in the treatment, we recognize that this anti-tumor immunity may need to be further improved in follow-up studies aiming to optimize the therapeutic regimen. We have added this discussion on page 19 in the revised manuscript.

Figure 7b-c. (b) Average tumor growth curves of both right and left tumor, and (c) survival rate of mice after the indicated treatment regimen. (n=9)

9) Figure S11: the sharp demarcation of vitiligo, which involved the lower but not upper body is intriguing, and begs the question whether there is a relationship with exposure to radiation. Although mice were shielded, was this area exposed to some scattered radiation?

Response: This is a good question. Yes, it is possible that radiation caused the appearance of vitiligo on the lower body. Although it has been reported that vitiligo can be a sign of reduced risk of disease progression and increased survival in pre-clinical and clinical studies of melanoma, the mechanism contributing to the observation of vitiligo are not clarified in our preclinical studies here. Therefore, we will further study this effect and because it is not critical to this manuscript, we have removed that data.

Reviewer #4 (Remarks to the Author): with expertise in radio-immunotherapy

In this manuscript, the authors developed a nanoparticle that are capable uptake antigens and increase antigen presentation to activate T cells. Below are my comments and suggestions.

1. “Moderate” word used in Abstract – line 24 –and in Introduction – line 55- is vague. Please define.

Response: “Moderate dose RT” is 8-12 Gy. We have added this information in the Abstract and Introduction.

2. Figure 1 in Results is not a result and it should be added as last Figure depicting the work model for this study.

Response: We revised the manuscript and the Figure 1 has been moved as suggested.

3. Is there a risk of radiation to increase the release of self antigens and promote autoimmunity using PCFe approach?

Response: The risk of generating autoimmunity is expected to be low in this approach. We covered sites outside the tumors with lead shield during radiation, which can minimize the release of self antigens from normal tissues. While self-antigens from normal cells may undergo antigen presentation, central tolerance should prevent adaptive immune recognition of these. To further minimize such risk, the PIC NP was injected intratumorally. According to the data shown in **Figure S32**, the PIC+RT and PIC+RT+anti-CTLA-4 treatment did not influence the biological functions of liver and kidney, which indicates that this regimen did not result in any identifiable autoimmunity. Of course this point will be further evaluated in the context of early phase clinical trials, which we are now beginning to design for PIC + RT. We have added brief discussion of these points on page 19 in the revised manuscript.

4. “Preparation, characterization of PCFe” – Results – line 129 – 140 should be in Methods not in Results.

Response: In the revised manuscript, we keep the “preparation and characterization of PIC” in the Results section because it is an important transition from the design of PIC to its biological evaluations. For example, we optimized the weight ratios of different components (PLL, ION and CpG) to make the prepared NP with highly positive zeta potential, so it can capture tumor antigens more efficiently, which was demonstrated to be critical in promoting the priming of a potent T cell immunity in the later experiments (**Figure 1-2**).

5. Figure 2K – How RT alone did not increase IFN β 1 compared to control? Based on previous studies, we should see increase after radiation.

Response: We repeated this experiment by radiating the B78 melanoma cells with 12Gy of radiation (we used 8Gy before). The data was shown in **Figure 2b** and **Figure S6a** in the revised manuscript. We found RT (12Gy) significantly increased the *Ifn β 1* expression in B78 cells at day 7 after treatment, which was consistent with our previous finding that the activation of IFN-I via cGAS/STING pathway in B78 cells peaked at day 7 post-treatment (PMID: 33995649). Further, we found that pre-treatment with PIC significantly augmented this IFN-I immune response in PIC+RT treated B78 cells when PIC was given prior to RT. This effect could not be replicated by the post-treatment with PIC after RT, further supporting a radio-sensitization effect of PIC.

Figure S6a and 2b. The relative mRNA expression of *Ifnβ1* in B78 cells at day 1, day 4 and day 7 after indicated treatment. (n=3)

6. *Ifnβ1* should be also measured using Elisa.

Response: We tested the levels of IFN-β secreted from dendritic cells using ELISA to evaluate the ability of PIC to stimulate TLR-9. It is reported that TLR-9 is chiefly expressed by plasmacytoid dendritic cells (pDCs). We found PIC treatment significantly increased the expression of co-stimulatory CD80 and CD86 on pDCs and induced the secretion of IFN-β, suggesting the stimulation of TLR-9 by PIC (**Figure 2i** and **Figure S9a**).

Figure 2i. The MFI of CD80 and CD86 on CD317⁺CD11c⁺ pDCs after indicated treatments. (n=3) PIC: 4.67 μg/mL; CpG: 0.5 μg/mL.

Figure S9a. The concentration of IFN-β secreted from CD11c⁺ dendritic cells after indicated treatment for 24 hours. (n=3) CpG: 0.5 μg/mL; PIC: 4.67 μg/mL.

7. CD206, CD204 and CD163 is more reliable marker for M2 than Arg1 and it would be interesting to analyze this marker after PCFe+RT. In addition, RAW cell line is very different from primary macrophages and the results maybe be very different compared to RAW.

Response: We removed the in vitro qPCR data of RAW264.7 cells that demonstrated the macrophage polarization effect of PIC, and evaluated this effect in vitro using primary macrophages that derived from bone marrow (BMDMs) in the revised manuscript. The data was supplemented in **Figure 2e-2h** and **Figure S7-S8**. We found RT increased the expression of CD206 and CD163, two M2 macrophages markers, in BMDMs at day 4 post-treatment; PIC treatment significantly decreased the expression of these two markers (CD206 and CD163) in BMDMs or radiated BMDMs. After calculating the expression ratios of M1: M2 macrophages markers, we found the ratios of CD80: CD206 and CD80: CD163 expression were higher in PIC+RT treated BMDMs, when compared to RT treated cells or untreated control cells at day 4 after treatment.

Figure 2e-2g. The MFI of CD206 and CD163 on CD11b⁺F4/80⁺ BMDMs at day 1 and day 4 after the cells were treated with (e) PIC and (f) RT+PIC. (g) The ratios of CD80 MFI to CD206 MFI on CD11b⁺F4/80⁺ BMDMs at day 1 and day 4 after indicated treatment. (n=4)

Figure S7b. The ratios of CD80 MFI to CD163 MFI on CD11b⁺F4/80⁺ BMDMs at day 1 and day 4 after indicated treatments. RT: 12 Gy. PIC: 4.67 µg/mL. (n=4)

Figure S8b and S8d. The quantification of M1-like macrophages (CD80⁺CD206⁻) and M2-like macrophages (CD206⁺CD80⁻) among CD11b⁺F4/80⁺ BMDMs and their ratios at (b) day 1 and (d) day 4 after indicated treatments. (n=4)

Figure 2h. The ratios of M1: M2 macrophage in CD11b⁺F4/80⁺ BMDMs at day 4 after indicated treatment. M1-like: CD80⁺CD206⁻; M2-like: CD206⁺CD80⁻. (n=4) PIC: 4.67 µg/mL; RT: 12 Gy.

8. Since RT+PCFe increased Arg1 and the method measuring both Arg1 and Nos2 is relative quantification, the data do not support the conclusion related to M2>M1 polarization effect in this study.

Response: We agree this was poorly worded in the prior manuscript and in addition to updating the data with BMDMs as noted above, we have updated our description of this data to be more accurate.

New flow cytometry data in vivo and in vitro also supports an effect of PIC in creating a more favorable

relative ratio of M1: M2 TAMs in the radiated tumor microenvironment. As shown in **Figure 4b-4d** and **Figure S18**, RT enhanced the percentage of both M1 macrophages (CD80⁺CD206⁻) and M2 macrophage (CD206⁺CD80⁻). PIC maintained the elevated levels of M1 macrophages but downregulated the levels of M2 macrophages when it was injected into the radiated tumors, thus increasing the ratios of M1: M2 macrophages. Higher ratios of CD80: CD206 expression were also observed in PIC+RT treated tumors compared to those treated with RT. The data shown in **Figure 2e-2h** and **Figure S7-S8** demonstrated that PIC treatment decreased the M2 macrophage markers (CD206 and CD163), enhanced the ratios of M1: M2 macrophage markers (CD80: CD206 and CD80: CD163), and increased the ratios of M1: M2 macrophages. (Data was shown in Comment #7)

Figure 4. The percentage of (b) CD80⁺CD206⁻ M1-like macrophage and (c) CD206⁺CD80⁻ M2-like macrophage in CD11b⁺F4/80⁺ macrophages in B78 tumors. (d) The ratios of M1:M2 macrophages in B78 tumors after indicated treatments. (n=5)

Figure S18d. The mean fluorescence intensity (MFI) of CD80 and CD206, and their ratios relative to total CD11b⁺F4/80⁺ macrophages in B78 tumors after indicated treatments. (n=5) PIC: 140µg/100µL/dose; RT: 12 Gy.

9. Figure S3. – Merged Images have different adjust compared to DAPI and FITC. Images should have exact the same adjust for all groups.

Response: We checked the adjust and updated the confocal images in **Figure S5**.

Figure S5. The confocal laser scanning microscopy (CLSM) images of B78 melanoma cells and RAW264.7 cells after treated with FITC labeled PIC for 2h.

10. “PCFe immunomodulates radiated tumor cells and the radiated tumor microenvironment in vivo” line 211- it should be irradiated and not ‘radiated”

Response: We thank the reviewer for pointing this out and have made the correction as indicated.

11. In vivo studies- – how the authors picked radiation dose of 12 Gy?

Response: Moderate dose, single fraction radiation (8-12Gy) has been reported to be optimal for in situ vaccine activation (PMID: 33827904). We have previously selectively tested various dose regimens for activating an in situ vaccination effect in the B78 melanoma model and from that work we have considerable experience with using 12 Gy. We have optimized the timing of anti-CTLA-4 and the timing of specimen collection for certain immune responses following 12 Gy RT in the B78 melanoma model (PMID: 27197149; 33858849; 32849544). Therefore we chose to work with this dose in vivo primarily because of our prior experience and optimization efforts with this dose.

REVIEWERS' COMMENTS

Reviewer #2 (Remarks to the Author):

The revised manuscript adds substantial new data further confirming key points of the initial submission. I appreciate the effort made by the authors to thoroughly respond to the reviewer comments. I find the revised manuscript substantially improved.

Reviewer #3 (Remarks to the Author):

The authors have answered satisfactorily all of my prior questions and the manuscript is greatly improved.

Reviewer #4 (Remarks to the Author):

The authors satisfactorily answered all my questions.

Reviewer #5 (Remarks to the Author):

Radiotherapy has been demonstrated to have synergistic effect with different forms of cancer immunotherapy including CpG as well as other cancer vaccination therapy. In this study, Morris et al. sought to develop a multifunctional nanoparticle called "PIC" which is composed of polylysine, iron oxide and CpG. The multifunctional nanoparticles were verified to amplify local anti-tumor immune responses through activation of cGAS/STING pathway and modulation of immunogenic tumor phenotypes when combined with radiotherapy (RT). Overall, this multifunctional nanoparticle holds great promise for combination cancer therapy and can be of substantial clinical value for cancer immunotherapy. However, there are still some points that should be addressed by the authors:

1. CpG plays a key role in immune stimulation by activating TLR9 cascade. It is unclear whether the combination therapy changes TLR9 signaling pathway.
2. The authors claim that the multifunction particles can absorb cancer antigens via electrostatic interactions. In vitro studies that mimic neoantigen or damage associated molecular patterns (DAMPs) absorption should be performed to test the hypothesis.

Reviewer #5 (Remarks to the Author):

Radiotherapy has been demonstrated to have synergistic effect with different forms of cancer immunotherapy including CpG as well as other cancer vaccination therapy. In this study, Morris et al. sought to develop a multifunctional nanoparticle called "PIC" which is composed of polylysine, iron oxide and CpG. The multifunctional nanoparticles were verified to amplify local anti-tumor immune responses through activation of cGAS/STING pathway and modulation of immunogenic tumor phenotypes when combined with radiotherapy (RT). Overall, this multifunctional nanoparticle holds great promise for combination cancer therapy and can be of substantial clinical value for cancer immunotherapy. However, there are still some points that should be addressed by the authors:

1. CpG plays a key role in immune stimulation by activating TLR9 cascade. It is unclear whether the combination therapy changes TLR9 signaling pathway.

Response: We agree with the reviewer that TLR9 activation is a critical immune stimulatory mechanism for generating anti-tumor immunity and this motivated our incorporation of CpG into PIC. In the current manuscript, we demonstrate that PIC and CpG have comparable capacities for upregulating CD80 and CD86 on plasmacytoid dendritic cells (pDCs) in vitro (**Fig 2i**), an effect that occurs downstream from TLR9 activation by CpG. This data suggests that the formulation of PIC from CpG+PLL+ION does not alter the capacity of the component CpG to activate TLR9. Furthermore, we have recently published on the combination of RT and CpG + OX40 (<https://pubmed.ncbi.nlm.nih.gov/34868010/>). In that separate publication, we did not observe any apparent detrimental effect of RT on the immune stimulatory effects of CpG on TLR9 activation. In this current manuscript, we compared the capacity of RT and PIC+RT to upregulate CD80 on pDCs following in vivo treatment (**Fig 4h**). We observed a significant increase in the fraction of pDCs that were CD80+ with PIC + RT compared to control tumors, but we did not detect a significant difference with RT alone compared to untreated control. This data is consistent with an effect of PIC activating TLR9 on pDCs in vivo and in combination with RT. In future studies beyond the scope of this manuscript, we will more directly evaluate the capacity of PIC to activate TLR9 signaling in varied immune cell lineages in vitro and in vivo, either alone or in combination with RT.

Figure 2i. The MFI of CD80 and CD86 on CD317⁺CD11c⁺ pDCs after indicated treatments.

Figure 4h. The percentage of CD80⁺ cells in CD317⁺CD11c⁺ pDCs in B78 tumors at day 15 after indicated treatment.

2. The authors claim that the multifunction particles can absorb cancer antigens via electrostatic interactions. In vitro studies that mimic neoantigen or damage associated molecular patterns (DAMPs) absorption should be performed to test the hypothesis.

Response: In the current manuscript, we provide in vitro data demonstrating that PIC is capable of absorbing peptides released from dead tumor cells (**Fig 1g**). In addition, we demonstrate in vitro that

antigens absorbed to PIC are internalized in dendritic cells (**Fig 2j**). Further, in vivo we demonstrate that antigen adsorbed to PIC stimulates effective antigen cross presentation and an antigen specific CD8 T cell response that is significantly greater than that achieved with the antigen alone (**Fig S10**). We also show, that the PLL component of PIC – which is the only electrostatic component of PIC that enables antigen absorption – is necessary for this in vivo capacity of PIC to enhance antigen cross presentation (**Fig S10**). This data strongly supports the conclusion that PIC is capable of absorbing cancer antigens via electrostatic interactions either in vitro or in vivo and that this effect is critical to the role of PIC in augmenting antigen cross presentation in vivo. In future studies beyond the scope of this manuscript, we will further evaluate whether the tumor antigens /DAMPs bound by PIC are broadly representative of all those released from tumor cells following radiation therapy or whether a select subgroup of released DAMPs and antigens are particularly well suited to absorption on PIC, uptake in antigen presenting cells, and antigen cross presentation to T cells.

Figure 1g. The protein concentrations in B78 cell lysates after treatment with PIC (0.14 mg/mL) for 4 hours.

Figure 2j. The percentage of FITC-Ova positive cells among CD11c⁺ DCs at 24 h after treatment.

Figure S10. (a) Scheme for the co-culture of B16-SIINFEKL cells with splenocytes extracted from Ova, Ova/PIC, or Ova/CpG/ION injected mice. Quantification of CD44⁺, CD69⁺ and IFN γ ⁺ cells out of (b) CD4⁺CD3⁺CD45⁺ and (c) CD8⁺CD3⁺CD45⁺ cells in splenocytes by flow cytometry.